

# Global Annual Mean Atmospheric Histories, Growth Rates and Seawater Solubility Estimations of the Halogenated Compounds HCFC-22, HCFC-141b, HCFC-142b, HFC-134a, HFC-125, HFC-23, PFC-14 and PFC-116

Pingyang Li [1], Jens Mühle [2], Stephen A. Montzka [3], David E. Oram [4,5], Benjamin R. Miller [3] and Toste Tanhua [1]

[1]GEOMAR Helmholtz Centre for Ocean Research Kiel, Kiel, Germany
[2] Scripps Institution of Oceanography, University of California, La Jolla, San Diego, California, USA
[3]Earth System Research Laboratory, National Oceanic and Atmospheric Administration, Boulder, Colorado 80305, United
States
[4]National Centre for Atmospheric Science, School of Environmental Sciences, University of East Anglia, Norwich, NR4
7TJ, UK
[5]Centre for Ocean and Atmospheric Sciences, School of Environmental Sciences, University of East Anglia, Norwich, NR4
7TJ, UK

*Correspondence to*: Pingyang Li (pli@geomar.de)

**Abstract.** We present consistent annual mean atmospheric histories and growth rates for the mainly anthropogenic halogenated compounds HCFC-22, HCFC-141b, HCFC-142b, HFC-134a, HFC-125, HFC-23, PFC-14 and PFC-116, all potentially useful oceanic transient tracers (tracers of water transport within the ocean), for the Northern and Southern Hemisphere with the aim of providing input histories for these compounds. Where available we utilize observations of the halogenated com-
pounds made by the Advanced Global Atmospheric Gases Experiment (AGAGE), National Oceanic and Atmospheric Administration (NOAA) and University of East Anglia (UEA). Prior to the direct observational record we estimated the atmospheric history concentrations from other sources such as archived air measurements, firn air measurements and published model calculations. The results show that the atmospheric mole fractions for each species have been increasing since they were initially produced. Recently, their atmospheric growth rates are decreasing for HCFCs (HCFC-22, HCFC-141b and
HCFC-142b), increasing for HFCs (HFC-134a, HFC-125, HFC-23), and stable with small fluctuation for PFCs (PFC-14 and PFC-116). The atmospheric histories (source functions) and natural background values show that HCFCs (HCFC-22, HCFC-141b and HCFC-142b) and HFCs (HFC-134a, HFC-125 and HFC-23) have the potential to be oceanic transient tracers for the next few decades only because of the recently imposed bans on production. When the atmospheric histories of the compounds are not monotonically changing, the equilibrium atmospheric concentrations (and ultimately the age associated with
that concentration) calculated from their concentration in the ocean are not unique, reducing the potential as transient tracer. Moreover, HFCs have potential to be oceanic transient tracers for a longer period in the future than HCFCs as the growth rates of HFCs are increasing and those of HCFCs are decreasing in the background atmosphere. PFC-14 and PFC-116, however, have the potential to be the tracers for longer period in the future thanks to their extremely long lifetimes, steady atmospheric growth rates and no explicit ban. In this work, we also derive solubility functions for HCFC-22, HCFC-141b,
HCFC-142b, HFC-134a, HFC-125, HFC-23, PFC-14 and PFC-116 in seawater to facilitate the use as oceanic transient tracers. These functions are based on the Clark-Glew-Weiss (CGW) water solubility functions fit and salting-out coefficients estimated by the poly-parameter linear free energy relationships (pp-LFERs). Here we also provide three methods of seawater solubility estimation for more compounds.





## 1 Introduction

The oceanic transient tracers are chemical tracers with a time varying source or sink. Chlorofluorocarbons (CFCs) have been used as traditional oceanographic transient tracers because of their continuously increasing atmospheric concentrations until some years ago. They are powerful tools in oceanography where they are used to, for instance, deduce transport times, esti-
mate mixing rates between water masses, study formation rates of new water masses, and determine the anthropogenic carbon ($C_{ant}$) content of seawater (Weiss et al., 1985; Waugh et al., 2006; Fine, 2011; Schneider et al., 2012; Stöven et al., 2016). The production and consumption of CFCs has been phased out as a consequence of the implementation of the Montreal Protocol (MP) on Substances that Deplete the Ozone Layer (first from developed nations by 1996, followed by developing nations by 2010) designed to stop the degradation of the earth's ozone layer (Fig. 1). Their atmospheric mole fractions
have been decreasing since the mid-2000s (Carpenter et al., 2014; Bullister, 2015), and although CFCs are valuable variables to quantify deep water transport, the use of CFCs as oceanographic transient tracers is difficult for recently ventilated water masses. During recent decades sulfur hexafluoride ($SF_6$) has been added to the suite of transient tracers measured in the ocean (Tanhua et al., 2004; Bullister et al., 2006). Its atmospheric levels are still increasing and it's the most measured of all the potential tracers in the atmosphere. But $SF_6$ is banned as a tracer gas and in all applications except high-voltage switch-
gear since 1 January 2006 in Europe. Since a combination of transient tracers is needed to constrain ventilation (Waugh et al., 2002; Stöven et al., 2015), it is necessary to explore other transient tracers with positive growth rates for understanding mixing processes in the ocean.

Generally, several requirements for a useful oceanic tracer can be defined: the tracer concentration should have a well-established, transient, source function (or well-defined decay function); have a low or well-known natural background; be
conservative (not produced or destroyed in seawater) in the marine environment and be measured relatively inexpensively, accurately, and rapidly. Potential candidates as transient tracers that fulfill at least some of the requirements listed above include hydrochlorofluorocarbons (HCFCs) such as HCFC-22, HCFC-141b and HCFC-142b, hydrofluorocarbons (HFCs) such as HFC-134a, HFC-125 and HFC-23 and perfluorocarbons (PFCs) such as PFC-14 and PFC-116. As a first step of evaluating the utility of these compounds as oceanic transient tracers, we synthesize the atmospheric concentration history
and review the solubility in this study. An upcoming work will evaluate in-field data of these compounds.

**HCFC-22:** Chlorodifluoromethane ($CHClF_2$) is the most abundant HCFC in the global atmosphere. It was discovered in 1928 but little used until after the World War II. Its commercialization started in 1936 (Calm and Domanski, 2004). It has been used dispersedly as domestic and commercial refrigeration, spray-can propellant, extruded polystyrene foam industries (McCulloch et al., 2003; Jacobson, 2012), and non-dispersedly as the feedstock in fluoropolymer production (Miller et al.,
2010). HCFC-22 was first measured in the atmosphere in 1979 (Rasmussen et al., 1980), a pronounced increase of its abundance in the 1990s was found in both hemispheres because HCFC-22 became an interim replacement for CFC-12 since the late 1980s (Xiang et al., 2014). There are no known natural emission sources for HCFC-22 (Saikawa et al., 2012). A considerable amount of literature has been published on the atmospheric histories of HCFC-22. In 1993, Montzka et al. (1993) presented scattered NOAA network measurements and model results of historic mole fractions from a 2-box model for HCFC-
22 from 1980 to 1993, and these have since been updated and augmented with measurements from an air archive and box models to build an atmospheric history for HCFC-22 from 1955 to 2014 (Montzka et al., 2010a; Montzka et al., 2015). In 1998, Miller et al. (1998) reported measurements of HCFC-22 at Cape Grim, Tasmania over the period of 1978-1996 and at La Jolla, California over the period of 1992-1997 combined with the AGAGE 12-box model results from 1978 to 1997. In 2004, O'Doherty et al. (2004) reported the in situ AGAGE measurements made at Mace Head, Ireland and Cape Grim, Tas-
mania for HCFC-22 from 1998 to 2002. In 2012, Saikawa et al. (2012) have reported on updated AGAGE and NOAA observations and archived air measurements, combined with the MOZART model to present the atmospheric mole fractions for HCFC-22 from 1995 to 2009.



**HCFC-141b:** 1,1-dichloro-1-fluoroethane (CH$_3$CCl$_2$F) has been widely used as a foam-blowing agent in rigid polyurethane foams for insulation purposes and of integral skin foams for motor vehicle dashboards, steering wheels and soles of shoes as a replacement for CFC-11. It was also employed as a solvent for lubricants, coatings and cleaning fluids for aircraft mainte-nance and electrical equipment as a replacement for CFC-113 (Derwent et al., 2007). The industry production and use of HCFC-141b has greatly increased since the early 1990s, as did its global concentration (Montzka et al., 1994; Oram et al., 1995; McCulloch, 2003; Montzka et al., 2009).

**HCFC-142b:** 1-chloro-1,1-difluoroethane (CH$_3$CClF$_2$) has largely been emitted from extruded polystyrene board stock as a foam-blowing agent combined with small emissions from refrigeration applications as a replacement for CFC-12 (TEAP, 2003). Previous studies have reported the atmospheric mole fractions of both HCFC-141b and HCFC-142b. In 2002, Sturrock et al. (2002) reported the observations and reconstructed records for HCFC-141b and HCFC-142b based on firn air samples collected in 1993 and in 1997-98 in the Antarctic. In 1995, Oram et al. (1995) published a paper in which they de-scribed the concentrations of HCFC-141b and HCFC-142b in the Cape Grim air archive between 1978 and 1994. The flask and archived air measurements for both compounds from NOAA were reported by Montzka et al. (1994).

**HFC-134a:** 1,1,1,2-tetrafluoroethane (CH$_2$FCF$_3$) is the most abundant HFC in Earth's atmosphere. It was first synthesized by Albert Henne in 1936 (Matsunaga, 2002). Extensive production and emission of HFC-134a began in the early 1990s. It was used as a preferred refrigerant in domestic, commercial and automotive air conditioning and refrigeration to replace CFC-12. It is also used to a lesser extent as foam blowing agent, cleaning solvent, fire suppressant, and propellant in me-tered-dose inhales and aerosols (Simmonds et al., 2015). Continuous and substantially increasing atmospheric levels of HFC-134a were found over the past two decades (Xiang et al., 2014). A number of researchers have reported the atmospheric his-tory of HFC-134a. The observational record of HFC-134a started from near-zero levels in the background atmosphere (Oram et al., 1996). He presented the measurements for HFC-134a at Cape Grim, Tasmania over the period of 1978-1995 and at Mace Head, Ireland between July 1994 and May 1995. In the same year, Montzka et al. (1996b) reported initial measure-ments from the NOAA network for HFC-134a from the late 1980s to mid-1995, which have since been updated (Montzka et al., 2015). In 1998, Simmonds et al. (1998) presented measurements for HFC-134a recorded at Mace Head, Ireland from October 1994 to March 1997. In 2004, O'Doherty et al. (2004) showed the in situ AGAGE measurements made at Mace Head, Ireland and Cape Grim, Tasmania for HFC-134a from 1998 to 2002. In the same year, Culbertson et al. (2004) pub-lished a paper in which they described measurements of HFC-134a air archived at Cape Meares, Oregon, from 1978 to 1997, at Point Barrow, Alaska, from 1995 to 1998, and at Palmer Station, Antarctica, from 1991 to 1997.

**HFC-125:** Pentafluoroethane (CHF$_2$CF$_3$) is the fifth most abundant HFC (O'Doherty et al., 2009). It is used primarily in re-frigerant blends for commercial refrigeration applications, has a minor use in fire-fighting equipment as a replacement for halons. Atmospheric mole fractions of HFC-125 are also rising consistently as one of the substitutes of CFCs (O'Doherty et al., 2009). In 2009, O'Doherty et al. (2009) reported in situ measurements and archived air measurements from Mace Head and Cape Grim, combined with the AGAGE 2-D 12-box model for HFC-125 from 1977 to 2009. In 2015, Montzka et al. (2015) reported results of flask measurements of HFC-125 spanning from 2007 to 2013.

**HFC-23:** Fluoroform or trifluoromethane, (CHF$_3$) is a by-product from the industrial production of HCFC-22. Historically it has been considered as waste and simply vented to the atmosphere, although process optimization and abatement can elimi-nate most or all emissions. HFC-23 was used as a feedstock for Halon-1301 (CBrF$_3$) production. Small amounts are report-edly used in semiconductor (plasma etching) fabrication, in very low temperature refrigeration (dispersive), and in specialty fire suppressant systems (dispersive) (McCulloch and Lindley, 2007). HFC-23 was first detected in the background atmos-phere in 1978 (Oram et al., 1998). It continued to increase in the atmosphere despite the voluntary and regulatory efforts in developed nations and mandatory abatement measures in developing nations financially-supported by the United Nations Framework Convention on Climate Change (UNFCC) Clean Development Mechanism (CDM) project. In the past three dec-ades a number of researchers have reported the atmospheric mole fractions of HFC-23. In 1998, Oram et al. (1998) reported



measured and model-generated concentrations of HFC-23 at Cape Grim over the period 1978-2005. In 2004, Culbertson et al. (2004) published a paper in which they described measurements of HFC-23 air archived at Cape Meares, Oregon, from 1978 to 1997, at Point Barrow, Alaska, from 1995 to 1998, and at Palmer Station, Antarctica, from 1991 to 1997. The updated in situ AGAGE measurements and results from the AGAGE 2-D atmospheric 12-box chemical transport model for HFC-

23 from 1978 to 2010 have presented by Miller et al. (2010). A history derived from multiple firn-air sample collections was also published in 2010 (Montzka et al., 2010a).

**PFC-14:** Tetrafluoromethane or carbon tetrafluoride ($CF_4$) is the most abundant perfluorocarbon (PFC) in the earth's atmosphere and is one of the most long-lived tracer gases with an atmospheric lifetime of more than 50000 years. The presence of carbon tetrafluoride in the atmosphere was first deduced by Gassmann (1974) from analysis of contaminant levels of PFC-14

in high-purity krypton samples. The first atmospheric measurements of PFC-14 were made by Rasmussen et al. (1979). It has a background atmospheric concentration due to its natural source from the rocks and soils, especially tectonic activity (Deeds et al., 2015). The pre-industrial level was $34.05 \pm 0.33$ ppt for PFC-14 (Trudinger et al., 2016). The primary anthropogenic sources of PFC-14 are aluminum production and the semiconductor industry (Khalil et al., 2003; Mühle et al., 2010). Consequently atmospheric concentrations have doubled since the early 20th century.

**PFC-116:** Hexafluoroethane ($C_2F_6$) is another long-lived tracer gas with an atmospheric lifetime of at least 10000 yr. The tropospheric abundance of PFC-116 was first determined by Penkett et al. (1981). It has a small natural abundance (Mühle et al., 2010; Trudinger et al., 2016); the pre-industrial level has been estimated to be 0.002 ppt (Trudinger et al., 2016). A more gradual increase in emission for PFC-116 was found in 1900. Like $CF_4$ it is also emitted as a by-product of aluminum production and during semiconductor manufacturing.

The major atmospheric degradation pathway of HCFCs and HFCs is through reaction with hydroxyl radicals (OH) in the troposphere (Montzka et al., 2010b). The only identified atmospheric loss pathway of PFCs is through reaction with atmospheric ions (Hayman et al., 1994). The Ozone Depleting Potential (ODP), Global Warming Potentials (GWP) and atmospheric lifetimes for HCFC-22, HCFC-141b, HCFC-142b, HFC-23, HFC-134a and HFC-125 (Hodnebrog et al., 2013; Laube et al., 2013; SPARC, 2013) are listed in the Table 1. The production and use histories of CFC-12, $SF_6$, HCFCs, HFCs and

PFCs are plotted in Fig. 1. HCFCs have been regulated with the aim to cease production and consumption by 2020 for non-Article 5 (developed) countries and 2030 for Article 5 (developing) countries (although this only covers dispersive applications) and phase-out beginning with a freeze in 1996 for developed nations and in 2013 for developing nations under the MP and its more recent amendments. Because of the high GWP of HFCs, 197 countries recently committed to cutting the production and consumption of HFCs by more than 80 % over the next 30 years under the Kigali amendment of the MP. Devel-

oped countries will reduce HFC consumption beginning in 2019. Most developing countries will freeze consumption in 2024, some in 2028. This measure most likely will slow down, and eventually reverse the atmospheric concentration of HFCs, similar to what is now observed for the CFCs.

In order to explore if these halogenated compounds can be used as transient ocean tracers, their atmospheric history (source functions) and natural background values should be established. Previous work has reconstructed annually-averaged atmos-

pheric concentration histories for some tracer gases for use in tracer oceanographic applications. For example, Walker et al. (2000) reported annual mean atmospheric mole fractions for CFC-11, CFC-12, CFC-113, and $CCl_4$ for the period 1910-1998 and updated the data to 2008 at the website (http://bluemoon.ucsd.edu/pub/cfchist/). On the basis of Walker's work, Bullister (2015) reported atmospheric histories for CFC-11, CFC-12, CFC-113, $CCl_4$, $SF_6$ and $N_2O$ for the period 1765-2015. Previous work related to our target compounds, concentrated on the atmospheric history for a specific period, often at a high tem-

poral resolution. We have listed these works above. For our purposes we are interested in a consistent record of the full atmospheric history at annual temporal resolution. As Trudinger et al. (2016) presented the consistent atmospheric histories of PFC-14 and PFC-116 from 1900 to 2015, we only study the growth rates for these two compounds and evaluate their utility to be oceanic transient tracers.





In this study, drawing on previous literature and published data, we present annual mean atmospheric mole fractions (mid-February, mid-year and mid-August) and growth rates for HCFC-22, HCFC-141b, HCFC-142b, HFC-134a, HFC-125, HFC-23, PFC-14 and PFC-116 for both the Northern (NH) and Southern Hemisphere (SH). The mid-February (the average of monthly means in January, February and March) and mid-August (the average of monthly means in July, August and September) is chosen to coincide with the coldest part of the year in NH and SH, respectively, i.e. the time of (deep) water mass formation when ambient trace gas concentration are carried from the surface to the interior ocean. The reconstructed atmospheric histories have been compiled from a combination of air measurements and model calculations. Ambient air measurements published by the Advanced Global Atmospheric Gases Experiment (AGAGE), National Oceanic and Atmospheric Administration (NOAA) and University of East Anglia (UEA) are all considered in this study. For years prior to atmospheric observations, the reconstructed dry mole fractions for each species were provided by a combination of atmospheric models, firn air measurements and the analysis of archived air samples. The aim of this work is to synthesize existing data and model results into one consistent data product of atmospheric history with annual values useful for ocean tracer applications; it is not intended to replace more detailed atmospheric studies. All reported values in this study are dry air mole fractions reported through March or July 2017. In a similar work, Meinshausen et al. (2017) provided consolidated datasets of historical atmospheric concentrations (mole fractions) of 43 Greenhouse Gases (GHGs). Compared with this study, the differences and added value of this study is that we: 1) added the data from the UEA not included in the Meinshausen et al. (2017) study, 2) used the same AGAGE calibration scales by converting NOAA and UEA data to the AGAGE scale, 3) estimated the propagated uncertainties based on the original standard deviations of monthly means or data points, 4) used a different method for data fit, and 5) presented the atmospheric histories for winter (mid-February in NH and mid-August in SH) that is especially useful for ocean transient tracers studies.

In addition, we explore if these compounds can used as oceanic transient tracers, by reporting on the solubility characteristics for each of the gases. There were no literature estimates that directly provide solubility functions of all target compounds in seawater, only very limited studies (with several data points) on the solubility of these compounds in seawater have been reported. Scharlin and Battino (1995) published four solubility data points in 15-30 °C and a salinity of 35.086 g kg⁻¹ of PFC-14 in seawater. In the present analysis, the water and seawater solubility functions of HCFC-22, HCFC-141b, HCFC-142b, HFC-134a, HFC-125, HFC-23, PFC-14 and PFC-116 are derived by the Combined Method based on the combination of the Clark-Glew-Weiss (CGW) fit to estimate their water solubility function and the polyparameter linear free energy relationships (pp-LFERs) to estimate their salting-out coefficients. Three concluded methods (the Revised Method II only based on the pp-LEFRs, the Combined Method and the Experimental Method) on seawater solubility estimation are also provided for more compounds.

## 2 Data and Methods

### 2.1 AGAGE in situ Measurements and Instrumentation

In situ atmospheric measurements have been made by the Advanced Global Atmospheric Gases Experiment (AGAGE) (Miller et al., 1998; Prinn et al., 2000; O'Doherty et al., 2004; O'Doherty et al., 2009; Miller et al., 2010; Mühle et al., 2010; Prinn et al., 2016). The data are available at the AGAGE website (http://agage.eas.gatech.edu/data_archive/). AGAGE provides measurements of more than 40 compounds, whereas we focus only on HCFC-22, HCFC-141b, HCFC-142b, HFC-134a, HFC-125, HFC-23, PFC-14 and PFC-116 (Table 2). There are more than 10 AGAGE and affiliated stations globally, mostly located at coastal or mountain sites. Here we exclude all AGAGE stations at tropical latitudes which are periodically subjected to air masses originating in the other hemisphere (Walker et al., 2000). Observations at the AGAGE remote stations Mace Head, Ireland (MHD, 53.33° N, 9.90° W) and Trinidad Head, California (THD, 41.05° N, 124.15° W) were assumed to represent the 30-90° N atmospheric mole fractions, whereas observations at Cape Grim, Tasmania (CGO, 40.68° S,



144.69° E) represent the 30-90° S mole fractions. Very small latitudinal gradients in the AGAGE Mace Head and Trinidad Head observations of different compounds are present but assumed to be of minor importance to this work. These stations, their locations and the date ranges of the samples used in this study are listed in Table 2. The "pollution-free" monthly mean atmospheric mole factions and standard deviations for all target compounds are used in this study.

All ambient air measurements were carried out using two similar measurement technologies over time, both on the basis of the cryogenic pre-concentration with gas chromatography and mass spectrometry (GCMS) system. The initially used instrument was the ADS (adsorption-desorption system)-GCMS. In the early/mid-2000 it was replaced by the Medusa-GCMS with doubled sampling frequency, differing sample pre-concentration methodologies, extended compound selection, and improved measurement precisions. For more information on the instrumentation and the working standards, see (Simmonds

et al., 1995; Miller et al., 2008; Arnold et al., 2012). For the measurement precision, see studies (Simmonds et al., 2017a; Simmonds et al., 2017b).

## 2.2 AGAGE archived air

To extend the available concentration record back in time, NH and SH air archive sample concentrations were collected from AGAGE for HFC-125 and HFC-134a. The Southern Hemisphere Cape Grim Air Archive (CGAA) samples, which are clean

background "baseline" air, were collected at the Cape Grim Baseline Air Pollution Station in Cape Grim, Tasmania by the Commonwealth Scientific and Industrial Research Organization (CSIRO). The samples were cryogenically collected into 34 L electro-polished stainless steel canisters (Langenfelds et al., 1996) since 1978. The CGAA samples were analyzed based on Medusa-9 at CSIRO. All NH archived air samples were collected in America and analyzed at SIO on laboratory-based Medusa-GCMS (Medusa-1 and Medusa-7). These samples were mostly collected during clean air conditions. The data have

been filtered followed by statistical outlier exclusion (Cunnold, 2002).

All air archive data for HFC-134a from AGAGE are first reported for HFC-134a (Table 2). The AGAGE archived air measurements for HFC-125 reported by O'Doherty et al. (2009) are used in this study (Table 2). The CGAA archived air measurements for HCFC-22 are reported (Miller et al., 1998). The Southern Hemisphere CGAA measurements by Medusa-3 and Medusa-9 for HFC-23 from AGAGE reported by Miller et al. (2010) are also reported here (Table 2).

## 25    2.3 AGAGE firn air

The firn layer is unconsolidated snow overlaying an ice sheet. Large volumes (hundreds of liters) of air trapped in firn enable analysis of trace gases requiring larger volumes of air than is available from ice cores (Sturrock et al., 2002). The air can be extracted from present to around 100 years ago and measured to reconstruct the atmospheric composition using a firn diffusion model. The first measurements of HCFC-141b and HFC-134a in firn were made by Butler et al. (1999) and showed that

there are no natural sources for these compounds.

The firn air samples for HCFC-141b and HCFC-142b were collected from six depths at Law Dome, Antarctic in 1997-98 at DSSW20K site (Sturrock et al., 2002). The fin air data is estimated from the published figures (Table 2). The firn air samples were measured by the instrument ADS-GCMS.

## 2.4 NOAA flask measurements

Flask air measurements of the compounds considered in this study have been made by the National Oceanic and Atmospheric Administration (NOAA) as early as 1992 (Montzka et al., 1994; Montzka et al., 1996b; Montzka et al., 2009; Montzka et al., 2015). The data is available at the NOAA website (ftp://ftp.cmdl.noaa.gov/hats/). There are many NOAA and affiliated stations globally. In order to be consistent with the chosen AGAGE stations, only NOAA observations at Mace Head, Ireland (MHD, 53.3° N, 9.9° W, 42 meters above sea level (masl)) for HCFC-22, HCFC-141b, HCFC-142b and HFC-134a, and

Trinidad Head, USA (THD, 41.0° N, 124.1° W, 120 masl) for HFC-125 are used for representing atmospheric mole fractions



from 30-90° N. The observations at Cape Grim, Australia (CGO, 40.682° S, 144.688° E, 164 masl) represent the 30-90° S mole fractions. These stations, their locations and the date ranges of the samples used in this study are listed in Table 2.

Air samples are analyzed in the NOAA/ESRL/GMD Boulder laboratories by gas chromatography and mass spectrometry (GC-MS) techniques for HCFC-22, HCFC-141b, HCFC-142b, HFC-134a and HFC-125. More details are given in Montzka

et al. (1993), Montzka et al. (1994), Montzka et al. (1996b) and Montzka et al. (2015). The working standards and measurement precision are also reported in the above studies.

### 2.5 NOAA archived air

Archived air and cruise air measurements from both hemispheres for HCFC-141b and HCFC-142b are given in Thompson et al. (2004). The archived air samples for HCFC-141b and HCFC-142b were filled at the Niwot Ridge (NWR) since 1986. The

cruise air samples were collected during the Soviet-American Gas and Aerosol Experiment (SAGA) II cruise in the Pacific Ocean in 1987 (37° N-30° S, 160° W-170° W).

Archived air and cruise air measurements for HFC-134a were presented in Montzka et al. (1996b). The archived air samples were collected at Niwot Ridge (NWR). Samples were obtained shipboard during two cruises, one in the Pacific Ocean in 1987 (SAGA II above) and in 1994 (41.3° N-46.8° S, 126.8° W-75.8° W) and one in the Atlantic Ocean in 1994 (45.9° N-

47.5° S, 13.7° W-60.4° W).

### 2.6 UEA archived air

The Cape Grim air archive measurements from University of East Anglia (UEA) are updated following the original publications for HCFC-141b, HCFC-142b (Oram et al., 1995) and HFC-134a (Oram et al., 1996). The UEA has analyzed whole air samples from the Cape Grim Baseline Air Pollution Station in Tasmania, Australia since 1978. The Cape Grim archived air

contains trace gas records known to be representative of background air in the Southern Hemisphere. The archive samples are often the same samples that have been analysed by AGAGE (Sect. 2.2)

The archived air samples were cryogenically collected at Cape Grim, Tasmania and analyzed by GCMS for HCFC-141b, HCFC-142b and HFC-134a (Oram et al., 1995; Oram et al., 1996) (Table 2). The working standards and measurement uncertainty were also shown in above studies.

**2.7 Models**

In order to fill the gap before atmospheric measurements, the results from two published models, a 2-box model for HCFC-22 (Montzka et al., 2010a) and the AGAGE 2-D atmospheric 12-box chemical transport model for HFC-23 (Cunnold et al., 1983; Cunnold et al., 1994; Miller et al., 2010; Rigby et al., 2011) and for PFC-14 and PFC-116 (Trudinger et al., 2016), are also included in this study (Table 2).

The 2-box model for HCFC-22 from Montzka et al. (2010a) considers the atmosphere as two boxes; one box representing each hemisphere. Each hemisphere is assumed to be well mixed and a standing vertical gradient is assumed. The tropospheric lifetime for HCFC-22 is 13.6 years (Montzka et al., 1993). Montzka et al. (2010a) calculated the atmospheric mole fractions from 1944 to 2009 becoming available for HCFC-22 by assuming a constant 0.95 scaling of global emissions estimated by the Alternative Fluorocarbons Environmental Acceptability Study (AFEAS) emissions by in the 2-box model.

The AGAGE two-dimensional atmospheric 12-box chemical transport model for HFC-23, PFC-14 and PFC-116 contains four lower tropospheric boxes, four upper tropospheric boxes and four stratospheric boxes, with boundaries at 30° N, 0° and 30° S in the horizontal, and 500 and 200 hPa in the vertical. It was described in detail by these studies (Cunnold et al., 1983; Cunnold et al., 1994; Rigby et al., 2011) and was used to estimate the atmospheric emission of several compounds, such as CFC-11 and CFC-12, and to simulate atmospheric mole fractions. Miller et al. (2010) inferred the model mole fractions for

HFC-23 prior to AGAGE in situ measurements (1978-2009) from inversion using the 2-D 12-box model, but not back to

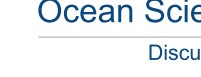 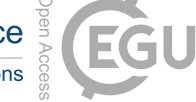

zero atmospheric concentrations. Trudinger et al. (2016) calculated the atmospheric mole fractions of PFC-14 and PFC-116 in each semi-hemisphere since 1900 by combining the data from ice core, firn, air archive and in situ measurements.

### 2.8 Calibration scale

The latest AGAGE absolute calibration scales for various trace gases are displayed on the AGAGE website

(http://agage.eas.gatech.edu/data_archive/agage/AGAGE_scale_2018_v1.pdf). The AGAGE in situ measurements were reported on the latest SIO absolute calibration scale for HCFC-22 (SIO-05), HCFC-141b (SIO-05), HCFC-142b (SIO-05), HFC-23 (SIO-07), HFC-134a (SIO-05), HFC-125 (SIO-14), PFC-14 (SIO-05) and PFC-116 (SIO-07). Archived air measurements were reported on the latest scale SIO-05 for HFC-134a and SIO-07 for HFC-23. The archived air measurements for HFC-125 reported by O'Doherty et al. (2009) on the calibration scale UB-98 (UB: University of Bristol) were converted to the

latest scale SIO-14 by the conversion factor (SIO-14/UB-98 = 1.0826) (See Table 3). The archived air measurements for HCFC-22 reported by Miller et al. (1998) on the calibration scale SIO-93 were converted to the latest scale SIO-05 by the conversion factor (NOAA-1992/SIO-93 = 0.997 ± 0.004, NOAA-2006/NOAA-1992 = 1.005, NOAA-2006/SIO-05 = 0.997 ± 0.003) (See Table 3). The firn air measurements for HCFC-141b and HCFC-142b were reported on the calibration scale UB-98. Conversion factors (NOAA-1994/ UB-98 = 1.006 ± 0.003, NOAA-1994/ SIO-05 = 0.994 ± 0.005) are used to trans-

fer data to the latest calibration scale SIO-05 for HCFC-141b (Prinn et al., 2000; Simmonds et al., 2017a). For HCFC-142b, the data can be converted from scale UB-98 to be the latest calibration scale SIO-05 by conversion factors (NOAA-1994/UB-98 = 0.937 ± 0.003, NOAA-1994/SIO-05 = 0.974 ± 0.005) (Prinn et al., 2000; Simmonds et al., 2017a).

All NOAA absolute calibration scales for various trace gases are displayed on their website (https://www.esrl.noaa.gov/gmd/ccl/scales.html). NOAA flask measurements were reported on the latest NOAA calibration

scale for HCFC-22 (NOAA-2006), HCFC-141b (NOAA-1994), HCFC-142b (NOAA-1994), HFC-134a (NOAA-1995) and HFC-125 (NOAA-2008). Archived air measurements for HCFC-141b and HCFC-142b were also reported on the latest scale. All data reported on the NOAA scales are converted here to the AGAGE calibration scale. The conversion factors between AGAGE and NOAA are shown in Table 3 and were derived on the basis of Table 12 from study (Prinn et al., 2000) and Table S4 from study (Simmonds et al., 2017a). The scale conversions between NOAA-1994 and SIO-98 in former study were

based on the comparison of gas concentrations in air samples in 1994-1995 and the scale conversions between NOAA-1994 and UB-98 were based on the measurements against NOAA standard and UB standard in 1997-1998. In latter study, the scale conversions between NOAA and AGAGE were based on the comparison of gas concentrations in air samples in 1998-2006 for HCFC-22, HCFC-141b, HCFC-142b and HFC-134 at CGO, SMO, THD and MSD. For HFC-125, the NOAA/AGAGE ratio was based the comparison in 2007-2015 at CGO, SMO and THD.

The archived air measurements from UEA are obtained on the NOAA-1994 scale for HCFC-141b and HCFC-142b, and on the NOAA-1995 scale for HFC-134a. It is important to note that the original UEA calibration scale for HCFC-141b, HCFC-142b and HFC-134a (Oram et al., 1995; Oram et al., 1996) has been superseded by the NOAA scale. All UEA measurements obtained on the NOAA scale are converted to the AGAGE calibration scale by the conversion factors (Table 3).

### 2.9 Smoothing spline fit and uncertainty

The monthly means for target compounds were obtained by fitting all data from different institutes with the smoothing spline. The smoothing spline method were based on the previous studies (Reinsch, 1967; Craven and Wahba, 1978; Wahba, 1983; Hutchinson and De Hoog, 1985; Wahba, 1990). The method is briefly described below. For more details, see Sect. S1 in Supplement.

For a set of $n$ data points taking values $y_i$ at times $t_i$, the smoothing spline fit $g(t)$ of the function $g(t_i)$ is defined to be the

minimizer of



$$csapsGCV = p \sum_{i=0}^{n} [\frac{g(t_i) - y_i}{\delta y_i}]^2 + \int g''(t)^2 dt \qquad (1)$$

Generally, the function gets an initial guess by sampling 15 various values of smoothing parameter $p$, from $10^{-4}$, $10^{-3}$, …, $10^{10}$. The initial guess is the first local maximum. If it does not exist, the minimum location is used instead. The generalized cross-validation is used to estimate the smoothing parameter $p$. After estimating the optimal smoothing parameter, the estimated variance ($VAR$) and 95 % Bayesian confidence intervals ($CI$) are calculated. The weights ($W$) are assumed to be the
standard deviations ($\delta y_i$) associated with the observed variables. It calculates the spline the way Reinsch (1967) specified.

### 2.10 Seawater solubility estimation method

Solubility has been reported in terms of the Henry's solubility coefficient $H$ (mol L⁻¹ atm⁻¹), the mole fraction solubility $x$ (mol mol⁻¹), the Bunsen solubility coefficient β (L L⁻¹, in STP condition), the Ostwald solubility coefficient $L$ (L L⁻¹), the weight solubility coefficient $c_w$ (mol kg⁻¹ atm⁻¹) or the Künen solubility coefficient $S$ (L g⁻¹). The definitions of solubility are
shown in the studies (Young et al., 1987; Gamsjäger et al., 2008; Gamsjäger et al., 2010). The relationship between different solubility terms are:

$$H = \frac{x}{(1-x) \cdot p^{\ominus} \cdot V_m} = \frac{\beta}{R \cdot T^{\ominus}} = \frac{L}{R \cdot T} = \frac{c_w \cdot M_l}{V_m} = \frac{S \cdot M_g}{R \cdot T^{\ominus} \cdot V_m} \qquad (2)$$

where $T^{\ominus} = 273.15\ K$ and $p^{\ominus} = 101.325\ kPa = 1\ atm$ are the standard temperature and pressure (STP); $V_m$ is molar volume of solvent, $V_m = 18.01528 \cdot 10^{-3}$ L mol⁻¹ for water; $R$ is ideal gas constant, 8.314459848 L kPa K⁻¹ mol⁻¹ (or 0.08205733847 L atm K⁻¹ mol⁻¹); $T$ is temperature in Kelvin and $M_l$ is Molar mass of solvent, 18.01528 g mol⁻¹ for water.
Here are two complete methods to estimate the solubility of compounds in freshwater and seawater.

### 2.10.1 Method I: the CGW model

The following method to estimate the solubility of gases in seawater was reported in Deeds (2008), briefly described here. The Clark-Glew-Weiss (CGW) solubility equation can be used to build the solubility of gases in water and seawater. It is derived from the integrated van 't Hoff equation and the Setchenow salinity dependence (Weiss, 1970, 1974). It is expressed
as a function of temperature and salinity.

$$lnL = a_1 + a_2 \cdot (\frac{100}{T}) + a_3 \cdot ln(\frac{T}{100}) + S[b_1 + b_2 \cdot (\frac{T}{100}) + b_3 \cdot (\frac{T}{100})^2] \qquad (3)$$

where $L$ is the Ostwald solubility coefficient in L L⁻¹ of a gas in seawater, $T$ is the absolute temperature in Kelvin, $S$ is the salinity in ‰ (or g kg⁻¹) and $a_i$ and $b_i$ are constants.

When $S = 0$, it becomes the freshwater solubility equation for a gas:

$$lnL_0 = a_1 + a_2 \cdot (\frac{100}{T}) + a_3 \cdot ln(\frac{T}{100}) \qquad (4)$$

where $L_0$ is the Ostwald solubility in L L⁻¹ of a gas in freshwater.

There are not complete studies on the solubility of our target gases in seawater based on the experiments and/or the CGW model. Fortunately, the solubility of a gas in seawater can be determined from their freshwater solubility, which can be represented by the following form using a modified Setschenow equation (Masterton, 1975):

$$\ln(L_0/L) = k_S \cdot I_V \qquad (5)$$

Here $L_0$ is the freshwater solubility, $L$ is the solubility in a mix-electrolyte solution, such as seawater, $k_s$ is the salting-out coefficient and $I_v$ is the ionic strength of solution. $k_s$ is an empirically-derived, temperature-dependent constant. It can be
estimated as a function of temperature using the freshwater and seawater solubility data by the least-square fit with a second-order polynomial (Masterton, 1975).





$$k_S = c_1 t^2 + c_2 t + c_3 = c_1(T - 273.15)^2 + c_2(T - 273.15) + c_3 \qquad (6)$$

where $t$ is the temperature in Celsius, $T$ is the temperature in Kelvin and $c_i$ is the constant.

The ionic strength of seawater ($I_V$, in g L$^{-1}$) from its salinity (Deeds, 2008) can be calculated by:

$$I_V = \frac{0.03600}{1.80655} \times S \times \rho(T, S) \qquad (7)$$

where $\rho(T, S)$ is the density of seawater in kg L$^{-1}$, estimated using an equation of state for seawater (Millero and Poisson, 1981).

$$\rho = \rho_0 + DS + ES^{3/2} + FS \qquad (8)$$

The seawater solubility of target compounds based on Method I can then be estimated by combining the freshwater solubility Eq. (4) and Eq. (5), (6), (7) and (8):

$$lnL = \left[ a_1 + a_2 \cdot \left( \frac{100}{T} \right) + a_3 \cdot ln \left( \frac{T}{100} \right) \right] \cdot \exp[-(c_1(T - 273.15)^2 + c_2(T - 273.15) + c_3)$$
$$\times \frac{0.03600}{1.80655} \times S \times \rho(T, S)] \qquad (9)$$

### 2.10.2 Method II: the pp-LFER model

The solubility estimation of compounds is based on a cavity model, the polyparameter linear free energy relationships (pp-LFERs) of Abraham (1993). In this model, the process of dissolution of a gaseous solute in a solvent involves setting up of

various exoergic solute-solvent interactions. Each of these interactions are presented in relevant solute parameters or descriptors. The selected Abraham model solute descriptors are the excess molar refraction ($E$) in cm$^3$ mol$^{-1}$/10, the solute dipolarity/ polarizability ($S$), the overall solute hydrogen-bond acidity ($A$) and basicity ($B$), and the McGowan's characteristic molar volume ($V$) in cm$^3$ mol$^{-1}$/100.

$$logSP = c + eE + sS + aA + bB + vV \qquad (10)$$

In this equation, the dependent variable $logSP$ is some properties of a series of solutes in a given phase (or phases). There-

fore, $SP$ could be $L$, the partition coefficient for a series of solutes in a given gas-solvent system or it could be $P$, the water-solvent partition coefficient for a series of solutes. The pp-LFERs have been applied and validated in many types of partition coefficients (Abraham et al., 2004; Abraham et al., 2012).

The Abraham model solute descriptors $E$, $S$, $A$, $B$, $V$ are calculated based on different methods. The $E$ descriptor describes the polarizability of a solute. For HCFC-141b, HCFC-142b, HFC-134a, HFC-125, HFC-23 and PFC-14, the values of $E$ de-

scriptor were given by Abraham et al. (2001). For HCFC-22, the value of $E$ descriptor was calculated by Eq. (11) on the basis of the number of iodine, bromine, chlorine and fluorine atoms ($nI$, $nBr$, $nCl$ and $nF$) in a halocarbon (Abraham et al., 2012). It is obtained based on the regression on 221 compounds. The methods of determining $S$, $A$, $B$ descriptors are reported in previous studies (Abraham et al., 1989; Abraham et al., 1991; Abraham, 1993). The $V$ descriptor, which is the measure of the size of a solute, is the molar volume of a solute calculated from McGowan's approach (McGowan and Mellors, 1986;

Abraham and McGowan, 1987). The $V_c$ descriptor is the corrected McGowan's characteristic molar volume. Note that the characteristic atomic volume for a fluorine atom needs correction from 10.48 cm$^3$ mol$^{-1}$ to be 12.48 cm$^3$ mol$^{-1}$ (Goss et al., 2006).

$$E = 0.641nI + 0.328nBr + 0.140nCl - 0.0984nF \qquad (11)$$
$$n = 221, \text{SD} = 0.083$$

In this work, $logSP$ refers to some solubility-related properties of a series of gaseous solutes in water. $SP$ is the gas-water

partition coefficient $K_W$, expressed as the Ostwald solubility coefficient $L_0$ (in L L$^{-1}$). The solubility of 408 gaseous compounds in water at 298.15 K has been correlated through Eq. (12) (Abraham et al., 1994; Abraham et al., 2001; Abraham et al., 2012). The resulting solubility of 82 gaseous compounds in water at 310.15 K are described in Eq. (13) (Abraham et al., 2001). The values of temperature ($T$) dependent coefficients $c$, $e$, $s$, $a$, $b$ and $v$ are shown in Eq. (12) and (13) at 298.15 K and





310.15 K, respectively. These coefficients are found by the method of multiple linear regression (MLR) analysis on series of water solubility of gaseous compounds. The following equations are the solubility functions of gaseous compounds in water at 298.15 K and 310.15 K based on their physical properties.

$$log L_0 = -0.994 + 0.577E + 2.549S + 3.813A + 4.841B - 0.869V \tag{12}$$
$$n = 408, \rho = 0.9976, \text{SD} = 0.151, T = 298.15 \text{ K}$$

$$log L_0 = -0.966 + 0.698E + 2.412S + 3.393A + 4.577B - 1.072V \tag{13}$$
$$n = 82, \rho = 0.9945, \text{SD} = 0.156, T = 310.15 \text{ K}$$

The solubility of a compound in salt solution can be determined from its solubility in water. This is also the quantitative description of the salting-out effect on neutral organic solutes. Both can be represented by the following form using a modified Setschenow relationship (Sander, 1999; Schwarzenbach et al., 2003; Endo et al., 2012):

$$log(L_0/L) = K_S \cdot [salt] \tag{14}$$

where $L_0$ is the Ostwald solubility coefficient in pure water (in L L$^{-1}$), $L$ is the Ostwald solubility coefficient in the salt solution (in L L$^{-1}$), $K_S$ is the molality-based Setschenow (or salting-out) coefficient (M$^{-1}$) for the salinity- and common logarithm-based Setschenow equation and is independent of [salt], and [salt] is the molality of the salt in mol L$^{-1}$. The relationship between [salt] and salinity ($S$, g L$^{-1}$) in seawater is $[salt] = S/M_{NaCl}$. $M_{NaCl}$ is the molar mass of Sodium chloride (NaCl, 58.44 g mol$^{-1}$). So the salt concentration in seawater is equivalent to 0.6 M NaCl (i.e., [NaCl] = ca. 0.6 M). It is the best to define the salt solution based on molality. Adding dry salt to a solution does not change the molality of other solutes as the molality is the mass of the solvent rather than the solution (Sander, 1999).

The salting-out coefficient $K_S$ should be estimated so as to calculating the solubility of a compound in salt solution. The $K_S$ can also be estimated by the poly-parameter linear free energy relationships (pp-LFERs) since $K_S$ is formally comparable with the Common Logarithm of the partition coefficient between the 1 M NaCl solution and freshwater (Abraham et al., 2012; Endo et al., 2012).

$$K_S = c + eE + sS + aA + bB + vV_c \tag{15}$$

$E$, $S$, $A$ and $B$ are same as the ones described above. $V_c$ is the corrected McGowan's characteristic molar volume in cm$^3$ mol$^{-1}$/100 with the corrected characteristic atomic volume for a fluorine atom (Goss et al., 2006).

The Setschenow coefficient $K_S$ of 43 diverse neutral compounds in NaCl solutions at 298.15 ± 2 K have been correlated through Eq. (16) (Endo et al., 2012):

$$K_S = 0.112(\pm0.021) - 0.020(\pm0.013)E - 0.042(\pm0.020)S - 0.047(\pm0.018)A$$
$$- 0.060(\pm0.022)B + 0.171(\pm0.017)V_c \tag{16}$$
$$n = 43, R^2 = 0.83, \text{SD} = 0.031, T = 298.15 \pm 2 \text{ K}$$

The pp-LFER model resulted in a relatively accurate fit ($R^2 = 0.83$, SD = 0.031). Using the above pp-LFER model, the Setschenow coefficient $K_S$ can be estimated for numerous compounds with various functional groups.

The solubility of compounds in seawater based on the pp-LFER model can be estimated by combining the Eq. (10), (12), (13) and (14):

$$L = 10^{\wedge}[-K_S \cdot S/M_{NaCl} + c + eE + sS + aA + bB + vV] \tag{17}$$

**2.10.3 Combined Method: combined the CGW model and the pp-LFER model**

There are two complete methods above to estimate the solubility of compounds in freshwater and seawater. The main differences are the methods in estimating the water solubility and salting-out coefficients. The CGW model method (Method I), reported in Deeds (2008), is mainly based on the Clark-Glew-Weiss (CGW) solubility model. The water solubility functions of compounds are constructed based on the CGW model and the salting-out coefficients are estimated as a function of temperature using the freshwater and seawater solubility data by the least-square fit with a second-order polynomial. More solubility measurements in water and seawater and chemical properties of compounds are considered in Method I. The second





method (Method II) is on the basis of the poly-parameter linear free energy relationships (pp-LFERs). The water solubility of compounds and salting-out coefficients are both estimated based on the pp-LFERs. Method II thinks more about the physical properties of compounds. Both methods have shortages and advantages. For Method I, there are frequently too few seawater solubility measurements for target compounds in order to build the second-order polynomial between the salting-out coeffi-

cient and temperature. For Method II, the water solubility functions for target compounds are only constructed at 298.15 K and 310.15 K (Abraham et al., 2001; Abraham et al., 2012).

The best way is the combination of Method I and Method II to construct the solubility of compounds in water and seawater. The freshwater solubility functions of compounds can be constructed based on the Clark-Glew-Weiss (CGW) solubility model (Method I) with the advantage of valid in larger temperature range. The seawater solubility functions of compounds

can be constructed on the basis of the pp-LFERs in estimating the salting-out coefficients (Method II) with the advantage of working for more compounds. By combining Eq. (4), (12) and (14), the solubility of compounds in seawater ($L$, Ostwald solubility coefficient in L L$^{-1}$) based on the Combined Method can be estimated by the following equation. This equation is used to estimate the seawater solubility of target compounds.

$$L = 10^{-K_S \cdot S / M_{NaCl}} \cdot exp\left[a_1 + a_2 \cdot (\frac{100}{T}) + a_3 \cdot ln(\frac{T}{100})\right] \qquad (18)$$

### 3 Results and Discussion

### 3.1 Atmospheric histories and growth rates

During late winter, typically January, February and March in the Northern Hemisphere and July, August and September in the Southern Hemisphere, heat is lost from the surface seawater which results in increased density of the surface seawater. The surface water tends to sink toward the ocean interior, and in the process carries concentration signals of atmospheric gases that have been equilibrated with the surface water in preceding months. Since we are interested in reporting the annual

means for the compounds for the use as oceanic tracers of water masses, it is useful to know the atmospheric mole fractions of these compounds in late winter compared to annual means. Mid-February and mid-August are nominally the coldest times in the Northern and Southern Hemispheres, respectively, and normally the main periods when water masses are formed. Based on this we reconstructed mid-February and mid-August annual mean atmospheric mole fractions for all species in the Northern and the Southern Hemispheres. The mid-year annual mean atmospheric mole fractions of these compounds can be

used to compare with the annual mean atmospheric mole fractions for CFC-11, CFC-12, CFC-113, and CCl₄ (Walker et al., 2000; Bullister, 2015).

As described in Sect. 2 there are a number of different data sets available for the compounds of interest, such as in situ atmospheric measurements, flask air measurements, archived air measurements, firn air measurements and model calculations from three institutes (Table 2). Once consolidated onto a single calibration scale, all data were fitted by a smoothing spline to

obtain the monthly means (Fig. 2a-9a) for each compound. The mid-year annual means (Fig. 2b-9b) were calculated by averaging the obtained monthly means of the corresponding 12 months. The mid-February and mid-August means are obtained by averaging the monthly means of January, February and March and the monthly means of July, August and September of the same year. The annual mean growth rates (Fig. 2b-9b) were calculated based on the mid-year annual means.

Figures 2-9 show annual means and growth rates in the northern (NH) and southern (SH) hemisphere lower tropospheric

HCFC-22, HCFC-141b, HCFC-142b, HFC-134a, HFC-125, HFC-23, PFC-14 and PFC-16 concentrations in parts-per-trillion (ppt) and the associated uncertainty. The concentrations are given with mid-February (e.g. 2000.125), mid-year (e.g. 2000.500) and mid-August (e.g. 2000.625) and expressed as the mole fraction of the trace gas in dry air (Table S1).

Global annual mean mole fractions of HCFCs, HFCs and PFCs have increased continuously in the background atmosphere throughout the whole atmosphere history record (Fig. 2-9). The mole fractions for all target compounds in the northern hem-





ispheric are larger than those in the southern hemisphere. In general, recent growth rates are decreasing for HCFCs, increasing for HFCs and are stable for PFCs.

### 3.1.1 HCFC-22

The monthly means atmospheric history of HCFC-22 (Fig. 2a) is determined using the smoothing spline to fit the combined data (Table 2a). The mid-February for NH, mid-year for both hemispheres and mid-August for SH annual means atmospheric history and annual growth rates of HCFC-22 are shown in Fig. 2b. The global annual mean mole fractions of HCFC-22 have increased continuously, initially exponentially followed by a period of more linear rise, throughout the atmospheric history record. The maximum of HCFC-22 annual mole fractions in NH and SH are $251.84 \pm 0.55$ ppt in mid-February 2017 and $229.02 \pm 0.46$ ppt in mid-August, respectively. The atmospheric mole fractions for HCFC-22 in the NH are 14% (median) larger than those in the SH over the entire record. The inter-hemispheric gradients (Fig. 2b) initially increased but have been diminishing since 2011 because of the decline of the growth rates.

The growth rates for HCFC-22 rose steadily until 1990, followed by a relatively stable period with small annual fluctuations. The average annual growth rates are at $5.71 \pm 0.24$ ppt $yr^{-1}$ for NH and $5.12 \pm 0.48$ ppt $yr^{-1}$ for SH between 1990 and 2003. This initial increase and then stable in the growth rate of HCFC-22 coincide with the large production and consumption reported for between 1950s and 1990s (Fig. 1) and a freeze of production magnitudes in the developed countries in 1996. A rapid increase between 2005 and 2008 was found in Fig. 2b. Corresponding step changes were also shown in 2005 in both observations (upper panel in Fig. 2b) and emissions (Xiang et al., 2014), in response to the United Nations Environment Programme (UNEP) production changes (UNEP, 2018). The growth rate peaked in 2008 at $8.45 \pm 0.28$ ppt $yr^{-1}$ for NH and $8.23 \pm 0.31$ ppt $yr^{-1}$ for SH before a period of sharp decline at an annual average rate of 0.59 ppt $yr^{-2}$ for NH and 0.53 ppt $yr^{-2}$ for SH. This suggests that global emissions are not growing as rapidly as before 2008, as also reported by Montzka et al. (2015) and Graziosi et al. (2015). This consistent with the accelerated phase out of HCFCs, that is, 75 % (base year 1989) reduction of HCFC production and consumption by 2010 in non-Article 5 countries against the previous 65 % reduction mandated by the 2007 Adjustment to the MP for Substances that Deplete the Ozone Layer (Graziosi et al., 2015). For HCFC-22, the dispersive application was phased out. This suggests that the 2007 Adjustments played a role. The annual growth rates of HCFC-22 are $3.73 \pm 0.30$ ppt $yr^{-1}$ for NH and $4.03 \pm 0.32$ ppt $yr^{-1}$ for SH in 2016.

### 3.1.2 HCFC-141b

The monthly means atmospheric history of HCFC-141b (Fig. 3a) is obtained by using the smoothing spline to fit the combined data shown in Table 2b. The mid-February, mid-year and mid-August annual mean atmospheric history and annual growth rates of HCFC-141b are shown in Fig. 3b. The annual mean mole fractions of HCFC-141b increased since 1990 and showed an S-shaped growth pattern with a slowdown in the second-half period of 2000s, especially in the NH. This slowdown was also shown in emissions, which attributed to the sharply drop of production and consumption of HCFC-141b in 2005 (Montzka et al., 2015). The HCFC-141b annual mixing has increased to maximum values of $26.16 \pm 0.16$ ppt in mid-February 2017 and $23.49 \pm 0.08$ ppt in mid-August for the end of the time-series for NH and SH. The atmospheric mole fractions for HCFC-141b in the NH are 17 % (median) larger than those in the SH on average. The inter-hemispheric gradients (Fig. 3b) began to increase in early 1990s, reached a peak in around 2000, and then increased to reach a second peak in around 2012 before declining. They exhibit initial exponential growth patterns before 1995 for NH and 1997 for SH. This coincides with intensified industrial production and consumption of HCFCs in the 1990s as production of CFCs was curtailed. Following a comparatively stable plateau period, the average annual growth rates were $1.99 \pm 0.23$ ppt $yr^{-1}$ for NH in the period 1994-2000 and $1.75 \pm 0.08$ ppt $yr^{-1}$ for SH between 1997 and 2000. This consistent with the UNEP production (Montzka et al., 2015) and consumption changes (UNEP, 2018). Subsequently, the growth rate declined until 2005 with the reason mentioned above. But since 2005, the production of HCFC-141b has increased substantially in developing countries.


The growth rates recovered to higher values and stayed at a relative stable level at 0.72 ± 0.23 ppt yr$^{-1}$ for NH and 0.65 ± 0.18 ppt yr$^{-1}$ for SH in 2006-2010. Since 2012 the growth rates appear to be in decline to close to those seen in 1980s. This decline coincides with the global production and consumption of HCFCs being capped in 2013 resulting from the production and consumption in developing countries being capped in 2013 (Montzka et al., 2015).

### 3.1.3 HCFC-142b

The monthly means atmospheric history of HCFC-142b (Fig. 4a) used the data sources from Table 2c. The mid-February, mid-year and mid-August annual means atmospheric history and annual growth rates of HCFC-142b are shown in Fig. 4b. The annual mean mole fractions of HCFC-142b also show a sigmoidal growth pattern, a slow initial increase until 1989 followed by a sharp increase in 1990s, the same slowdown in mid-2000s as HCFC-141b, and finally a plateau in recent years.

The HCFC-142b annual mole fractions reached a maximum of 23.50 ± 0.15 ppt in mid-February 2017 (NH) and 22.00 ± 0.07 ppt in mid-August (SH). The atmospheric mole fractions for HCFC-142b in the NH are 30% (median) larger than those in the SH. The inter-hemispheric gradients (Fig. 4b) exhibit two peaks, one in around 2000 and one in 2007/2008 followed by a substantial drop. The annual mean growth rates of HCFC-142b show an approximately bimodal distribution pattern. The first plateau period happens in 1990s with an average value of 1.22 ± 0.25 ppt yr$^{-1}$ for NH and 1.04 ± 0.07 ppt yr$^{-1}$ for

SH. The growth rates of the other one are 1.29 ± 0.10 ppt yr$^{-1}$ for NH in 2007 and 1.26 ± 0.05 ppt yr$^{-1}$ for SH in 2008. There seems to be a bit of a dip in growth rate around 2005 for all three HCFCs. The peak-valley-peak distribution patterns of HCFCs consistent completely with the UNEP consumption changes (UNEP, 2018). This is followed by a dramatic decline to 2015 at an annual average rate of 0.19 ppt yr$^{-1}$ for both hemispheres. This decline in both atmospheric mole fraction and emissions follows reduced production and consumption in developed countries and a levelling off of production and con-

sumption in developing countries (Carpenter et al., 2014). The current growth rates are similar to those seen in the 1980s before the raid increase in emissions.

### 3.1.4 HFC-134a

The monthly means atmospheric history of HFC-134a (Fig. 5a) is obtained by using the smoothing spline to fit the combined data in Table 2d. The mid-February, mid-year and mid-August annual means atmospheric history and annual growth rates of

HFC-134a are shown in Fig. 5b. The annual mean mole fractions of HFC-134a have increased continuously since around 1990, initially exponentially followed by a more linear growth in the atmosphere. The monotonically increase was also found in emissions for HFC-134a (Xiang et al., 2014; Montzka et al., 2015; Simmonds et al., 2017a). The HFC-134a annual mole fractions reached a maximum of 100.31 ± 0.31 ppt in mid-February 2017 (NH) and 84.59 ± 0.19 ppt in mid-August (SH), by the end of the current time-series. The atmospheric mole fractions for HFC-134a in the NH are 20 % (median) larg-

er than those in the SH. The inter-hemispheric gradients (Fig. 5b) have continued to grow since 1992, most rapidly between 1995 and 2004. The annual mean growth rates of HFC-134a were shown in the bottom panel in Fig. 5b. They start to increase in around 1992, rapid increase in the same period as the inter-hemispheric gradients, followed by the slowly increase. The maximum growth rates are shown at the end of the time-series, that is, 6.17 ± 0.16 ppt yr$^{-1}$ for NH and 5.69 ± 0.13 ppt yr$^{-1}$ for SH.

### 3.1.5 HFC-125

The monthly means atmospheric history of HFC-125 (Fig. 6a) is obtained by using the smoothing spline to fit the combined data in Table 2e. The mid-February, mid-year and mid-August annual means atmospheric mole fractions and annual growth rates of HFC-125 are shown in Fig. 6b. The annual mean mole fractions and growth rates of HFC-125 both increased exponentially throughout the atmospheric history record. This could be attributed to the continuing increase in emissions for

HFC-125 (O'Doherty et al., 2009; Montzka et al., 2015; Simmonds et al., 2017a). The HFC-125 annual mole fractions



reached a maximum of 24.50 ± 0.22 ppt (NH) and 19.37 ± 0.06 ppt (SH) at the end of the time-series. The growth rate for HFC-125 also reached a peak of 2.63 ± 0.11 ppt yr$^{-1}$ for NH and 2.41 ± 0.04 ppt yr$^{-1}$ for SH by the end of the time-series. The atmospheric mole fraction for HFC-125 in the NH is 47 % (median) larger than those in the SH. The inter-hemispheric gradients (Fig. 6b) generally show an increasing trend.

### 3.1.6 HFC-23

The monthly means atmospheric history of HFC-23 (Fig. 7a) is determined by combining the data in Table 2f. The mid-February, mid-year and mid-August annual means atmospheric history and annual growth rates of HFC-23 are shown in Fig. 7b. The annual mean mole fractions of HFC-23 have increased with a quadratic growth pattern since 1978. The HFC-23 atmospheric mole fractions reached a peak of 30.20 ± 0.04 ppt for NH and 28.45 ± 0.03 ppt for SH in mid-August by the end of the time-series. The atmospheric mole fractions for HFC-23 in the NH are 7 % (median) large than those in the SH. The inter-hemispheric gradients (Fig. 7b) exhibit an increasing trend with large fluctuations over the time-series. The annual mean growth rates for HFC-23 show a gradually growth pattern with large fluctuations, that is, a higher rate was shown in 2006, fell back in 2009 and subsequently increased to around 2014. These consistent with the change in their emissions (Carpenter et al., 2014; Simmonds et al., 2017b). The slowing in HFC-23 growth rate was in response to emission reductions in developed countries that began in the late 1990's, combined with the HFC-23 destruction program of the CDM of the UN-FCCC for developing countries that started around 2007 (Miller and Kuijpers, 2011; Carpenter et al., 2014). The higher values in growth rates could be attributed to the increase in production of HCFC-22 with no subsequent incineration of HFC-23 (Miller and Kuijpers, 2011; Carpenter et al., 2014). The annual growth rates at the end of the time-series for HFC-23 are 0.78 ± 0.02 ppt yr$^{-1}$ for NH and 0.93 ± 0.02 ppt yr$^{-1}$ for SH.

### 3.1.7 PFC-14

The monthly means atmospheric history of PFC-14 (Fig. 8a) is estimated using the smoothing spline to fit the data in Table 2g. Trudinger et al. (2016) used a model to determine the atmospheric abundance of PFC-14 since 1900 inferred from ice core, firn, air archive and in situ measurements. Here we included more data, updated and extended the time series as presented by Trudinger et al. (2016). The mid-February, mid-year and mid-August annual means atmospheric history and annual growth rates of PFC-14 are shown in Fig. 8b. PFC-14 has a natural background atmospheric concentration of 34.05 ± 0.33 ppt (Trudinger et al., 2016). The annual mean mole fractions of PFC-14 begun to increase around 1900, went up a step in 1943/1944. The step change was also found in emissions, which is most likely attributed to increasing aluminium production during World War II (Barber and Tabereaux, 2014; Trudinger et al., 2016), for example for construction of aircraft. Afterwards, the mole fractions of PFC-14 began to increase rapidly in the 1970s. Since then it has continued to grow, reaching maximum 83.92 ± 0.04 ppt for NH in mid-February 2017 and 82.27 ± 0.02 ppt for SH at the end of the time-series. The atmospheric mole fractions for PFC-14 in the NH are 1 % (median and average) larger than those in the SH. The inter-hemispheric gradients (Fig. 8b) exhibited a peak in 1980 and a valley in 2009, and have been increasing recently. The atmospheric growth rates have increased since 1905 and reached a higher value in 1940s because of the increased aluminium production. Then the rates began to increase from 1950s and peak in 1980 before declining. The relative lower rates in around 1950s could be attributed to reduce electricity consumption during aluminium production. Whereas the decline latter were the concerted effort by the aluminium and semiconductor industries (Trudinger et al., 2016). The dip in 2009 could be related to the impact of Global Financial Crisis on global aluminium and semiconductor production (Trudinger et al., 2016). The growth rates of PFC-14 have been increasing again during the last five years maybe because of the increased primary aluminium production year after year (Trudinger et al., 2016) and the global economic recovery. The growth rates for PFC-14 have initial increased slowly, experienced a small peak in 1943/1944, stayed a relatively stable period, followed by a big peak in 1980/1981 and subsequently decreased to a stable value with small fluctuations. The small peak values are 0.41 ±


0.05 ppt yr$^{-1}$ and 0.34 ± 0.04 ppt yr$^{-1}$ for each hemisphere whereas the large peak values are 1.22 ± 0.05 ppt yr$^{-1}$ for NH and 1.18 ± 0.04 ppt yr$^{-1}$ for SH. The stable average growth rates in 2010-2016 are 0.71 ± 0.06 ppt yr$^{-1}$ for NH and 0.70 ± 0.05 ppt yr$^{-1}$ for SH.

### 3.1.8 PFC-116

The monthly means atmospheric history of PFC-116 (Fig. 9a) used the data source in Table 2h. Here we updated and extended the time series of PFC-116 based on the study (Trudinger et al., 2016). The mid-February, mid-year and mid-August annual means atmospheric history and annual growth rates of PFC-116 are shown in Fig. 9b. PFC-116 has a pre-industrial background atmospheric level of 0.002 ppt (Trudinger et al., 2016). PFC-116 shows a similar atmospheric trend to PFC-14. The annual mean mole fractions have increased since around 1900, going up a step in 1943/1944, and increased significantly

in the 1970s, reaching 4.682 ± 0.007 ppt for NH by mid-February 2017 and 4.516 ± 0.004 ppt for SH by mid-August. The atmospheric mole fractions for PFC-116 in the NH are 6 % (median) larger than those in the SH. The percentage is higher could be that the emissions of PFC-116 is larger than those of PFC-14 from aluminium production and more emissions are from NH than SH. The inter-hemispheric gradients (Fig. 9b) reached a peak in 1999, followed by a slow decline in recent years. The growth rates for PFC-116 begun to increase slowly, experience a higher value in 1943/1944, stayed comparatively

stable for a period of time, climbed gradually to a large peak in the end of 1990s and subsequently declined to a stable value with small fluctuations. The higher value of PFC-116 in 1940s has the same reasons as PFC-14. The large peak values of PFC-116 are 0.038 ± 0.007 ppt yr$^{-1}$ for NH and 0.033 ± 0.008 ppt yr$^{-1}$ for SH. The growth rates by the end of the time-series are 0.078 ± 0.003 ppt yr$^{-1}$ for both hemispheres.

### 3.2 Growth Patterns

The atmospheric history trends of target compounds generally follow expected patterns based on the history of their known industrial applications and production bans. We can make out three distinct behavioural patterns. Pattern I: the annual mean mole fractions show Sigmoidal (S-shaped) growth and the annual growth rates exhibit a normal distribution over the whole time period, for example HCFC-141b and HCFC-142b. The largest inter-hemispheric gradients approximately coincide with the maximum annual growth rates. In recent times the inter-hemispheric gradients of HCFC-141b and HCFC-142b are di-

minishing, as a result of reducing emissions. Pattern II: the annual mean mole fractions show initial exponential growth followed by a period of linear increase, whilst the growth rates show a Sigmoidal pattern, for example HFC-134a and HFC-23 (combined the modelled mole fractions output of HFC-23 from 1950 to 2016 in Fig. 1 from Simmonds et al. (2017b)). The atmospheric concentration histories of these compounds in the future will likely experience a plateau phase, followed by a decline following the consumption restrictions imposed by the 2016 Kigali Amendment to the Montreal Protocol). Pattern III:

The annual mean mole fractions and growth rates both show exponential (J-shaped) growth, for example HFC-125. The atmospheric history and growth rate of HFC-125 will likely follow a similar path to the compounds in Pattern II as they are subjected to the same regulations. Common for both Pattern II and Pattern III compounds is that their inter-hemispheric gradients and growth rates are both increasing with the overall trends. An interesting thing is that the trends of inter-hemispheric gradients are similar to the trends of growth rates in both hemispheres for each compound. This could attributed to the

growth rates in both hemispheres are really similar.

The annual mean mole fractions of the remaining halocarbons, HCFC-22, PFC-14 and PFC-116, have increased throughout the time-series and continue to increase today. The growth rates of these compounds initially increased and experienced a peak before declining. The annual mean mole fractions of PFC-14 and PFC-116 show small peaks during World War II, a period of intense aluminium production. The growth trend for HCFC-22 is more likely to decrease and follow the trends of

HCFC-141b and HCFC-142b as they are subjected to the same regulations. Different from all other target compounds, the annual mean growth rates of PFC-14 and PFC-116 stayed stable after a short decline without a ban. This could attribute to





the changing sources of both PFC-14 and PFC-116. Emissions from aluminium industry dominated for a long time but have been declining for the past decade or so. These are offset by increasing emissions from the electronics industry. The long lifetimes of PFCs in the atmosphere makes a decrease in the atmospheric concentration unlikely.

Considering the combined growth patterns and the production and consumption histories for these gases (Fig. 1), the se-
quence of atmospheric change of HCFCs and HFCs coincide with the replacement sequence of CFCs. In 1980s, CFCs were found to be a threat to the ozone layer (Molina and Rowland, 1974; Rowland and Molina, 1975). To facilitate the phase out of the more potent CFCs as ozone depleting substances (ODPs), HCFC production and consumption increased rapidly in developed countries in the 1990s and in developing countries in the mid-2000s as industrial usage as CFCs was curtailed. This consistent with the atmospheric growth rates of HCFCs reached a peak in the 1990s and/or 2000s. Following the 2007
amendment to the Montreal Protocol, the production and consumption of HCFCs was accelerated phase out. With a large bank of HCFC-22 exist in refrigeration systems, emissions are expected to continue (Carpenter et al., 2014). The atmospheric mole fractions of HCFCs tend toward stable values or decline as a consequence of the freeze of HCFC production and consumption for dispersive uses in 2013 in Article 5 countries, Moreover, the growth rates of HCFCs are decreasing. HFCs have been developed as potential substitutes for both CFCs and HCFCs because they pose no harm to the ozone layer. Their
production and consumption has increased rapidly over the past decade or so. This accounts for the exponential growth of the atmospheric mole fractions of many HFCs and the J-shaped or S-shaped patterns of their growth rates.

From Fig. 2-9, it is clear that the mole fractions for all target compounds in the NH are larger than those in the SH but follow similar trends as the majority of emissions (typically >95%) of anthropogenic halocarbons occur at mid-latitudes of the NH (O'Doherty et al., 2009; Saikawa et al., 2012; Carpenter et al., 2014; UNEP, 2018). Previous studies also reported that all
target compounds are released from the NH rather than from the SH (Ashford et al., 2004; Montzka et al., 2009). All of these show that a significant portion of the compounds that are released in the NH are transported to the SH. As for HCFCs, global and regional studies (Montzka et al., 2009; Saikawa et al., 2012; Carpenter et al., 2014) suggest a shift of HCFC emissions from mid-latitudes to lower latitudes, corresponding to recent shifts of production and consumption from developed countries to developing countries (mostly distributed at lower latitudes).

Although the growth rates in both hemispheres are really similar, there are lags in the atmospheric concentrations and growth rates between NH and SH for most target compounds. This may because all CFC and most HCFC and HFC emissions will eventually be mixed into the SH (interhemispheric mixing times are typically around one year). If all emissions stop, a long-lived compound would expect to reach an identical concentration in both hemispheres. For compounds with shorter lifetimes this even out might take longer as the shorter the lifetime, the bigger the interhemispheric gradient.

**3.3 Solubility in seawater**

The seawater solubility functions for HCFC-22, HCFC-141b, HCFC-142b, HFC-134a, HFC-125, HFC-23, PFC-14 and PFC-116 are estimated based on their freshwater solubility as no direct studies of the solubility functions of the target compounds in seawater have been published.

Available freshwater solubility data of all species from previous studies were collected and compiled. The freshwater solu-
bility data were collected for HCFC-22 (Boggs and Buck Jr, 1958; Hine and Mookerjee, 1975; Wilhelm et al., 1977; Mclinden, 1990; Maaßen, 1995; Reichl, 1995; Zheng et al., 1997; Abraham et al., 2001; Battino et al., 2011; Sander et al., 2011), HCFC-141b (Maaßen, 1995; Abraham et al., 2001; Kutsuna, 2013), HCFC-142b (Mclinden, 1990; Maaßen, 1995; Reichl, 1995; Abraham et al., 2001), HFC-134a (Mclinden, 1990; Maaßen, 1995; Reichl, 1995; Zheng et al., 1997; Abraham et al., 2001), HFC-125 (Alexandre et al., 2000; Battino et al., 2011; Mclinden, 1990; Reichl, 1995; Abraham et al., 2001;
HSDB, 2015), HFC-23 (Parmelee, 1953; Hine and Mookerjee, 1975; Wilhelm et al., 1977; Zheng et al., 1997; Abraham et al., 2001; Battino et al., 2011; Sander et al., 2011), PFC-14 (Ashton et al., 1968; Hine and Mookerjee, 1975; Wilhelm et al., 1977; Wen and Muccitelli, 1979; Cosgrove and Walkley, 1981; Smith et al., 1981; Park et al., 1982; Scharlin and Battino,



1992; Abraham et al., 2001; Battino et al., 2011; Sander et al., 2011) and PFC-116 (Wen and Muccitelli, 1979; Park et al., 1982; Bonifácio et al., 2001; Battino et al., 2011). These data for all species were converted to a common solubility unit (Ostwald solubility, $L_0$, in L L$^{-1}$) and fitted with the Clark-Glew-Weiss (CGW) function of temperature to build the freshwater solubility equations (Fig. S1-S8). Only the observed water solubility values from Abraham et al. (2001) are involved in

the fits. Those calculated using the Revised Method II (described below) are involved only for comparison. The water solubility functions in Ostwald solubility unit for HCFC-22, HFC-134a, HFC-125, HFC-23 and PFC-116 (Fig. S1, S4-S6, S8) were compared with the results from Deeds (2008). For HFC-125, there are three fit curves shown in Fig. S5. Considering match with the results (Table S2, Fig. S1-S8) calculated by the Revised Method II (only based on physical properties of compounds), the curve in the bottom is chose as the water solubility fit. In Deeds (2008), the freshwater solubility functions

in the Bunsen solubility unit ($\alpha$, in L L$^{-1}$, in standard temperature and pressure (STP)) were converted to the Ostwald solubility unit for comparison. The freshwater solubility function in Ostwald solubility unit for PFC-14 (Fig. S7) was compared with the results from Clever (2005) and Deeds (2008). The fit for water solubility functions agrees to within 4.0 %, 7.8 %, 2.5 %, 6.8 %, 5.9 % 2.3 %, 0.95 % and 3.5 % with majority (two-thirds) of the data for HCFC-22, HCFC-141b, HCFC-142b, HFC-134a, HFC-125, HFC-23, PFC-14 and PFC-116, respectively. The constants $a_1$, $a_2$, $a_3$ for solubility functions of target

compounds in water are given in Table 4.

In order to validate the calculation method of solubility, the solubility for CFC-12 in water calculated by the Combined Method and by the method from Warner and Weiss (1985) were compared. Warner and Weiss (1985) estimated the freshwater and seawater solubility function of CFC-12 by experiment and a different model fit without using a salting-out coefficient. The freshwater solubility function of CFC-12 calculated by the Combined Method (Fig. S9) was constructed by collecting

freshwater solubility data from various literature (Parmelee, 1953; Hine and Mookerjee, 1975; Wilhelm et al., 1977; Park et al., 1982; Warner and Weiss, 1985; Scharlin and Battino, 1994; Reichl, 1995; Abraham et al., 2001; Sander et al., 2011). The freshwater solubility of CFC-12 from Warner and Weiss (1985) match data from other studies. Moreover, the fits based on the function in Warner and Weiss (1985) and the CGW fit in this study match very well (Fig. S9). The average Relative Standard Deviation (RSD) of water solubility estimated by the two methods for CFC-12 in the range of 298.15-338.15 K is

0.17 %. This means that our method for estimating the freshwater solubility is valid.

Based on the experimental results of the freshwater and seawater solubility of CFC-12 from Warner and Weiss (1985), the salting-out coefficient $K_S$ (calculated by the Combined Method) is independent of salinity and is a function of only temperature (Fig. S10), which can also be known by description of the Method I and the Method II in Sect. 2.10. In the study of Warner and Weiss (1985), the average of $K_S$ is $0.229 \pm 1.41 \cdot 10^{-15}$ L g$^{-1}$ at 298.15 K when the salinity is in the range of 0-40.

The RSD is $6.16 \cdot 10^{-13}$ %, which is minor enough to be neglected. So the $K_S$ is independent of salinity. In Fig. S10, a quadratic relationship between the salting-out coefficient and temperature was found. The $K_S$ is in the range of 0.229-0.249 L g$^{-1}$ (at a mean of $0.235 \pm 0.00547$ L g$^{-1}$) at salinity 35 when the temperature is in the range of 298.15-338.15 K (0-40 °C). The RSD is 2.33 %. This means that the effect of temperature on the salting-out coefficient is also very small.

In order to estimate the solubility functions for target compounds in seawater, their salting-out coefficients ($K_S$) should also

be estimated. As shown in Eq. (15), $K_S$ can be estimated based on the descriptors of all target compounds. With the exception of PFC-116, the $E$, $S$, $A$, $B$, $V$ values for compounds were obtained from studies (Abraham et al., 2001; Abraham et al., 2012). For PFC-116, the excess molar refraction ($E$) was calculated by Eq. (11). The dipolarity/ polarizability ($S$) for $C_2F_6$ was estimated as -0.350 based on the $S$ of $CF_4$ (-0.250) and $C_3F_8$ (-0.450) (Abraham et al., 2001). $A$ and $B$ for $C_2F_6$ are both zero since it includes only C-halogen atom bonds (without carbon-hydrogen bonds). The corrected $V$ ($V_c$) were obtained from

(Abraham and McGowan, 1987; Goss et al., 2006). The values of all descriptors for the target compounds are shown in Table 5. The errors in calculating the descriptors were estimated as 0.088, 0.047, 0.128, 0.081, 0.095, 0.051, 0.071 and 0.088 for HCFC-22, HCFC-141b, HCFC-142b, HFC-134a, HFC-125, HFC-23, PFC-14 and PFC-116, respectively (Abraham et al., 2001). On the basis of the $E$, $S$, $A$, $B$, $V$ descriptors, $K_S$ was estimated at $298.15 \pm 2$ K and also shown in Table 4. As we

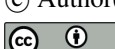



know from the description of Method II, the $K_S$ changes with temperature by changing the coefficients $c$, $e$, $s$, $a$, $b$, $v$ in Eq. (15). As we described in Method I and discussed in Method II for CFC-12, the salting-out coefficient is a second-order polynomial of temperature. From the discussion of CFC-12, the effect of temperature on the salting-out coefficient is very small. Moreover, very limited studies on the $K_S$ have been reported, so here we assume that the salting-out coefficient $K_S$ is a constant, and doesn't change with temperature. Thus the values of salting-out coefficient can be calculated using Eq. (16) for target compounds and the results are shown in Table 4.

Based on the calculation method shown in Sect. 2.10, the seawater solubility functions for HCFC-22, HCFC-141b, HCFC-142b, HFC-134a, HFC-125, HFC-23, PFC-14 and PFC-116 were constructed in the corresponding temperature range (Table 4). The Ostwald solubility in seawater at 1 atm, 25°C and 35 g kg$^{-1}$ were estimated to be 0.669 L L$^{-1}$ for HCFC-22, 0.537 L L$^{-1}$ for HCFC-141b, 0.268 L L$^{-1}$ for HCFC-142b, 0.292 L L$^{-1}$ for HFC-134a, 0.063 L L$^{-1}$ for HFC-125, 0.249 L L$^{-1}$ for HFC-23, 0.00388 L L$^{-1}$ for PFC-14 and 0.00102 L L$^{-1}$ for PFC-116, respectively (Table 4). For Comparison, the solubility of CFC-12, CFC-11, CFC-113, CCl$_4$ and SF$_6$ in seawater are converted to the Ostwald solubility unit at 1 atm, 25°C and 35 g kg$^{-1}$. They are 0.0504 L L$^{-1}$, 0.177 L L$^{-1}$, 0.0518 L L$^{-1}$, 0.568 L L$^{-1}$ and 0.00401 L L$^{-1}$, respectively. The percentage of the seawater solubility on the freshwater solubility is interesting to explore. In previous studies, Mackay et al. (2006) reported that many hydrocarbons have solubility in seawater about 75 % of their solubility in distilled water. Moore et al. (1995) reported that the solubility of short-lived halocarbons (e.g. CH$_3$I, CHBr$_3$, CH$_2$Br$_2$, CHBr$_2$Cl, and CHCl$_3$) in seawater is 80 % of their solubility in freshwater. The percentages for CFCs and SF$_6$ are also presented here for comparison. The solubility of CFC-12 in seawater is 72.9 % of its solubility in freshwater (Warner and Weiss, 1985). The percentages are 71.7 % for CFC-11 (Warner and Weiss, 1985), 73.5 % for CFC-113 (Bu and Warner, 1995), 78.4 % for CCl$_4$ (Bullister and Wisegarver, 1998) and 72.7 % for SF$_6$ (Bullister et al., 2002). For our target compounds, the percentages of the seawater solubility on the freshwater solubility at 298.15 K and salinity of 35 are 79.2 % for HCFC-22, 75.5 % for HCFC-141b, 76.1 % for HCFC-142b, 76.6 % for HFC-134a, 73.3 % for HFC-125, 79.3 % for HFC-23, 75.7 % for PFC-14 and 71.4 % for PFC-116. Similar to the CFCs and SF$_6$, the percentages for HCFCs, HFCs and PFCs are also in the range of around 70-80 %. The Ostwald solubility coefficients of target compounds in seawater in the available temperature range at the permanent salinity 35 and in the salinity range of 0-40 at permanent temperature 298.15 K for target compounds are shown in Fig. 10 and Fig. 11, respectively. The estimated seawater solubility values for all target compounds fall monotonously with the increase of temperature and salinity. The latter is in the form of linear decline.

The overall uncertainties of water solubility estimation for target compounds are calculated as the root-mean-square (RMS) level of the misfit between the measurements and fitted values. They are 0.0352 L L$^{-1}$ for HCFC-22, 0.0283 L L$^{-1}$ for HCFC-141b, 0.0065 L L$^{-1}$ for HCFC-142b, 0.0141 L L$^{-1}$ for HFC-134a, 0.0016 L L$^{-1}$ for HFC-125, 0.0132 L L$^{-1}$ for HFC-23, 9.0695e-05 L L$^{-1}$ for PFC-14, 9.1858e-05 L L$^{-1}$ for PFC-116. The uncertainties of seawater solubility at different salinity are estimated by the propagation of uncertainty from water solubility and salting-out coefficients. The overall uncertainties of seawater solubility are estimated as the RMS of errors in different salinity. They are 0.039 L L$^{-1}$ for HCFC-22, 0.036 L L$^{-1}$ for HCFC-141b, 0.032 L L$^{-1}$ for HCFC-142b, 0.032 L L$^{-1}$ for HFC-134a, 0.030 L L$^{-1}$ for HFC-125, 0.032 L L$^{-1}$ for HFC-23, 0.030 L L$^{-1}$ for PFC-14, 0.029 L L$^{-1}$ for PFC-116. Further experimental studies are required to build the seawater solubility functions for all species.

We also used Method I to build the seawater solubility function for PFC-14 (Table S3) as only the seawater solubility was measured for PFC-14. The seawater solubility data of PFC-14 estimated using Method I are the same as those in Scharlin and Battino (1995). Whereas the constructed seawater solubility function can be used over a greater temperature range. Comparison of the Ostwald solubility coefficients ($L$, L L$^{-1}$) of PFC-14 in seawater estimated by the Combined Method with the measured results from Scharlin and Battino (1995) is shown in Table 6. The estimated solubility of PFC-14 in seawater at 293.15 K is the closest to the measured values. The RSD of calculated value and measured value is only 0.79 %.





The difference (RSD) between the Method II and the Combined Method is the difference of the solubility of compounds in water as the methods for estimating the salting-out coefficient and seawater solubility are the same. For water solubility calculations, Method II uses the pp-LFERs, based only on the physical properties of compounds, whereas the Combined Method uses the CGW model based on the measurements. For Method II, the water solubility of compounds calculated by Eq. (12)

and (13) are based on the McGowan's characteristic molar volume ($V$), not the corrected $V$. Inspired by the study of Endo et al. (2012) and Goss et al. (2006), the corrected $V$ is also used to calculate the water solubility. From Table S2, with the exception of HFC-125 at both temperatures and HCFC-141b, HFC-134a and HFC-23 at 310.15 K, the water solubility values of target compounds and CFC-12 calculated based on corrected $V$ are closer to both the observed values (Abraham et al., 2001) and the CGW fitted values than when they are calculated based on $V$. Based on the above discussion, the $V$ in Eq. (12)

and (13) could change to corrected $V$ for all target compounds except HFC-125. This would be the Revised Method II ($V$ for HFC-125 and corrected $V$ for other target compounds). The water solubility functions of our target compounds and CFC-12 are shown in Table S4 and the calculated water solubility values based on the Revised Method II are also shown in Fig. S1-S9. Small difference of water solubility calculated by the Revise Method II and the Combined Method verifies the reliability of both methods.

For comparison, with the exception of the seawater solubility function of CFC-12 constructed by Warner and Weiss (1985), the functions were also constructed by the Revised Method II and the Combined Method (Table 4). The Ostwald solubility coefficients calculated on the basis of the above three methods in the available temperature range at the permanent salinity 35 and the coefficients in the salinity range of 0-40 at permanent temperature 298.15 K for CFC-12 are shown in Fig. 12. The RSDs of the seawater solubility values for CFC-12 estimated by the Revised Method II and Warner and Weiss (1985)

are 3.41 % at 298.15 K and 13.47 % at 310.15 K, based only on the propagation of uncertainty of salting-out coefficients. The average RSD of the seawater solubility values estimated by Warner and Weiss (1985) and the Combined Method is 1.40 ± 0.85 % in the coincidental temperature range at the permanent salinity 35 ‰. The average RSD of the seawater solubility values estimated by Warner and Weiss (1985) and the Combined Method is 2.94 ± 0.44 % in the same salinity range of 0-40 at permanent temperature 298.15 K. These results show that the seawater solubility values estimated by Combined Method

are very close to the ones measured in Warner and Weiss (1985). Without measurements on solubility of compounds in seawater, the Combined Method is a good way to estimate the seawater solubility.

Based on the discussion above, some conclusions on the methods to estimate the seawater solubility of halogenated compounds can be made as follows. It is worth noting that these methods can potentially be applied to many more compounds.

1. The Revised Method II is a good way to estimate the seawater solubility of compounds particularly in cases where neither

water solubility data nor seawater solubility data have been reported from experiments. This method is only based on the physical properties of compounds. The water solubility values and the salting-out coefficients are both estimated using the pp-LFERs.

2. The Combined Method is a better way to estimate the seawater solubility of compounds in cases where there are water solubility data points but no seawater solubility data points reported from experiments. This is the case for the current study.

The water solubility function is constructed based on the CGW fit and the salting-out coefficient is estimated using the pp-LFERs.

3. The Method in the studies of (Warner and Weiss, 1985; Bu and Warner, 1995; Bullister and Wisegarver, 1998; Bullister et al., 2002) is the best way to estimate the seawater solubility. Here, both the water solubility values and seawater solubility values are obtained from experiments.

## 3.4 Transient Tracer potential and comparison with CFC-12

The production and consumption history of CFC-12 is shown in Fig. 1. CFC-12 was first synthesized by the Belgian scientist Frédéric Swartz during 1890s (Jacobson, 2012). In 1931, CFC-12 was produced by the DuPont manufacturer under the trade



name Freon. In 1974, Molina and Rowland (1974) proposed that CFC-12 could destroy the stratospheric ozone. In 1984, the "ozone hole" was discovered by the British Antarctic Survey (Farman et al., 1985). The prime suspect is the Chlorine inject-ed into the stratosphere by the CFCs. In 1987, governments negotiated the Montreal Protocol. Following revisions mandated the freezing of CFC-12 in 1989 for developed countries and in 1999 for developing countries. Following revisions also man-

dated the complete phase-out of CFC-12. For developed countries, CFC-12 production and consumption ban was advanced to 1 January 1996. For developing countries, its ban was mandated in 2010 (UNEP, 2006).

The history of CFC-12 used as an oceanic transient tracer is also presented here. In 1970, Lovelock (1971) first detected CFC-12 using GC-ECD. In 1973, Lovelock et al. (1973) first proposed that CFC-12 can be used as a transient tracer to study water masses in the ocean. Afterwards, large numbers of studies using CFC-12 as an oceanic transient tracer were published

(Gammon et al., 1982; Weiss et al., 1985; Smethie et al., 1988; Körtzinger et al., 1999; Tanhua et al., 2008; Smith et al., 2016; Fine et al., 2017). In the 1990s, the World Ocean Circulation Experiments (WOCE) used CFC-12 as the normal tracer to investigate global ocean circulation and mixing. CFC-12 is still extensively used as a tracer, although its production was prohibited in 1996 and its atmospheric concentration subsequently peaked in the early 2000's and is now in slow decline.

As one of the requirements to be a useful oceanic tracer, well-established and transient source functions (i.e. atmospheric

abundance histories), have been established for HCFC-22, HCFC-141b, HCFC-142b, HFC-134a, HFC-125, HFC-23, PFC-14 and PFC-116. With the exception of HCFC-141b (Fig. 3a), the other compounds in this series are all steadily increasing in the atmosphere. Therefore they all have the potential to be used as oceanic transient tracers if only considering their source functions.

Although HCFCs, except HCFC-141b, are continuing to increase in the global atmosphere, declined growth rates since 2007

are found. Combined with the ban on HCFCs in 2007, with freezing in 1996/2013 and phase out in 2020/30 (Fig. 1), HCFCs can likely be used as oceanic transient tracers for the next several decades for recently ventilated waters. Due to a fall in emissions of HCFC-141b and its relatively short atmospheric lifetime compared to HCFC-22 and HCFC-142b, the use of HCFC-141b will be more limited than HCFC-22 and HCFC-142b considering that its atmospheric concentrations has al-ready began to decrease. Therefore, atmospheric lifetimes of compounds are quite important particularly once emissions

have fallen. When the atmospheric history of the compound decrease; it's not monotonically changing, the equilibriums' atmospheric concentration calculated from its concentration in the ocean is not unique. Consequently there will be two pos-sible apparent ages for water masses so that this compound will have more limited use as an oceanic tracer unless emissions increase again.

The concentrations of HFCs are continuously increasing in the atmosphere as do their growth rates. Combined with the re-

strictions of HFC consumption in the 2016 Kigali Amendment (with freezing of consumption in 2019 and phase out in 2024/28 (Fig. 1), HFCs can be also used as oceanic transient tracers for young waters for the next several decades. Moreover, HFCs have a higher potential to be oceanic transient tracers than HCFCs considering the increasing growth rates in the background atmosphere.

PFCs are increasing in the atmosphere over a well-known natural background concentration. Combined with an atmospheric

lifetime of over 50 000 years for PFC-14 and 10 000 years for PFC-116, PFCs have great potential to be oceanic transient tracers. PFC-14 has the potential to be the tracers for a longer period thanks to its longer lifetime, steady atmospheric growth rate and no current ban, as discussed by Deeds et al. (2008). But PFC-14 is very difficult to measure in seawater samples because it is extremely volatile, so difficult to trap and separate chromatographically. PFC-116 can be also be used as a tran-sient tracer similar to PFC-14. The challenge for PFC-116 to be tracer is challenging analytical conditions considering its

low concentrations in the atmosphere and a low solubility in the seawater.

Well established source functions and the solubility functions in seawater are only two of the requirements for an oceanic tracer. To be an oceanic transient tracer, the compound should also be conservative in the marine environment and can be capable of rapid, relatively inexpensive and accurate measurement. This work provides the two requirements for potential





new ocean transient tracers, additional studies on conservation in seawater and measured method of target compounds are needed to qualify these compounds to be tracers.

## 4 Conclusions

This work has established the source functions for HCFC-22, HCFC-141b, HCFC-142b, HFC-134a, HFC-125, HFC-23, PFC-14 and PFC-116 based on a synthesis of available data and models in a way that is optimized for oceanic transient tracer work in two ways; 1) the atmospheric concentrations are calculated for the time of water masses formation (late winter) and 2) the seawater solubility of these compounds are reviewed for the first time. In general, the concentrations of these compounds are continuously rising over the past three decades and still increasing today. For HCFC-141b and HCFC-142b the annual mean mole fractions show sigmoidal growth and the growth rates have the shape of a normal distribution. For HFC-134a and HFC-23, the annual mean mole fractions show initial exponential growth followed by linear increase and the growth rates show a sigmoidal pattern. For HFC-125 the annual mean mole fractions and growth rates both show an exponential increase. The source functions and natural background concentrations for all compounds show that HCFC-22, HCFC-141b, HCFC-142b, HFC-134a, HFC-125 and HFC-23 have the potential to be oceanic transient tracers for the next few decades, though their growth rates are expected to reverse, particularly for the HCFCs, due to the restriction on production and consumption imposed by the Montreal Protocol. HFCs have a higher potential to be oceanic transient tracers than HCFCs due to the increasing growth rates in the atmosphere, though these are likely to fall as a result of the recent Kigali Amendment. PFC-14 and PFC-116 have the potential to be the tracers for a longer period due to their longer lifetimes, more consistent atmospheric growth rates and because no production ban is currently in place, though they were listed in the Kyoto Protocol and industrial practices are changing to try to reduce/minimise emissions. In addition, we have used three different methods to estimate the seawater solubility of the compounds of interest based on available theoretical concepts and experimental data. The seawater solubility functions of these compounds were subsequently constructed, completing the input function of these potentially useful oceanic transient tracers.

### Acknowledgments

We acknowledge the Advanced Global Atmospheric Gases Experiment (AGAGE) programs, Scripps Institution of Oceanography (SIO) and National Oceanic and Atmospheric Administration Earth System Research Laboratory Global Monitoring Division (NOAA/ESRL/GMD) for making their atmospheric data available. We thank the efforts of the station operators, managers and support staff at the different monitoring sites of AGAGE, NOAA and the University of East Anglia (UEA), especially Ray Weiss, Paul Krummel, Paul Steele, Martin K. Vollmer and Ray Wang. In particularly, we thank SIO, Diane Ivy and Paul Fraser from the Commonwealth Scientific and Industrial Research Organization (CSIRO), for supplying the archived air samples. We are especially thankful to Simon O'Doherty (University of Bristol, UK) and Johannes Laube (UEA, UK) for providing data. The author is greatly indebted to Matthew Taliaferro for providing useful Matlab code. We also thank financial support from the China Scholarship Council (CSC). We dedicate this work to John Bullister whose tireless work in updating and establishing the source functions for the more traditional transient tracer was a great inspiration to us and served as a role model for our work.



**Table 1.** Atmospheric Metrics of HCFC-22, HCFC-141b, HCFC-142b, HFC-134a, HFC-125, HFC-23, PFC-14 and PFC-116

| Compound | Molecular formula | Atmospheric lifetime [a] | ODP [b] | GWP [c] 100-yr horizon |
|---|---|---|---|---|
| HCFC-22 | $CHClF_2$ | 12 | 0.025 | 1,765 |
| HCFC-141b | $C_2H_3Cl_2F$ | 9.4 | 0.082 | 782 |
| HCFC-142b | $C_2H_3ClF_2$ | 18 | 0.025 | 1,982 |
| HFC-134a | $CH_2FCF_3$ | 14 | 0 | 1,301 |
| HFC-125 | $C_2HF_5$ | 31 | 0 | 3,169 |
| HFC-23 | $CHF_3$ | 228 | 0 | 12,398 |
| PFC-14 | $CF_4$ | > 50,000 | 0 | 6,626 |
| PFC-116 | $C_2F_6$ | > 10,000 | 0 | 11,123 |

[a] See (SPARC, 2013)

[b] ODP: Ozone depletion potential, see (Laube et al., 2013)

[c] GWP: Global warming potential, see (Hodnebrog et al., 2013)



**Table 2a.** Data used for HCFC-22

| Data | Network | Station | Latitude °N | Longitude °E | Instrument | Data availability | Scale | Reference |
|------|---------|---------|-------------|--------------|------------|-------------------|-------|-----------|
| NH | | | | | | | | |
| In situ | AGAGE | Mace Head | 53.3 | -9.9 | ADS | 1999.01-2004.12 | SIO-05 | (Prinn et al., 2000; Prinn et al., 2016) |
| In situ | AGAGE | Mace Head | 53.3 | -9.9 | Medusa | 2003.11-2017.03 | SIO-05 | (Prinn et al., 2000; Prinn et al., 2016) |
| Flask | NOAA | Mace Head | 53.3 | -9.9 | GCMS | 1998.10-2017.07 | NOAA-2006 | (Montzka et al., 1996a) |
| In situ | AGAGE | Trinidad Head | 41.0 | -124.1 | Medusa | 2005.05-2017.03 | SIO-05 | (Prinn et al., 2000; Prinn et al., 2016) |
| Model | NOAA | NH | 30-90 | - | 2-D box | 1944-2009 | NOAA-2006 | (Montzka et al., 2010a) |
| SH | | | | | | | | |
| In situ | AGAGE | Cape Grim | -40.7 | 144.7 | ADS | 1998.03-2004.12 | SIO-05 | (Prinn et al., 2000; Prinn et al., 2016) |
| In situ | AGAGE | Cape Grim | -40.7 | 144.7 | Medusa | 2004.01-2017.03 | SIO-05 | (Prinn et al., 2000; Prinn et al., 2016) |
| Flask | NOAA | Cape Grim | -40.7 | 144.7 | GCMS | 1991.11-2017.07 | NOAA-2006 | (Montzka et al., 1996a) |
| Archived air | AGAGE | Cape Grim | -40.7 | 144.7 | Medusa | 1978.04-1996.12 | SIO-93 | (Miller et al., 1998) |
| Model | NOAA | SH | -30 ~ -90 | - | 2-D box | 1944-2009 | NOAA-2006 | (Montzka et al., 2010a) |

**Table 2b.** Data used for HCFC-141b

| Data | Network | Station | Latitude °N | Longitude °E | Instrument | Data availability | Scale | Reference |
|------|---------|---------|-------------|--------------|------------|-------------------|-------|-----------|
| NH | | | | | | | | |
| Archived air | NOAA | Niwot Ridge | 40.0 | - | GCMS | 1987.01-1994.03 | NOAA-1994 | (Thompson et al., 2004) |
| In situ | AGAGE | Mace Head | 53.3 | -9.9 | ADS | 1994.11-2004.12 | SIO-05 | (Prinn et al., 2000; Prinn et al., 2016) |
| In situ | AGAGE | Mace Head | 53.3 | -9.9 | Medusa | 2003.11-2017.03 | SIO-05 | (Prinn et al., 2000; Prinn et al., 2016) |
| Flask | NOAA | Mace Head | 53.3 | -9.9 | GCMS | 1998.10-2017.07 | NOAA-1994 | (Montzka et al., 1996a) |
| In situ | AGAGE | Trinidad Head | 41.0 | -124.1 | Medusa | 2005.03-2017.03 | SIO-05 | (Prinn et al., 2000; Prinn et al., 2016) |
| SH | | | | | | | | |
| Firn air | AGAGE | Antarctic | -90.0 | -4.8 | ADS | 1935.06-1991.11 | UB-98 | (Sturrock et al., 2002) |
| Archived air | NOAA | - | -29.4 | - | GCMS | 1987.06 | NOAA-1994 | (Thompson et al., 2004) |
| Archived air | UEA | Cape Grim | -40.7 | 144.7 | GCMS | 1978.04-2011.06 | NOAA-1994 | (Oram et al., 1995); |
| In situ | AGAGE | Cape Grim | -40.7 | 144.7 | ADS | 1998.02-2004.12 | SIO-05 | (Prinn et al., 2000; Prinn et al., 2016) |
| In situ | AGAGE | Cape Grim | -40.7 | 144.7 | Medusa | 2004.01-2017.03 | SIO-05 | (Prinn et al., 2000; Prinn et al., 2016) |
| Flask | NOAA | Cape Grim | -40.7 | 144.7 | GCMS | 1994.10-2017.07 | NOAA-1994 | (Montzka et al., 1996a) |





**Table 2c.** Data used for HCFC-142b

| Data | Network | Station | Latitude °N | Longitude °E | Instrument | Data availability | Scale | Reference |
|------|---------|---------|-------------|--------------|------------|-------------------|-------|-----------|
| NH | | | | | | | | |
| Archived air | NOAA | Niwot Ridge | 40.0 | - | GCMS | 1987.01-1994.03 | NOAA-1994 | (Thompson et al., 2004) |
| In situ | AGAGE | Mace Head | 53.3 | -9.9 | ADS | 1994.10-2004.12 | SIO-05 | (Prinn et al., 2000; Prinn et al., 2016) |
| In situ | AGAGE | Mace Head | 53.3 | -9.9 | Medusa | 2003.11-2017.03 | SIO-05 | (Prinn et al., 2000; Prinn et al., 2016) |
| Flask | NOAA | Mace Head | 53.3 | -9.9 | GCMS | 1998.10-2017.07 | NOAA-1994 | (Montzka et al., 1996a) |
| In situ | AGAGE | Trinidad Head | 41.0 | -124.1 | Medusa | 2005.03-2017.03 | SIO-05 | (Prinn et al., 2000; Prinn et al., 2016) |
| SH | | | | | | | | |
| Firn air | AGAGE | Antarctic | -90.0 | -4.8 | ADS | 1936.06-1992.05 | UB-98 | (Sturrock et al., 2002) |
| Archived air | NOAA | - | -29.4 | - | GCMS | 1987.06 | NOAA-1994 | (Thompson et al., 2004) |
| Archived air | UEA | Cape Grim | -40.7 | 144.7 | GCMS | 1978.04-2011.06 | NOAA-1994 | (Oram et al., 1995); |
| In situ | AGAGE | Cape Grim | -40.7 | 144.7 | ADS | 1998.03-2004.12 | SIO-05 | (Prinn et al., 2000; Prinn et al., 2016) |
| In situ | AGAGE | Cape Grim | -40.7 | 144.7 | Medusa | 2004.01-2017.03 | SIO-05 | (Prinn et al., 2000; Prinn et al., 2016) |
| Flask | NOAA | Cape Grim | -40.7 | 144.7 | GCMS | 1992.01-2017.07 | NOAA-1994 | (Montzka et al., 1996a) |

**Table 2d.** Data used for HFC-134a

| Data | Network | Station | Latitude °N | Longitude °E | Instrument | Data availability | Scale | Reference |
|------|---------|---------|-------------|--------------|------------|-------------------|-------|-----------|
| NH | | | | | | | | |
| Archived air | AGAGE | La Jolla | 32.87 | -117.25 | Medusa1 | 1973.06-2016.04 | SIO-05 | this study |
| Archived air | AGAGE | La Jolla | 32.87 | -117.25 | Medusa7.Diane | 1973.06-2011.06 | SIO-05 | this study |
| Archived air | AGAGE | La Jolla | 32.87 | -117.25 | Medusa7 | 1973.10-2012.04 | SIO-05 | this study |
| Archived air | AGAGE | La Jolla | 32.87 | -117.25 | Medusa9.Ben | 1976.01-1999.04 | SIO-05 | this study |
| Archived air | NOAA | Niwot Ridge | 40.0 | - | GCMS | 1976.01-1999.04 | SIO-05 | (Montzka et al., 1996b) |
| In situ | AGAGE | Mace Head | 53.3 | -9.9 | ADS | 1994.10-2004.12 | SIO-05 | (Prinn et al., 2000; Prinn et al., 2016) |
| In situ | AGAGE | Mace Head | 53.3 | -9.9 | Medusa | 2003.11-2017.03 | SIO-05 | (Prinn et al., 2000; Prinn et al., 2016) |
| Flask | NOAA | Mace Head | 53.3 | -9.9 | GCMS | 1998.10-2017.07 | NOAA-1994 | (Montzka et al., 1996a) |
| In situ | AGAGE | Trinidad Head | 41.0 | -124.1 | Medusa | 2005.03-2017.03 | SIO-05 | (Prinn et al., 2000; Prinn et al., 2016) |



| SH | | | | | | | | |
|---|---|---|---|---|---|---|---|---|
| Archived air | AGAGE | Cape Grim | -40.7 | 144.7 | Medusa1 | 1995.02-2011.06 | SIO-05 | this study |
| Archived air | AGAGE | Cape Grim | -40.7 | 144.7 | Medusa7 | 1995.02-2005.03 | SIO-05 | this study |
| Archived air | AGAGE | Cape Grim | -40.7 | 144.7 | Medusa9.Ben | 1978.04-2006.12 | SIO-05 | this study |
| Archived air | NOAA | - | -29.4 | - | GCMS | 1987.06 | NOAA-1995 | (Montzka et al., 1996b) |
| Archived air | UEA | Cape Grim | -40.7 | 144.7 | GCMS | 1990.05-2012.12 | NOAA-1995 | (Oram et al., 1996) |
| In situ | AGAGE | Cape Grim | -40.7 | 144.7 | ADS | 1998.02-2004.12 | SIO-05 | (Prinn et al., 2000; Prinn et al., 2016) |
| In situ | AGAGE | Cape Grim | -40.7 | 144.7 | Medusa | 2004.01-2017.03 | SIO-05 | (Prinn et al., 2000; Prinn et al., 2016) |
| Flask | NOAA | Cape Grim | -40.7 | 144.7 | GCMS | 1994.10-2017.07 | NOAA-1995 | (Montzka et al., 1996a) |

**Table 2e.** Data used for HFC-125

| Data | Network | Station | Latitude °N | Longitude °E | Instrument | Data availability | Scale | Reference |
|---|---|---|---|---|---|---|---|---|
| NH | | | | | | | | |
| Archived air | AGAGE | La Jolla | 32.87 | -117.25 | Medusa1 | 1973.06-2015.11 | SIO-14 | (O'Doherty et al., 2009) |
| Archived air | AGAGE | La Jolla | 32.87 | -117.25 | Medusa7.Diane | 1973.06-2011.06 | UB-98 | (O'Doherty et al., 2009) |
| Archived air | AGAGE | La Jolla | 32.87 | -117.25 | Medusa7 | 1973.10-2012.04 | SIO-14 | (O'Doherty et al., 2009) |
| Archived air | AGAGE | La Jolla | 32.87 | -117.25 | Medusa9.Ben | 1980.05-2007.03 | SIO-14 | (O'Doherty et al., 2009) |
| In situ | AGAGE | Mace Head | 53.3 | -9.9 | ADS | 1998.02-2004.12 | SIO-14 | (Prinn et al., 2000; Prinn et al., 2016) |
| In situ | AGAGE | Mace Head | 53.3 | -9.9 | Medusa | 2003.11-2017.03 | SIO-14 | (Prinn et al., 2000; Prinn et al., 2016) |
| In situ | AGAGE | Trinidad Head | 41.0 | -124.1 | Medusa | 2005.03-2016.08 | SIO-14 | (Prinn et al., 2000; Prinn et al., 2016) |
| Flask | NOAA | Trinidad Head | 41.0 | -124.1 | GCMS_M2 | 2007.01-2015.04 | NOAA-2008 | (Montzka et al., 1996a) |
| SH | | | | | | | | |
| Archived air | AGAGE | Cape Grim | -40.7 | 144.7 | Medusa1 | 1995.02-2011.06 | SIO-14 | (O'Doherty et al., 2009) |
| Archived air | AGAGE | Cape Grim | -40.7 | 144.7 | Medusa7.Diane | 1995.02-2001.09 | UB-98 | (O'Doherty et al., 2009) |
| Archived air | AGAGE | Cape Grim | -40.7 | 144.7 | Medusa7 | 1995.02-2004.12 | SIO-14 | (O'Doherty et al., 2009) |
| Archived air | AGAGE | Cape Grim | -40.7 | 144.7 | Medusa9.Ben | 1978.04-2006.12 | SIO-14 | (O'Doherty et al., 2009) |
| In situ | AGAGE | Cape Grim | -40.7 | 144.7 | ADS | 1998.02-2004.12 | SIO-14 | (Prinn et al., 2000; Prinn et al., 2016) |
| In situ | AGAGE | Cape Grim | -40.7 | 144.7 | Medusa | 2004.02-2017.03 | SIO-14 | (Prinn et al., 2000; Prinn et al., 2016) |
| Flask | NOAA | Cape Grim | -40.7 | 144.7 | GCMS_M2 | 2007.01-2015.04 | NOAA-2008 | (Montzka et al., 1996a) |



**Table 2f.** Data used for HFC-23

| Data | Network | Station | Latitude °N | Longitude °E | Instrument | Data availability | Scale | Reference |
|---|---|---|---|---|---|---|---|---|
| NH | | | | | | | | |
| In situ | AGAGE | Mace Head | 53.3 | -9.9 | Medusa | 2007.10-2017.03 | SIO-07 | (Prinn et al., 2000; Prinn et al., 2016) |
| In situ | AGAGE | Trinidad Head | 41.0 | -124.1 | Medusa | 2007.09-2017.03 | SIO-07 | (Prinn et al., 2000; Prinn et al., 2016) |
| Model | AGAGE | NH | 30-90 | - | 2-D 12-box | 1978.01-2009.12 | SIO-07 | (Miller et al., 2010) |
| SH | | | | | | | | |
| Archived air | AGAGE | Cape Grim | -40.7 | 144.7 | Medusa3 | 2005.04-2009.11 | SIO-07 | (Miller et al., 2010) |
| Archived air | AGAGE | Cape Grim | -40.7 | 144.7 | Medusa9 | 1978.04-2006.12 | SIO-07 | (Miller et al., 2010) |
| In situ | AGAGE | Cape Grim | -40.7 | 144.7 | Medusa | 2007.01-2017.03 | SIO-07 | (Prinn et al., 2000; Prinn et al., 2016) |
| Model | AGAGE | SH | -30 ~ -90 | - | 2-D 12-box | 1978.01-2009.12 | SIO-07 | (Miller et al., 2010) |

**Table 2g.** Data used for PFC-14

| Data | Network | Station | Latitude °N | Longitude °E | Instrument | Data availability | Scale | Reference |
|---|---|---|---|---|---|---|---|---|
| NH | | | | | | | | |
| In situ | AGAGE | Mace Head | 53.3 | -9.9 | Medusa | 2006.05-2017.03 | SIO-05 | (Prinn et al., 2000; Prinn et al., 2016) |
| In situ | AGAGE | Trinidad Head | 41.0 | -124.1 | Medusa | 2006.04-2017.03 | SIO-05 | (Prinn et al., 2000; Prinn et al., 2016) |
| Model | AGAGE | NH | 30-90 | - | 2-D 12-box | 1900-2015 | SIO-05 | (Trudinger et al., 2016) |
| SH | | | | | | | | |
| In situ | AGAGE | Cape Grim | -40.7 | 144.7 | Medusa | 2006.05-2017.03 | SIO-05 | (Prinn et al., 2000; Prinn et al., 2016) |
| Model | AGAGE | SH | -30 ~ -90 | - | 2-D 12-box | 1900-2015 | SIO-05 | (Trudinger et al., 2016) |

**Table 2h.** Data used for PFC-116

| Data | Network | Station | Latitude °N | Longitude °E | Instrument | Data availability | Scale | Reference |
|---|---|---|---|---|---|---|---|---|
| NH | | | | | | | | |
| In situ | AGAGE | Mace Head | 53.3 | -9.9 | Medusa | 2003.11-2017.03 | SIO-07 | (Prinn et al., 2000; Prinn et al., 2016) |
| In situ | AGAGE | Trinidad Head | 41.0 | -124.1 | Medusa | 2005.05-2017.03 | SIO-07 | (Prinn et al., 2000; Prinn et al., 2016) |
| Model | AGAGE | NH | 30-90 | - | 2-D 12-box | 1900-2015 | SIO-07 | (Trudinger et al., 2016) |





| SH | | | | | | | | |
|---|---|---|---|---|---|---|---|---|
| In situ | AGAGE | Cape Grim | -40.7 | 144.7 | Medusa | 2004.04-2017.03 | SIO-07 | (Prinn et al., 2000; Prinn et al., 2016) |
| Model | AGAGE | SH | -30 ~ -90 | - | 2-D 12-box | 1900-2015 | SIO-07 | (Trudinger et al., 2016) |



**Table 3.** Primary scale conversion factors for HCFC-22, HCFC-141b, HCFC-142b, HFC-134a, HFC-125 and HFC-23 between AGAGE (UB and SIO) and NOAA [a]

| HCFC-22 | SIO-93 | SIO-98 | SIO-05 | NOAA-1992 |
|---|---|---|---|---|
| SIO-98 | 1.0053 [b] | - | - | - |
| NOAA-1992 | 0.997 ± 0.004 [c] | 0.993 ± 0.007 [b] | - | - |
| NOAA-2006 | - | - | 0.997 ± 0.003 [d] | 1.005 [e] |
| **HCFC-141b** | | UB-98 | SIO-05 | |
| NOAA-1994 | | 1.006 ± 0.003 [b] | 0.994 ± 0.005 [d] | |
| **HCFC-142b** | | UB-98 | SIO-05 | |
| NOAA-1994 | | 0.937 ± 0.003 [b] | 0.974 ± 0.005 [d] | |
| **HFC-134a** | | UB-98 | SIO-05 | |
| NOAA-1995 | | 1.035 ± 0.004 [b] | 1.001 ± 0.005 [d] | |
| **HFC-125** | | UB-98 | SIO-14 | |
| SIO-14 | | 1.0826 [f] | - | |
| NOAA-2008 | | - | 0.946 ± 0.008 [d] | |

[a] Example for using conversion factor: HCFC-22 measurement results reported on the SIO-98 multiply 1.0053 equal their results reported on SIO-93. AGAGE: Advanced Global Atmospheric Gases Experiment, UB: University of Bristol, SIO: Scripps Institution of Oceanography, NOAA: National Oceanic and Atmospheric Administration.

[b] (Prinn et al., 2000).

[c] (Miller et al., 1998).

[d] (Simmonds et al., 2017a).

[e] NOAA calibration scales for various trace gases (https://www.esrl.noaa.gov/gmd/ccl/scales.html).

[f] The AGAGE scale (http://agage.eas.gatech.edu/data_archive/agage/AGAGE_scale_2017_v1.pdf).





**Table 4.** Ostwald solubility coefficients of HCFC-22, HCFC-141b, HCFC-142b, HFC-134a, HFC-125, HFC-23, PFC-14, PFC-116 and CFC-12 in seawater

| Compound | $a_1$ | $a_2$ | $a_3$ | $K_S$ | $T_{min}$ (K) | $T_{max}$ (K) | $L_0$ at 1 atm, 25 °C (L L$^{-1}$) | $L$ at 1 atm, 25 °C, 35.0 ‰ (L L$^{-1}$) |
|---|---|---|---|---|---|---|---|---|
| HCFC-22 | -66.9256 | 109.8625 | 27.3778 | $0.169 \pm 0.037$ | 278.15 | 353.15 | 0.844 | 0.669 |
| HCFC-141b | -85.6439 | 138.0940 | 35.6875 | $0.204 \pm 0.043$ | 278.15 | 353.15 | 0.711 | 0.537 |
| HCFC-142b | -73.3682 | 118.3104 | 29.8797 | $0.198 \pm 0.037$ | 278.15 | 353.15 | 0.352 | 0.268 |
| HFC-134a | -67.1680 | 109.1227 | 27.0984 | $0.193 \pm 0.034$ | 278.15 | 353.15 | 0.381 | 0.292 |
| HFC-125 | -51.8823 | 84.5045 | 19.3067 | $0.224 \pm 0.028$ | 283.15 | 343.15 | 0.086 | 0.063 |
| HFC-23 | 30.0046 | -31.6631 | -18.8072 | $0.168 \pm 0.028$ | 278.15 | 348.15 | 0.313 | 0.249 |
| PFC-14 | -113.8218 | 162.6686 | 49.4215 | $0.202 \pm 0.016$ | 273.15 | 328.15 | 0.00513 | 0.00388 |
| PFC-116 | -102.0437 | 147.9210 | 41.9999 | $0.244 \pm 0.017$ | 278.15 | 328.15 | 0.00143 | 0.00102 |
| CFC-12 | -101.3445 | 156.4709 | 42.2833 | $0.204 \pm 0.034$ | 273.15 | 348.15 | 0.069 | 0.052 |

$$L = 10^{-K_S \cdot S/M_{NaCl}} \cdot exp \left[ a_1 + a_2 \cdot (\frac{100}{T}) + a_3 \cdot ln(\frac{T}{100}) \right]$$





**Table 5**. *E*, *S*, *A*, *B*, *V* descriptors of HCFC-22, HCFC-141b, HCFC-142b, HFC-134a, HFC-125, HFC-23, PFC-14, PFC-116 and CFC-12

| Species | Chemical Formula | $E$ | $S$ | $A$ | $B$ | $V$ | Corrected $V$ |
|---|---|---|---|---|---|---|---|
| HCFC-22 | $CHClF_2$ | -0.056 | 0.380 | 0.040 | 0.050 | 0.4073 | 0.4473 |
| HCFC-141b | $C_2H_3Cl_2F$ | 0.084 | 0.430 | 0.005 | 0.054 | 0.6530 | 0.6729 |
| HCFC-142b | $C_2H_3ClF_2$ | -0.080 | 0.240 | 0.060 | 0.056 | 0.5482 | 0.5882 |
| HFC-134a | $CH_2FCF_3$ | -0.410 | 0.342 | 0.060 | 0.040 | 0.4612 | 0.5412 |
| HFC-125 | $C_2HF_5$ | -0.510 | -0.019 | 0.105 | 0.064 | 0.4789 | 0.6445 |
| HFC-23 | $CHF_3$ | -0.427 | 0.183 | 0.110 | 0.034 | 0.3026 | 0.3626 |
| PFC-14 | $CF_4$ | -0.550 | -0.250 | 0.000 | 0.000 | 0.3203 | 0.4003 |
| PFC-116 | $C_2F_6$ | -0.590 | -0.350 | 0.000 | 0.000 | 0.4966 | 0.6166 |
| CFC-12 | $CCl_2F_2$ | 0.027 | 0.125 | 0.000 | 0.000 | 0.5297 | 0.5697 |



**Table 6.** Comparison of the Ostwald solubility coefficients ($L$, L L$^{-1}$) of PFC-14 in seawater in this study with the results from the literature

| $T$ (K) | $t$ (°C) | $S$ (‰) | $L$, this study | $L$, (Scharlin and Battino, 1995) | $RSD$ [a] (%) |
|---|---|---|---|---|---|
| 288.15 | 15 | 35.086 | 0.005052 | 0.005169 | 1.62 |
| 293.15 | 20 | 35.086 | 0.004578 | 0.004527 | 0.79 |
| 298.15 | 25 | 35.086 | 0.004217 | 0.004027 | 3.26 |
| 303.15 | 30 | 35.086 | 0.003944 | 0.003635 | 5.77 |

[a] Relative standard deviation (RSD) of the Ostwald solubility coefficients estimated in this study and measured in Scharlin and Battino (1995)



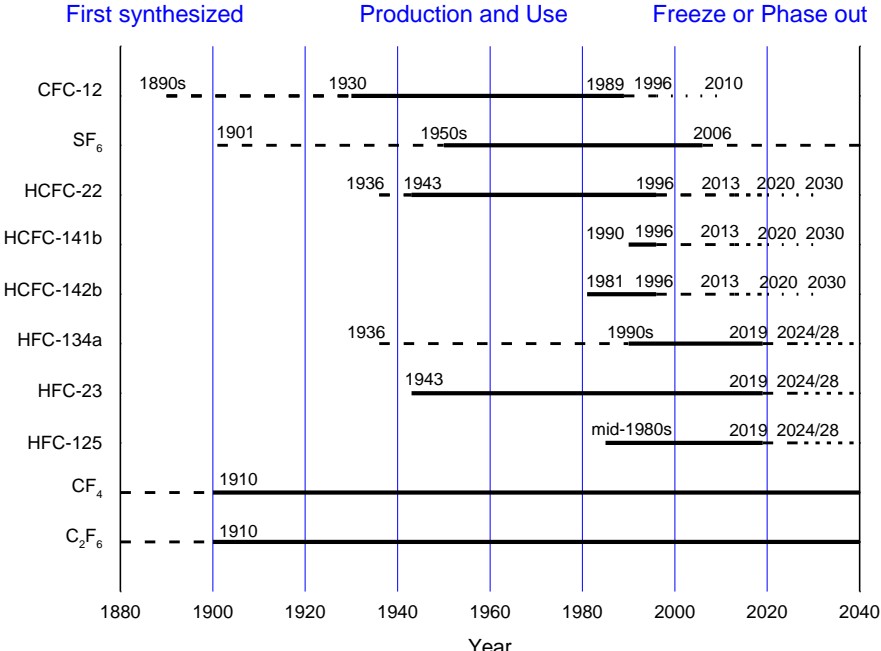

**Figure 1.** Comparison of production and use histories of CFC-12, SF₆, HCFCs, HFCs and PFCs




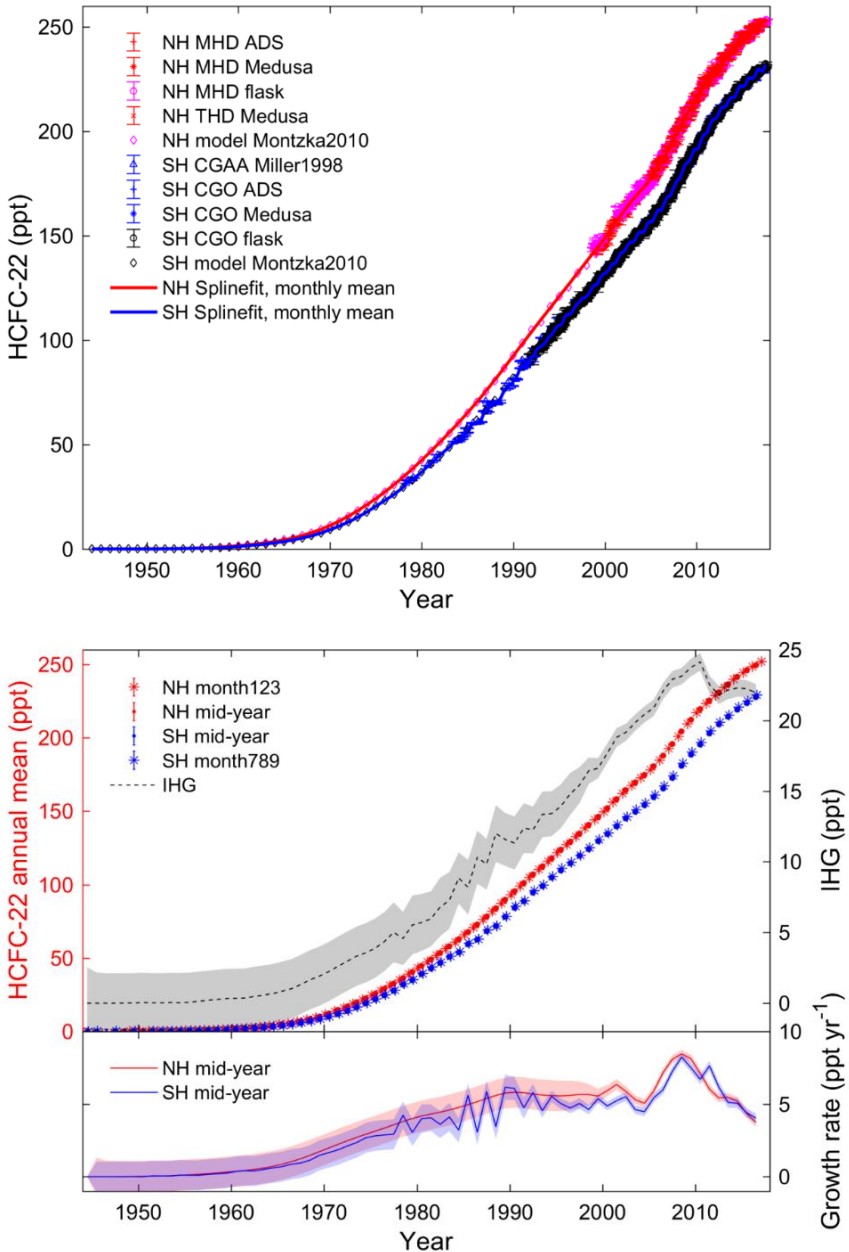

**Figure 2. (a)** The northern hemisphere (NH) and southern hemisphere (SH) monthly means atmospheric history for HCFC-22 obtained from the smoothing spline fit on all collected data. The data includes the AGAGE in situ measurements at MHD and THD for NH and at CGO for SH, NOAA flask air measurements (MHD and CGO) and 2-D box model mole fractions (Montzka et al., 2010a) for both hemispheres. The SH Cape Grim Air Archive measurements (Miller et al., 1998) is also included. **(b)** The top panel shows the reconstructed mid-February, mid-year and mid-August annual mean atmospheric mole fractions for HCFC-22 in the NH and SH and the inter-hemispheric gradients (IHG, in black using the right axes). The low panel shows the mid-year annual growth rates in ppt yr$^{-1}$. Shadings in the figure reflect the uncertainty.





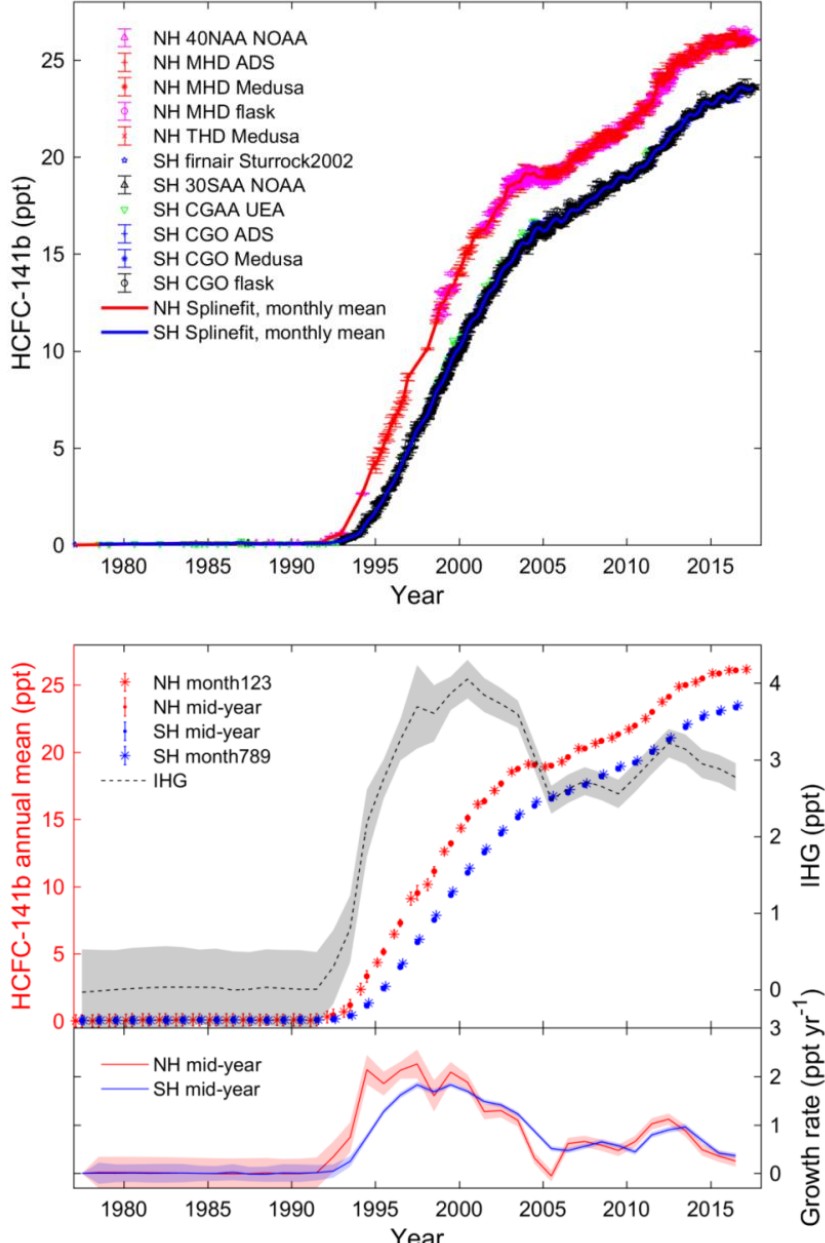

**Figure 3. (a)** The northern hemisphere (NH) and southern hemisphere (SH) monthly means atmospheric history for HCFC-141b obtained from the smoothing spline fit on all collected data. The data includes the AGAGE in situ measurements at MHD and THD for NH and at CGO for SH, NOAA flask (MHD and CGO) and archived air measurements for both hemispheres. The SH firn air record (Sturrock et al., 2002) is also included. **(b)** The top panel shows the reconstructed mid-February, mid-year and mid-August annual mean atmospheric mole fractions for HCFC-141b in the NH and SH and the inter-hemispheric gradients (IHG, in black using the right axes). The low panel shows the mid-year annual growth rates in ppt yr$^{-1}$. Shadings in the figure reflect the uncertainty.

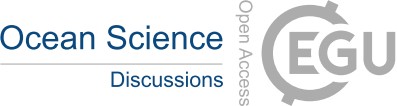



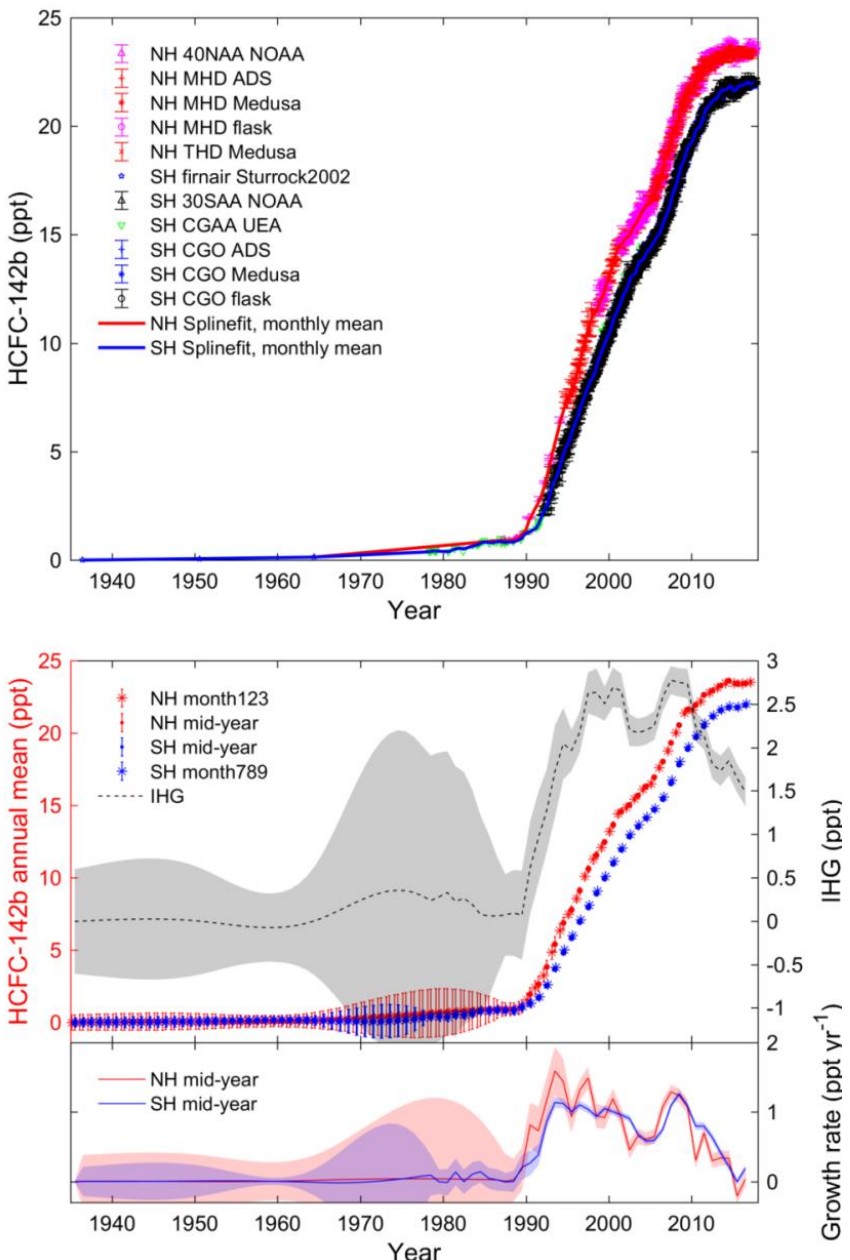

**Figure 4. (a)** The northern hemisphere (NH) and southern hemisphere (SH) monthly means atmospheric history for HCFC-142b obtained from the smoothing spline fit on all collected data. The data includes the AGAGE in situ measurements at MHD and THD for NH and at CGO for SH, NOAA flask (MHD and CGO) and archived air measurements for both hemispheres. The SH firn air record (Sturrock et al., 2002) is also included. **(b)** The top panel shows the reconstructed mid-February, mid-year and mid-August annual mean atmospheric mole fractions for HCFC-142b in the NH and SH and the inter-hemispheric gradients (IHG, in black using the right axes). The low panel shows the mid-year annual growth rates in ppt yr$^{-1}$. Shadings in the figure reflect the uncertainty.




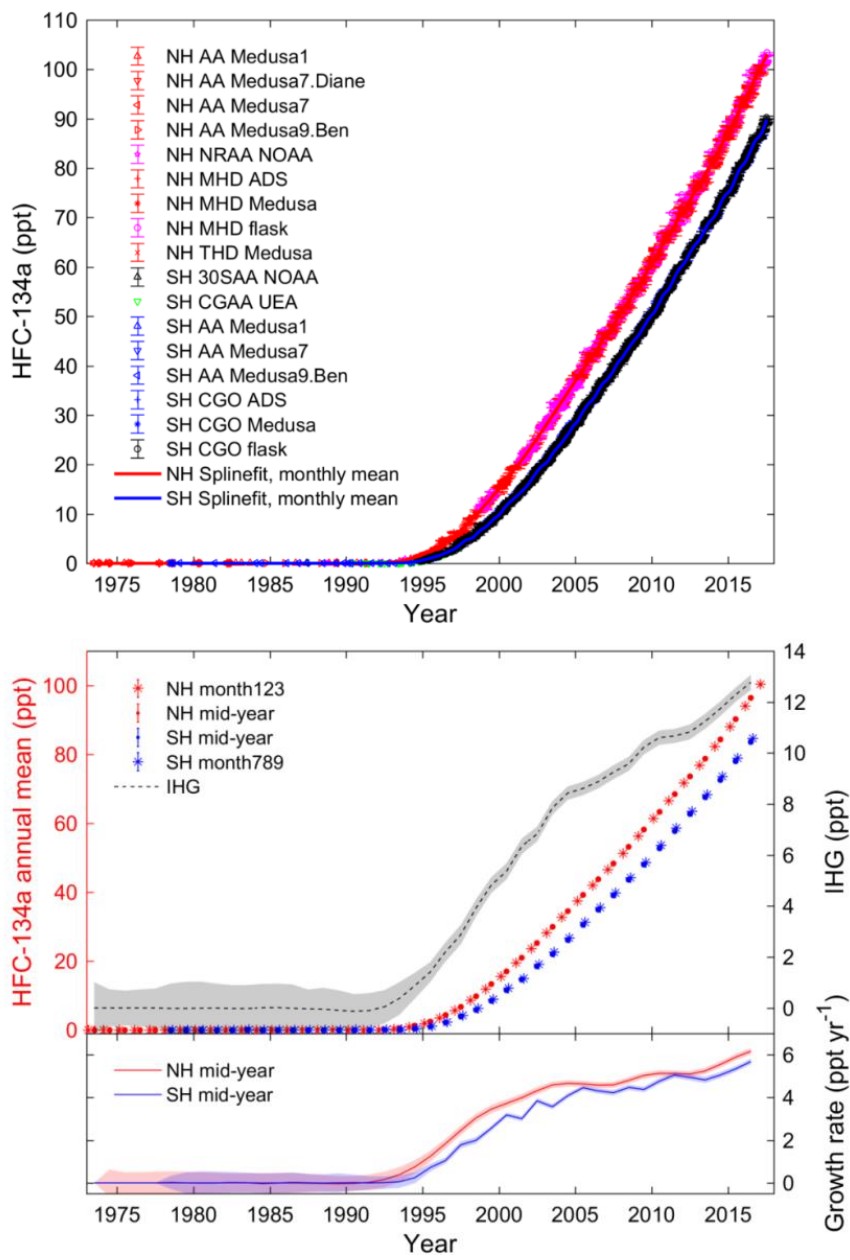

**Figure 5. (a)** The northern hemisphere (NH) and southern hemisphere (SH) monthly means atmospheric history for HFC-134a obtained from the smoothing spline fit on all collected data. The data includes the AGAGE in situ measurements at MHD and THD for NH and at CGO for SH, NOAA flask (MHD and CGO) and archived air measurements for both hemispheres. The AGAGE air archive measurements for both hemispheres and UEA CGAA data are also included. **(b)** The top panel shows the reconstructed mid-February, mid-year and mid-August annual mean atmospheric mole fractions for HFC-134a in the NH and SH and the inter-hemispheric gradients (IHG, in black using the right axes). The low panel shows the mid-year annual growth rates in ppt yr⁻¹. Shadings in the figure reflect the uncertainty.





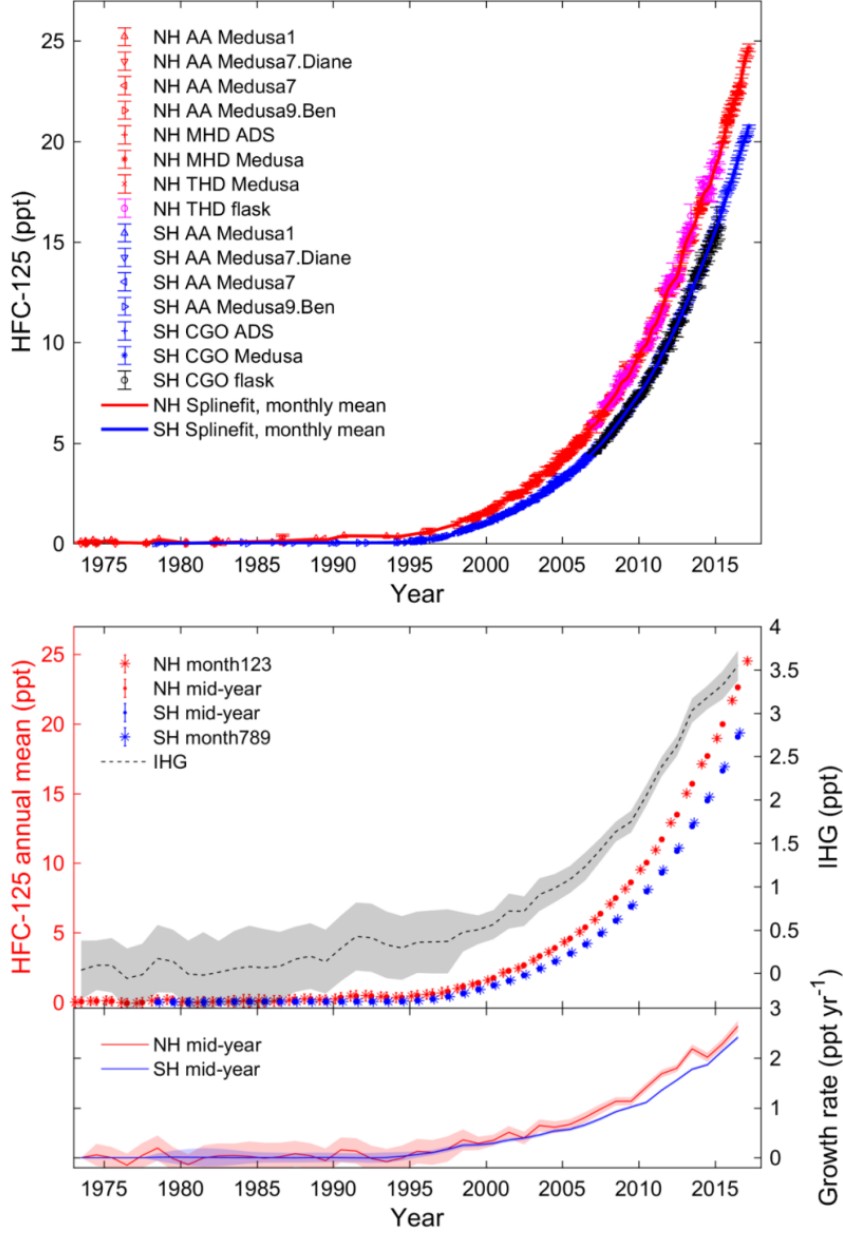

**Figure 6. (a)** The northern hemisphere (NH) and southern hemisphere (SH) monthly means atmospheric history for HFC-125 obtained from the smoothing spline fit on all collected data. The data includes the AGAGE in situ measurements at MHD and THD for NH and at CGO for SH, NOAA flask (MHD and CGO) and archived air measurements for both hemispheres. The AGAGE air archive measurements for both hemispheres and UEA CGAA data are also included. **(b)** The top panel shows the reconstructed mid-February, mid-year and mid-August annual mean atmospheric mole fractions for HFC-125 in the NH and SH and the inter-hemispheric gradients (IHG, in black using the right axes). The low panel shows the mid-year annual growth rates in ppt yr[-1]. Shadings in the figure reflect the uncertainty.

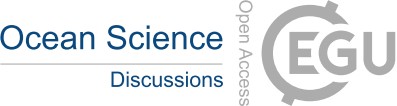

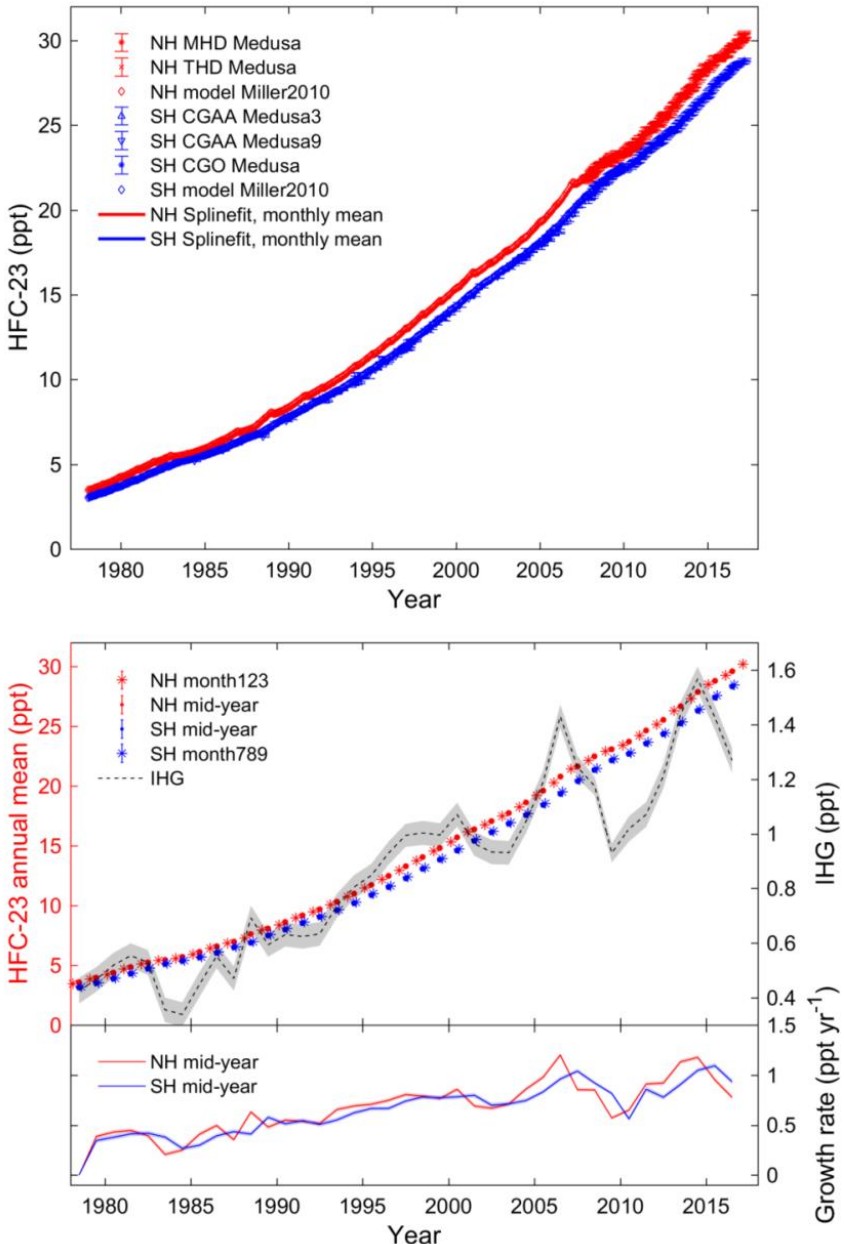

**Figure 7. (a)** The northern hemisphere (NH) and southern hemisphere (SH) monthly means atmospheric history for HFC-23 obtained from the smoothing spline fit on all collected data. The data includes the AGAGE in situ measurements at MHD and THD for NH and at CGO for SH, 2-D 12-box model data (Miller et al., 2010) for both hemispheres. The AGAGE CGAA measurements are also included. **(b)** The top panel shows the reconstructed mid-February, mid-year and mid-August annual mean atmospheric mole fractions for HFC-23 in the NH and SH and the inter-hemispheric gradients (IHG, in black using the right axes). The low panel shows the mid-year annual growth rates in ppt yr$^{-1}$. Shadings in the figure reflect the uncertainty.





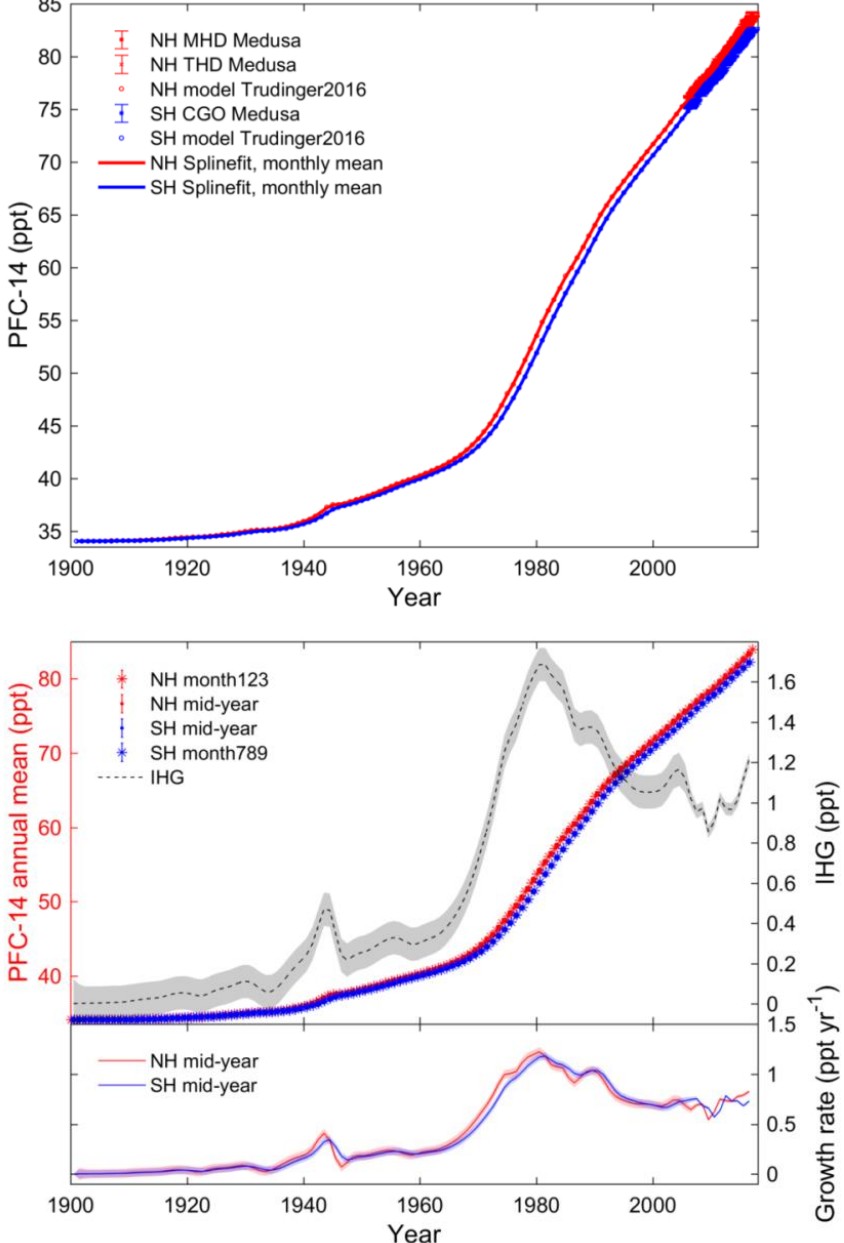

**Figure 8. (a)** The northern hemisphere (NH) and southern hemisphere (SH) monthly means atmospheric history for PFC-14 obtained from the smoothing spline fit on all collected data. The data includes the AGAGE in situ measurements at MHD and THD for NH and at CGO for SH, model data (Trudinger et al., 2016) for both hemispheres. **(b)** The top panel shows the reconstructed mid-February, mid-year and mid-August annual mean atmospheric mole fractions for PFC-14 in the NH and SH and the inter-hemispheric gradients (IHG, in black using the right axes). The low panel shows the mid-year annual growth rates in ppt yr$^{-1}$. Shadings in the figure reflect the uncertainty.





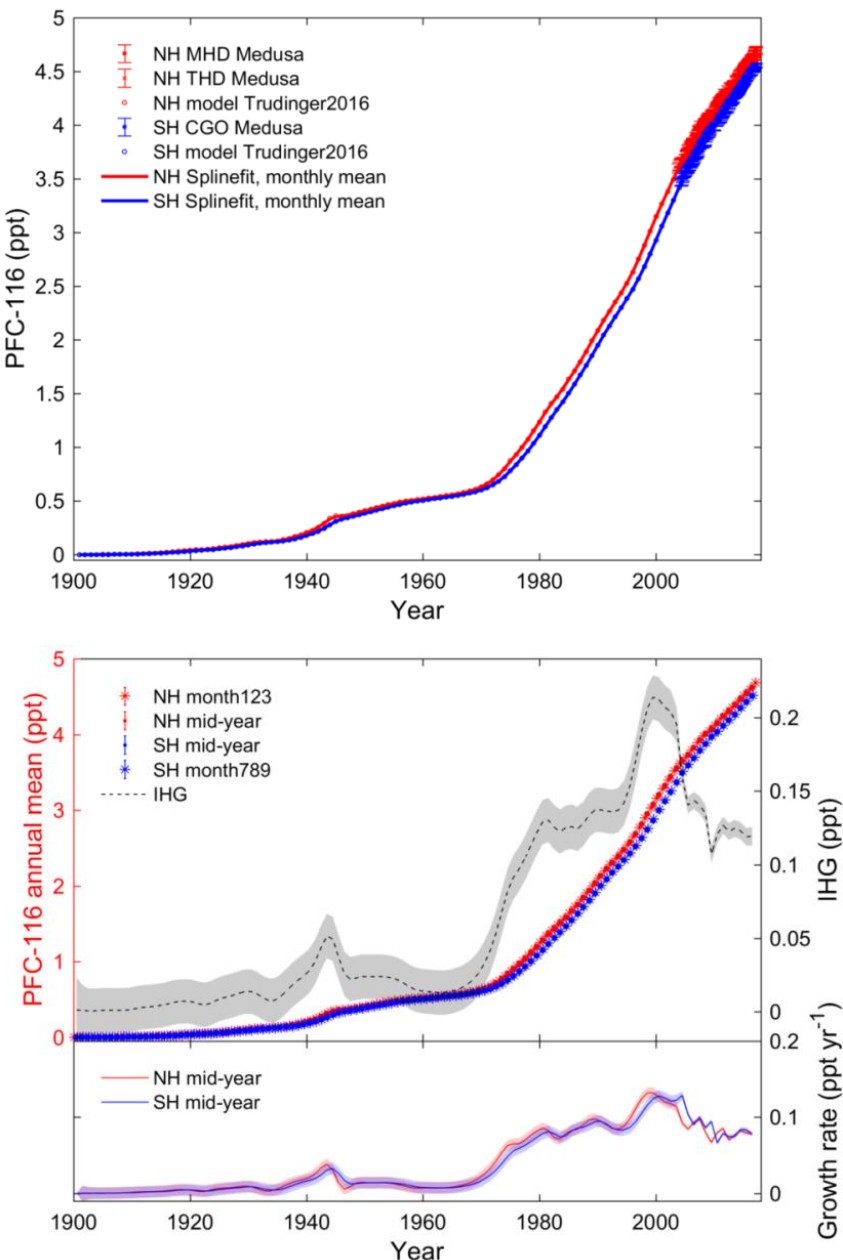

**Figure 9. (a)** The northern hemisphere (NH) and southern hemisphere (SH) monthly means atmospheric history for PFC-116 obtained from the smoothing spline fit on all collected data. The data includes the AGAGE in situ measurements at MHD and THD for NH and at CGO for SH, model data (Trudinger et al., 2016) for both hemispheres. **(b)** The top panel shows the reconstructed mid-February, mid-year and mid-August annual mean atmospheric mole fractions for PFC-116 in the NH and SH and the inter-hemispheric gradients (IHG, in black using the right axes). The low panel shows the mid-year annual growth rates in ppt yr$^{-1}$. Shadings in the figure reflect the uncertainty.





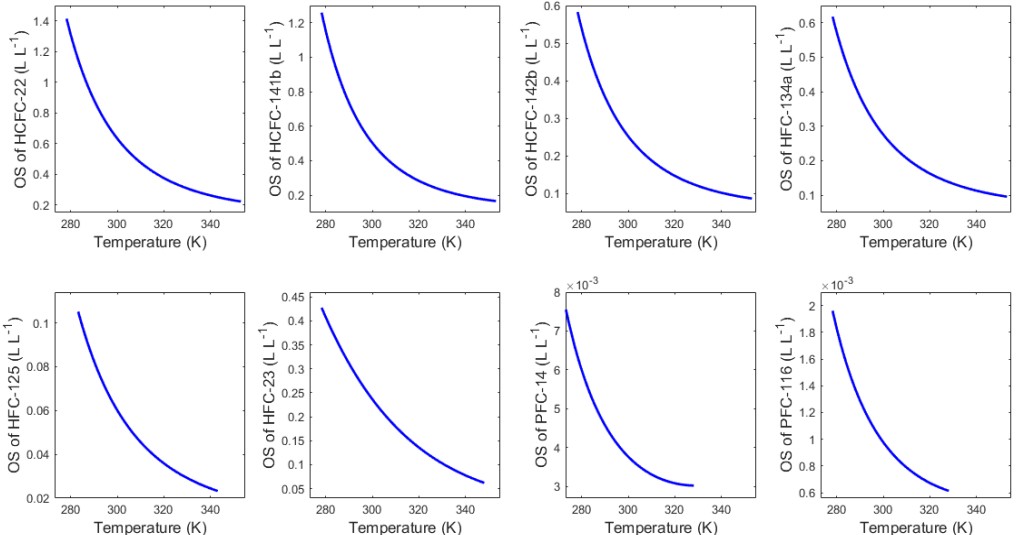

**Figure 10**. The Ostwald solubility (OS) coefficients in the available temperature range in seawater for each compound. The salinity is set at 35.

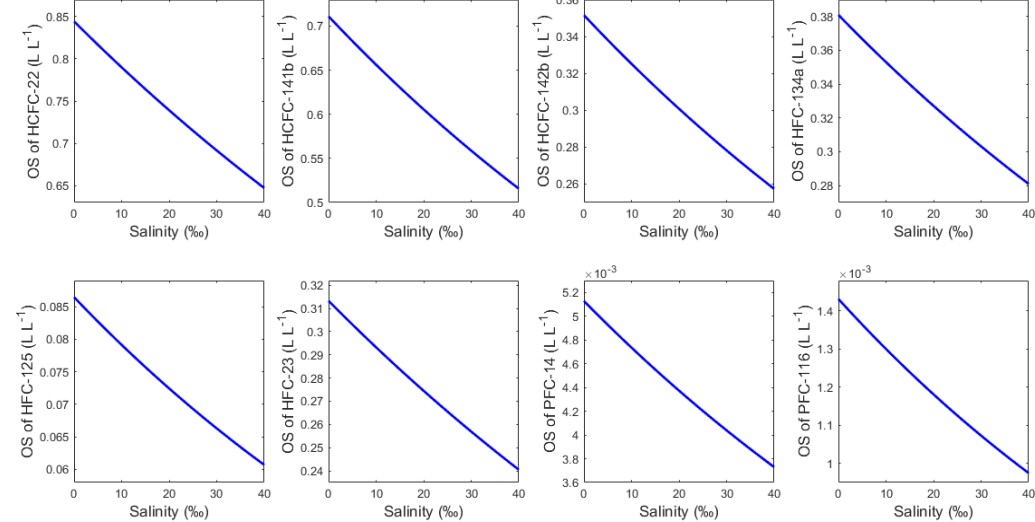

**Figure 11**. The Ostwald solubility (OS) coefficients in the salinity range of 0-40 in seawater for each compound. The temperature is set at 298.15 K.




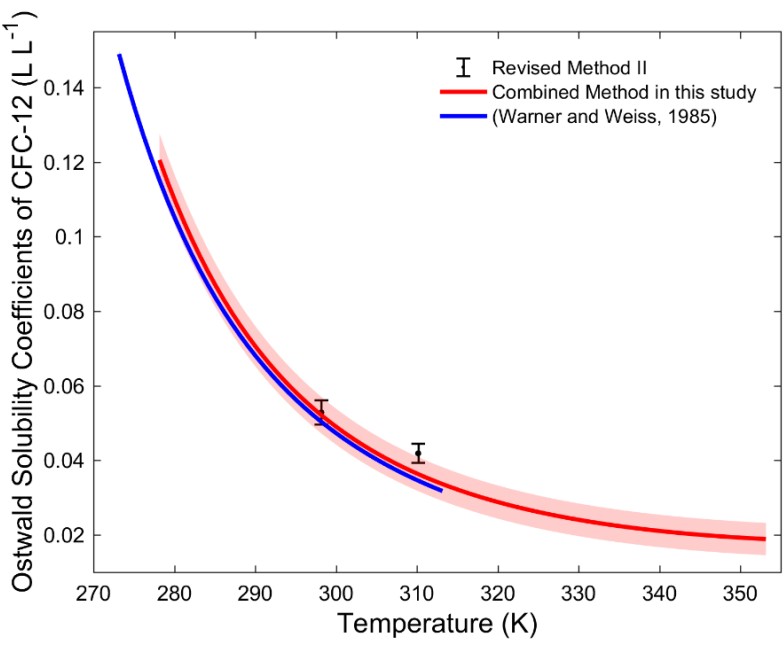

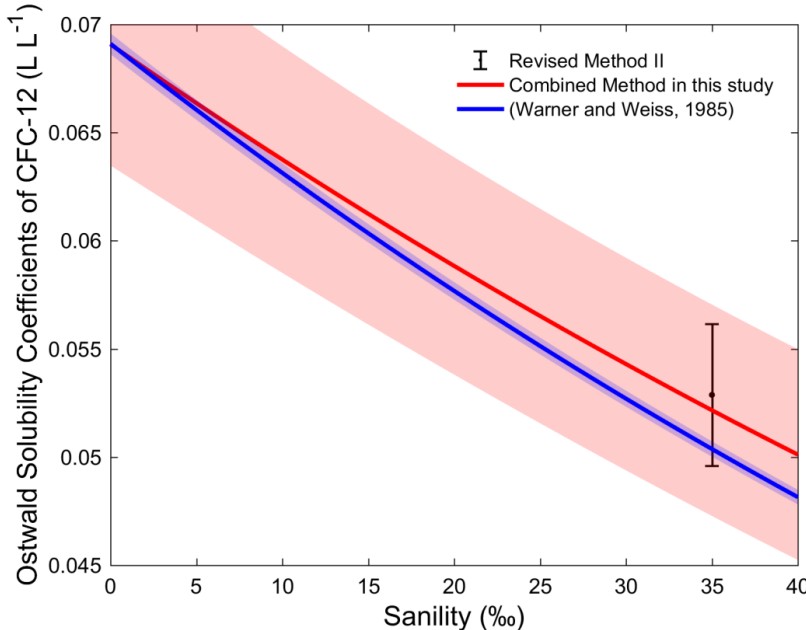

**Figure 12.** **(a)** Comparison of the Ostwald solubility coefficients in the available temperature range at the permanent salinity 35 and **(b)** comparison of the Ostwald solubility coefficients in the salinity range of 0-40 at permanent 298.15 K for CFC-12 from the seawater solubility functions constructed by the Revised Method II, the Combined Method in this study and Warner and Weiss (1985). Error bar and shadings in the figure reflect the uncertainty.



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
