# Peer review of "Atmospheric Histories, Growth Rates and Solubilities in Seawater and other Natural Waters of the Potential Transient Tracers HCFC-22, HCFC-141b, HCFC-142b, HFC-134a, HFC-125, HFC-23, PFC-14 and PFC-116"

_Ocean Science, 2018_

## Referee Comment (RC1) · Anonymous Referee #1 · 9 Sep 2018

"GENERAL COMMENTS"

The authors synthesize the atmospheric concentration history and review the solubility of HCFC-22, HCFC-141b, HCFC-142b, HFC-134a, HFC-125, HFC-23, PFC-14 and PFC-116. This study is valuable as a first step of evaluating the utility of these compounds as oceanic transient tracers. Some revisions seem to be needed. In particular, the following two revisions are needed.

[Figure]

First, estimate of Ostwald solubility coefficients by LFERs (Method II) should be revised. According to Abraham et al. (2001), the solvation parameter method of Abraham relies on two linear free energy relationships, LFERs, one for processes within condensed phases, Eq. (i), and one for processes involving gas to condensed phase transfer, Eq. (ii).

$\log SP = c + eE + sS + aA + bB + vV$ (i)

$\log SP = c + eE + sS + aA + bB + lL16$ (ii)

Here L16 is the solute gas-hexadecane partition coefficient at 298 K; l is a coefficient for L16; and other coefficients and descriptors are the same as those described in the manuscript. Therefore, Ostwald solubility coefficients in pure water (L0) can be estimated by use of Eq. (ii) (processes involving gas to condenses phase transfer), while salting-out coefficients (ks) can be estimated by use of Eq. (i) (processes within condensed phases). This point should be noted clearly in the manuscript.

In this study, as seen in Eqs. 10, 12 and 13 (Sect. 2.10.2), the relationship for water to solvent (processes within condensed phases, Eq. (i)) was used to estimate Ostwald solubility coefficient in pure water (L0). It should be revised as mentioned above. Furthermore, Tables S2 and S4 should be revised. When Eq. (ii) is used to estimate L0, may the discussion in Sect. 3.3 (page 20), which includes comparison between the Method II and the Revised Method II, lead to the same result as described in the manuscript?

Second, hydrolysis of HCFC-22 should be taken into consideration in discussion of transient tracer potential (for example, Sect. 3.4) because rate constants for hydrolysis of HCFC-22 in alkaline aqueous solutions are much larger than those for hydrolysis of other HCFCs and HFCs [for example, le Noble, W. J. Am. Chem. Soc., 87, 2434-2438 (1965); Kutsuna, S. et al. Int. J. Chem. Kinet. 43, 639-647 (2011)].

May hydrolysis of HCFC-22 in seawater make a significant influence on transient tracer

potential of HCFC-22? If hydrolysis of HCFC-22 is significant, what would be expected when HCFC-22 is used as a transient tracer?

"SPECIFIC COMMENTS"

Page 4, lines 21-22: Combustion in thermal power station has been pointed out as a tropospheric sink of PFCs [Ravishankara, A. R. et al., Science, 259, 194-199 (1993)]. It should be cited.

Page 4, lines 28-30, "Cutting the production and consumption of HFCs by more than 80 % over the next 30 years under the Kigali amendment of the MP": Under the Kigali amendment, the amount of cutting the production and consumption of HFCs is based on the amount scaled by GWP of each HFC. This point had better be described.

Page 5, lines 17-18: Why were the same AGAGE calibration scales used by converting NOAA and UEA data to the AGAGE scale? Is the reason explained somewhere in the manuscript?

Page 10, line 3, "The ionic strength of seawater (Iv, in g L−1): The unit of ionic strength should be checked.

Page 10, Eq. (8): DS, ES and FS seems to need definition.

Page 10, line 13, unit of the McGowan's characteristic molar volume in Eq. 8: According to Abraham et al. (2001), unit of the McGowan's characteristic molar volume is not cm3 mol−1/100 but dm3 mol−1/100. The unit should be checked.

Page 12, line 34, Figures 2-9: The method to calculate IHG and its error (gray parts in Figures 2-9) seems to need explanation.

Page 14, line 11: What does it mean by "30% (median) larger"? It seems to need more explanation.

Page 16, lines 27-29: Atmospheric lifetime of HFC-23 is much longer than that of HFC-134a. Hence, time-profiles are expected to be different between HFC-134a and

HFC-23 after the consumption restrictions imposed by the 2016 Kigali Amendment to the Montreal Protocol. This point should be discussed.

Page 16, lines 34-35, "the growth rates in both hemispheres are really similar": Do the growth rates mean those of HFC-125 or some target compounds? What does it mean by "really similar"?

Page 17, lines 28-29, "For compounds with shorter lifetimes...": This sentence is difficult to understand. It should be revised.

Page 18, lines 7-9: The experimental data for HFC-125 are scattered as seen in Fig. S5. This point had better be described more clearly.

Page 18, lines 37-38: The dipolarity/polarizability (S) for $C_2F_6$ was estimated to be equal to the average of the S of $CF_4$ and $C_3F_8$. This estimate might have substantial errors. Influence of errors of S on the salting-out coefficients of $C_2F_6$ should be evaluated.

Page 19, line 32: How is uncertainty from salting-out coefficients estimated? It seems to need more explanation.

Supplement, Eq. (2): What does it mean by i = 0 in summation?

"TECHNICAL CORRECTIONS"

Page 5, line 21: "can used" is "can be used".

Page 6, line 32: "fin air" is "firn air".

Page 12, line 11: "By combining Eq. (4), (12) and (14)" may be "By combining Eqs. (4), (14) and (16)".

Page 12, line 39 – page 13, line 1: "hemispheric" is "hemisphere".

Page 13, line 21; page 13, line 39: "This consistent" is "This is consistent"

Page 15, line 13: "These consistent" is "These are consistent"

Page 16, line 17: "0.038 ± 0.007" is "0.138 ± 0.007".

Page 16, line 17: "0.033 ± 0.008" is "0.133 ± 0.008".

Page 16, line 34: "This could attributed" is "This could be attributed".

Page 17, line 9: "This consistent" is "This was consistent".

Page 17, line 13: "countries, Moreover," is "countries. Moreover,".

Page 18, line 9: "is chose" is "is chosen".

Page 18, line 32: "298.15-338.15 K" is "273.15-313.15 K".

Page 19, line 31: "9.0695e-05" is "$9.0695 \times 10{-5}$".

Page 19, line 31: "9.1858e-05" is "$9.1858 \times 10{-5}$".

Page 21, line 38: "can be also be" is "can also be".

---

## Referee Comment (RC2) · Anonymous Referee #2 · 19 Sep 2018

This work the authors compile a large set of data to investigate the atmospheric history, growth rates, seawater solubility of a series of halocarbons. These are useful information to start understanding the suitability for these gases to be used as tracers. The paper is well-written, although I found that since a major motivation for the authors to investigate all these parameters was to assess the possibility to use these halocarbon as tracers, the title should be modified to reflect this. This paper presented an important initial step to assess new generation of ocean tracers and should be published

after minor revisions. My specific comments are listed below.

Page 5 lines 3 to 5: the terminology of "mid-February" and "mid-August" used to define means from January to March and July to September is focusing. I suggest the authors use a different terminology for these.

Page 6 line 33: should read "The firn air data. . ."

Page 12 line 23: should read "Based on this, we reconstructed. . ."

Page 12 lines 30 to 33: Similar to Page 5, why use such terminology? The "mid-year" mean definition for average of the monthly means also sound confusing.

Figures 2 through 9 and 12, a and b should be labeled.

Page 13 line 30: Define "S-Shape" here, as later in the paper, the authors mentioned it means Sigmoidal, and this should be defined before the "S-shape" terminology is used.

Page 21 line38: suggest revise to "because it is extremely volatile, therefore, it is difficult to trap and separate chromatographically"

Page 21 line 41: suggest revise to ". . .are only two of the many requirements. . ."

---

## Referee Comment (RC3) · Anonymous Referee #3 · 21 Sep 2018

While the overall goal of this paper is very worthwhile, there are many issues listed below by page (P), line (L) and Table numbers (especially the major points indicated) that need to be addressed before this paper is acceptable for OS.

P2 L20: MAJOR POINT-1. This is reasonable for all species for oceanic production, and for PFCs for oceanic destruction. But you provide no evidence that non-negligible destruction is ruled out for the HCFCs and HFCs that could be e.g. prone to hydrolysis.

[Figure]

Provide this evidence or state up front that the use of these species as ocean tracers depends on their verification as stable in ocean water.

P3 L5: P4: Please also cite relevant ALE/GAGE/AGAGE papers.

P3 L13: Please also cite relevant ALE/GAGE/AGAGE papers.

P3 L21: "They" not "He"

P4 L14: Need references for this doubling statement.

P5 L3-6: MAJOR POINT-2. But you need to calculate polar air concentrations (where oceanic down-welling maximizes) and not entire extratropical averages. This would best be done by assimilation of all station data into a 2D or 3D model, or at least by using the high latitude AGAGE and NOAA station data only (see also the later P6 L38 comment).

P5 L14-20: The case for differences from the prior Meinshausen et al, 2017 study would be strengthened by addressing MAJOR POINT-2.

P5 L21-25: Again, MAJOR POINT-1, what about potential in situ destruction of these H-containing species?

P5 L35: Change to Prinn, Weiss et al 2018a (new CDIAC website) and add Prinn, Weiss et al 2018b (references given at end of this review).

P5 L38-39: Much more relevant for making this point are the Prinn, Cunnold et al 1992 and Prinn, Weiss et al 2000 studies for the Samoa and Barbados sites showing the way ENSO and Atlantic Hurricanes enhance interhemispheric mixing and thus affect the measurements.at these stations.

P6 L9-10: Add Prinn, Weiss et al, 2018b reference for latest instrumentation.

P6 L10-11: Add Prinn, Weiss et al, 2018b (their Table 1) reference for precisions of ALL 8 of your compounds.

P6 L38 to P7 L2: MAJOR POINT-2 again. I do not understand why you are using only these AGAGE stations and NOAA sampling sites and neglecting AGAGE (e.g. Ny Alesund) and NOAA sites much closer to the polar down-welling regions? Also, MAJOR POINT-3, how are you weighting the ability of the NOAA (4 samples/month?) and AGAGE (900 samples/month?) measurements for computing monthly means? Surely the AGAGE monthly means are much more precise.

P8 L5: Please reference instead the main AGAGE website (agage.mit.edu) that connects to this daughter website only after potential users have read the substantial guidelines for ethical use of AGAGE data on the main website.

P8 L5-18: Some/many of these appear out of date. Check Prinn, Weiss et al, 2018b (their Section 2.6) for SIO-year calibrations, and their Table 5 for latest AGAGE/NOAA conversion factors.

P8-P9 Section 2.9 & Supplement S1: This old smoothing spline method is not very powerful compared to recent machine learning methods (e.g. Bodesheim et al, https://www.earth-syst-sci-data.net/10/1327/2018/). Also, the method appears to be using only the station/sample measurement precisions in computing the errors (see MAJOR POINT-3 about these). Also, MAJOR POINT-4, there is an additional error ("representation error") that takes into account that these point station measurements are not measuring the large volume of the surface atmosphere (extratropical, polar regions) that you are implicitly presuming that they do. In inverse and assimilation techniques these representation errors usually dominate the total measurement error except when mole fraction gradients are negligible (i.e. emissions are negligible). Thus, the errors you are reporting are lower limits to the real errors.

P22 L4-7: This conclusion suggesting "in a way that is optimized" is presently debatable given MAJOR POINTS 2,3,4.

Tables 2a-2h: Change Prinn et al, 2016 to Prinn, Weiss et al 2018a (new CDIAC website) and add Prinn, Weiss et al 2018b (ESSD paper).

Table 3: Some/many of these may be out of date. Check Prinn, Weiss et al, 2018b Section 2.6 for SIO (year) values and Table 5 for AGAGE/NOAA conversion factors.

References: Add Prinn, R.G., D.M. Cunnold, P.G. Simmonds, F.N. Alyea, R. Boldi, A. Crawford, P.J. Fraser, D. Gutzler, D.E. Hartley, R. Rosen, and R. Rasmussen, Global average concentration and trend for hydroxyl radicals deduced from ALE/GAGE trichloroethane (methyl chloroform) data for 1978–1990, J. Geophys. Res., 97, 2445–2461, 1992.

References: Replace Prinn et al 2016 old CDIAC website with this Prinn et al 2018a ESS-DIVE CDIAC website. Prinn, R. G., Weiss, R. F., Arduini, J., Arnold, T., Fraser, P. J., Ganesan, A. L., Gasore, J., Harth, C. M., Hermansen, O., Kim, J., Krummel, P. B., Li, S., Loh, Z. M., Lunder, C. R., Maione, M., Manning, A. J., Miller, B. R., Mitrevski, B., Mühle, J., O'Doherty, S., Park, S., Reimann, S., Rigby, M., Salameh, P. K., Schmidt, R., Simmonds, P. G., Steele, L. P., Vollmer, M. K., Wang, R. H., and Young, D.: The ALE/GAGE/AGAGE Network (DB 1001), http://cdiac.ess-dive.lbl.gov/ndps/alegage.html (https://doi.org/10.3334/CDIAC/atg.db1001), 2018a

References: Add Prinn, Weiss et al 2018b. Prinn, R. G., R. F. Weiss, J. Arduini, T. Arnold, H. L. DeWitt, P. J. Fraser, A. L. Ganesan, J. Gasore, C. M. Harth, O. Hermansen, J. Kim, P. B. Krummel, S. Li, Z. M. Loh, C. R. Lunder, M. Maione, A. J. Manning, B. R. Miller, B. Mitrevski, J. Mühle, S. O'Doherty, S. Park, S. Reimann, M. Rigby, T. Saito, P. K. Salameh, R. Schmidt,P. G. Simmonds, L. P. Steele, M. K. Vollmer, R. H. Wang, B. Yao, Y. Yokouchi, D. Young, and L. Zhou: History of chemically and radiatively important atmospheric gases from the Advanced Global Atmospheric Gases Experiment (AGAGE), Earth Syst. Sci. Data, 10, 985-1018, https://doi.org/10.5194/essd-10-985-2018, 2018b.

---

## Referee Comment (RC4) · Anonymous Referee #4 · 27 Sep 2018

Review of os-2018-89 (Li et al.)

The authors have provided two important components for the use of eight halogenated compounds as tracers of ocean circulation. They have reconstructed the atmospheric source functions and estimated their solubilities in water and seawater. These two items should enable reconstructions of the surface seawater concentrations of these compounds as a function of time. However the utility of these compounds as transient tracers depends more strongly on whether they are conservative in seawater, and whether they can be precisely and accurately measured on relatively small volumes of seawater in a timely fashion. A cautionary tale can be found in the use of CFC-113 as a tracer in the ocean. A great deal of time and effort went into developing the analytical method and determining its solubility. However CFC-113 exhibits non-conservative behavior in seawater. This paper needs to be revised before publication. There are a few scientific issues to be addressed and numerous grammatical corrections.

Scientific Issues:
Why not direct the reader to the AGAGE website (https://agage.mit.edu/data/agage-data) for the recent atmospheric measurements? Why were these eight compounds chosen out of the many compounds measured by AGAGE?

The authors provide late winter atmospheric concentrations for these compounds. Are these values much different then a linear interpolation between the mid-year values? Is there a systematic offset that makes it important to include the NH Feb and SH Aug concentrations?

Most of the use of transient tracers is to study the ventilation of waters colder than 10 C. However the measured freshwater solubilities presented in this manuscript do not constrain the solubility curves at temperatures colder than 298 K where the temperature dependence of the solubility becomes significant.

Editorial Comments:
P2,L10 – CFC concentrations are not "variables"
P2,L41 – Define MOZART
P3,L29 – Is it important that HFC-125 is the 5th most abundant HFC? It makes me wonder why the 2nd through 4th most abundant  HFCs are not considered by the authors.
P4,L31 – The atmospheric growth rate can be reversed – not the concentrations
P4,L40 – intercalibrated rather than consistent?
P6,L10 – "see studies"?
P6,L18 – be more specific than America
P6,L21 – all data for HFC134a are first reported for HFC134a?
P7,L34 – "by in"
P9, EQN1 – csapsGCV not defined
P10,EQN8 – D,E,F not defined

P10,L14 – "some properties"?

P10,L25 – There is no Vc in the equations

P11, L13 – Use approximately when dropping significant figures

P11,L16 – "so as to"?

P11,L30 – Note that CGW model method does not provide a method for estimating Lo

P12,L2 – Methods do not think

P12,L18 – A better description of the ventilation process is needed (e.g. the mixed layer deepens…)

P13,L4 – Curves and symbols in the upper panels of many figures are difficult to distinguish.

P13,L14 – "stable" is not the correct word

P13,L36 – As written "they " refers to the interhemispheric gradients

P14,L13 – Bimodal has a specific meaning, choose a more appropriate word for describing this curve.

P14,L15 – 'other one" implies a second plateau.

P14,L17 – verb missing

P14,L21 – "rapid" instead of "raid"

P14,L31 – "are" not "were"

P15,L8 – How do the authors distinguish "exponential" from "quadratic" from 'linear"?

P15,L10 – larger instead of "large"

P15,L13 – missing verb

P15,L35 – "reduced"

P16,L12-13 – sentence needs rewritten

P16,L22 – What does "annual growth rates exhibit a normal distribution" mean?

P17,L10 – "accelerated phase out" not correct

P17,L11 – "bank of HCFC-22 exist" ?

P17,L29 – "even out" is not a noun

P18,L18 – Warner and Weiss did use a salting out coefficient as defined in Eqn 6

P18,L24 – Fitting the solubility at temperatures between 25-65 C is not an important finding

P18,L29 and L31 – What happened to the factor of $10^{15}$?

P21,L2 – the Cl radical is the culprit, not the prime suspect

P21,L10 – Use e.g. before presenting a subset of the manuscripts that use the CFCs as ocean tracers

---

## Author Comment (AC1) · 29 Nov 2018

The comment was uploaded in the form of a supplement:
https://www.ocean-sci-discuss.net/os-2018-89/os-2018-89-AC1-supplement.pdf

---

## Author Response (AR1)

**Responses to referee comments on* "Global Annual Mean Atmospheric Histories, Growth Rates and Seawater Solubility Estimations of the Halogenated Compounds HCFC-22, HCFC-141b, HCFC-142b, HFC-134a, HFC-125, HFC-23, PFC-14 and PFC-116" *by* Pingyang Li et al.**

We greatly thank four anonymous referees for your constructive suggestions and comments. Below, we address all the comments and describe our responses to them where we refer to the revised manuscript (Italic) by listing the page and line numbers of changes. Simple comments are sorted in "TECHNICAL CORRECTIONS" and done as it said. Similar comments are sorted together to be response, especially for the comments from Anonymous Referee #3. In the revised manuscript, all changes from the original text are marked.

Additional, we added two co-authors, Ray F. Weiss and Paul J. Fraser, as they contributed some important data and helped to revise the paper very carefully. We also changed the corresponding author to Toste Tanhua (ttanhua@geomar.de) as Toste is my supervisor and my email address may not be used after one or two years.

**Anonymous Referee #1**

**"GENERAL COMMENTS"**

Comment: The authors synthesize the atmospheric concentration history and review the solubility of HCFC-22, HCFC-141b, HCFC-142b, HFC-134a, HFC-125, HFC-23, PFC-14 and PFC-116. This study is valuable as a first step of evaluating the utility of these compounds as oceanic transient tracers. Some revisions seem to be needed. In particular, the following two revisions are needed.

Response: We thank the Reviewer for the positive comments. Following are the responding responses to the comments.

Comment: First, estimate of Ostwald solubility coefficients by LFERs (Method II) should be revised. According to Abraham et al. (2001), the solvation parameter method of Abraham relies on two linear free energy relationships, LFERs, one for processes within condensed phases, Eq. (i), and one for processes involving gas to condensed phase transfer, Eq. (ii).

logSP = c + eE + sS + aA + bB + vV (i)

logSP = c + eE + sS + aA + bB + lL16 (ii)

Here L16 is the solute gas-hexadecane partition coefficient at 298 K; l is a coefficient for L16; and other coefficients and descriptors are the same as those described in the manuscript. Therefore, Ostwald solubility coefficients in pure water (L0) can be estimated by use of Eq. (ii) (processes involving gas to condense phase transfer), while salting-out coefficients (ks) can be estimated by use of Eq. (i) (processes within condensed phases). This point should be noted clearly in the manuscript.

In this study, as seen in Eqs. 10, 12 and 13 (Sect. 2.10.2), the relationship for water to solvent (processes within condensed phases, Eq. (i)) was used to estimate Ostwald solubility coefficient in pure water (L0). It should be revised as mentioned above. Furthermore, Tables S2 and S4 should be revised. When Eq. (ii) is used to estimate L0, may the discussion in Sect. 3.3 (page 20), which includes a comparison between Method II and the Revised Method II, lead to the same result as described in the manuscript?

Response: Thank you very much for your suggestions. I understand you think that only Eq. (ii) can be used to estimate the Ostwald solubility coefficients from the expression in Abraham et al. (2001). But both Eq. (i) and Eq. (ii) can be used to do the estimations from Abraham's more studies (Abraham et al., 1994; 2012). The abstract in Abraham et al. (1994) and Table 2 in Abraham et al. (2012) both described this. Actually, Abraham et al. (2001) also described this by reporting the water-gas partition coefficients of Eq. (i) in Table 1 and the gas-water partition

coefficients of Eq. (ii) in Table 2. In Table 1, the "solvent" in "water to solvent partition coefficients" could also be a gas phase.

Here we also explain this by the definition of gas-solvent partition coefficients. Gas-water,  $K_w$ , or gas-solvent,  $K_s$ , partition coefficients can be defined in terms of equilibrium concentrations of the solute, through Eq. (1).

$$K_s = \frac{\text{Conc. of solute in solvent, in mol dm}^{-3}}{\text{Conc. of solute in the gas phase, in mol dm}^{-3}}$$
(1)

The Ostwald solubility coefficients usually expressed as the gas-water partition coefficients. Based on the above definition, the values calculated from Eq. (i), water-gas partition coefficients, should be the reciprocal of the real solubility coefficients. But they are not. When Abraham dealt this in studies (Abraham et al., 1994; 2001; 2012), he already treated the *SP* in Eq. (i) as the Ostwald solubility coefficients. So now we know that both *SP* estimated from Eq. (i) and Eq. (ii) could be the Ostwald solubility coefficients.

In order to explain it more clearly in the revised manuscript, we added the Eq. (ii) to estimate the Ostwald solubility coefficients ( $L_0$ ) and compared the results to the ones calculated by equations based on V (Eq. (i)) and  $V_c$  (Table 4). In order to describe Section 2.7.2 more clearly, we also adjust the text.

This addition and comparison don't influence the final results as the Ostwald solubility coefficient estimated by the equation based on the  $V_c$  (the Revised Method II) for most compounds is the best method for the pp-LFERs model method. While compared with the pp-LFERs model method, the CGW model method for estimation of water solubility is the better one to be chosen for Combined Method (Table S4).

Comment: Second, hydrolysis of HCFC-22 should be taken into consideration in the discussion of transient tracer potential (for example, Sect. 3.4) because rate constants for hydrolysis of HCFC-22 in alkaline aqueous solutions are much larger than those for hydrolysis of other HCFCs and HFCs [for example, le Noble, W. J. Am. Chem. Soc., 87, 2434-2438 (1965); Kutsuna, S. et al. Int. J. Chem. Kinet. 43, 639-647 (2011)].

May hydrolysis of HCFC-22 in seawater make a significant influence on transient tracer potential of HCFC-22? If hydrolysis of HCFC-22 is significant, what would be expected when HCFC-22 is used as a transient tracer?

Response: Thank you very much for your question. We also aware of this issue.

le Noble (1965) studied the effects of hydrostatic pressure on the rate constants of the base-catalyzed and neutral hydrolysis of HCFC-22. The rate constants of hydrolysis decrease with increasing of pressure in water at 25 °C. When HCFC-22 was in the 0.06 *N* aqueous sodium hydroxide solutions, it was deduced that substrate decomposed at least 90 % to form ion. Kutsuna et al. (2011) concluded that the removal efficiency of HCFC-22 directly through the hydrolysis in alkaline aqueous is higher than previously expected temperature, such as 353 K.

When le Noble (1965) and Kutsuna et al. (2011) discussed the rate constants for hydrolysis of HCFC-22 in alkaline aqueous, the aqueous is 0.06 N and 0.25-7 mM aqueous NaOH solutions, that is, pH is 12.78 and in the range of 10.4-11.85, respectively. However, seawater pH is typically limited to a range between 7.5 and 8.4. Moreover, ocean acidification, an emerging global problem, is getting worse. The rate constants for hydrolysis of HCFC-22 in different concentrations of alkaline aqueous would be different.

As far as we know from previous studies, HCFCs, HFCs and PFCs are relatively stable in seawater as the ocean partial lifetimes (partial atmospheric lifetimes with respect to oceanic uptake) of HCFCs and HFCs ranged from thousands to millions of years (Table 1) and PFCs have atmospheric lifetimes on the order of thousands of years and very low solubility in seawater. However, the ocean partial lifetimes were estimated without considering the biological degradation of most of these compounds in the ocean as they have not been investigated. We know from the experience with CFC-113 and CCl4, for instance, that the half time of tracers in seawater can be lower than initially predicted. This manuscript is already very long, and we focus on the source of the tracers, i.e. the atmospheric history and the solubility, in this paper. We will discuss measurements (techniques, constraints, possibilities etc.) and sinks in a follow up paper that we are preparing right now. For instance, we will discuss the stability of these compounds by comparing to measurements of SF6 and CFC-12 from ocean samples collected during a few years from 3 different oceans.

Based on above discussion, "As shown in Table 1, the atmospheric lifetimes of HCFCs and HFCs with respect to hydrolysis in seawater are very long (Yvon-Lewis and Butler, 2002; Carpenter et al., 2014), ranging from thousands to millions of years, indicating that HCFCs and HFCs are relatively stable in the seawater. PFCs have atmospheric lifetimes on the order of thousands of years and very low solubilities in seawater." have been added at page 4, line 19-22.

**"SPECIFIC COMMENTS"**

Comment: Page 4, lines 21-22: Combustion in thermal power station has been pointed out as a tropospheric sink of PFCs [Ravishankara, A. R. et al., Science, 259, 194-199 (1993)]. It should be cited.

Response: "Combustion in thermal power stations has been pointed out as a tropospheric sink of PFCs (Cicerone, 1979; Ravishankara et al., 1993; Morris et al., 1995)" has been added at page 4, line 16-17.

Comment: Page 4, lines 28-30, "Cutting the production and consumption of HFCs by more than 80 % over the next 30 years under the Kigali amendment of the MP": Under the Kigali amendment, the amount of cutting the production and consumption of HFCs is based on the amount scaled by GWP of each HFC. This point had better be described.

Response: "Because of the high GWP of HFCs, 197 countries recently committed to cutting the production and consumption of HFCs by more than 80 % over the next 30 years under the Kigali amendment of the MP. Developed countries will reduce HFC consumption beginning in 2019." has been changed to

"Because of the high GWP of HFCs, 197 countries recently committed to cutting the production and consumption of HFCs by more than 80% over the next 30 years under the Kigali amendment of the MP, although not all of these countries have ratified this Amendment. The reductions in HFC production and consumption are based on GWP-weighted quantities. Developed countries that have ratified the Amendment have agreed to reduce HFC consumption beginning in 2019." at page 4, line 26-30.

Comment: Page 5, lines 17-18: Why were the same AGAGE calibration scales used by converting NOAA and UEA data to the AGAGE scale? Is the reason explained somewhere in the manuscript?

Response: NOAA and UEA data are converted to AGAGE scale as all target compounds are reported in the AGAGE network but only five and three target compounds are reported separately in the NOAA and the UEA network.

"Ambient air measurements published by the Advanced Global Atmospheric Gases Experiment (AGAGE), National Oceanic and Atmospheric Administration (NOAA) and University of East Anglia (UEA) are all considered in this study." has been changed to

"In order to provide a comprehensive and consistent view of halogenated compounds atmospheric distribution and changes over time, ambient air measurements published by the Advanced Global Atmospheric Gases Experiment (AGAGE), the Scripps Institution of Oceanography (SIO), the Commonwealth Scientific and Industrial Research Organization (CSIRO), the National Oceanic and Atmospheric Administration (NOAA) and University of East Anglia (UEA) are considered in this study. The calibration scale differences of these networks in the form of scale conversion factors are determined. SIO and CSIRO data are reported on AGAGE scales. NOAA and UEA data are converted to AGAGE scales by these conversion factors." at page 5, line 9-15.

Comment: Page 10, line 3, "The ionic strength of seawater (Iv, in g  $L^{-1}$ ): The unit of ionic strength should be checked.

Response: From the empirical equation (7), the ionic strength of a gas  $(I_v)$  in seawater is g L-1.

Comment: Page 10, Eq. (8): DS, ES and FS seems to need definition.

Response: Eq. (8) is the equation of state of seawater. We decide to delete it in the revised manuscript and only cite papers because the equation can be found in the oceanographic textbook. For more detail, see Millero and Poisson (1981).

$$\rho = \rho_0 + DS + ES^{3/2} + FS \tag{8}$$

Here  $\rho$  is the density of seawater in kg m-3, from 0 to 40 °C and from 0.5 to 43 salinity (S), have been used to determine a new 1-atm equation of state for seawater.

$$D = 8.24493 \times 10^{-1} - 4.0899 \times 10^{-3}t + 7.6438 \times 10^{-5}t^2 - 8.2467 \times 10^{-7}t^3 + 5.3875 \times 10^{-9}t^4$$
(9)

$$E = -5.72466 \times 10^{-3} + 1.0227 \times 10^{-4}t - 1.6546 \times 10^{-6}t^2$$
(10)

$$F = 4.8314 \times 10^{-4} \tag{11}$$

And  $\rho_0$  is the density of water (Bigg, 1967).

$$\rho_0 = 999.842594 + 6.793952 \times 10^{-2}t - 9.095290 \times 10^{-3}t^2 + 1.001685 \times 10^{-4}t^3$$
(12)
- 1.120083 \times 10^{-6}t^4 + 6.536336 \times 10^{-9}t^5

where t is the temperature in degree Celsius ( $^{\circ}C$ ).

Comment: Page 10, line 13, unit of the McGowan's characteristic molar volume in Eq. 8: According to Abraham et al. (2001), unit of the McGowan's characteristic molar volume is not  $\text{cm}^3 \text{ mol}^{-1}/100$  but  $\text{dm}^3 \text{ mol}^{-1}/100$ . The unit should be checked.

Response: According to the original studies (Abraham and McGowan, 1987; McGowan and Mellors, 1986) and other newer studies (Abraham et al., 2012; Abraham et al., 2004), units of the McGowan's characteristic molar volumes are  $\text{cm}^3 \text{ mol}^{-1}/100$ . It seems that the unit in Abraham et al. (2001) was wrongly written.

Comment: Page 12, line 34, Figures 2-9: The method to calculate IHG and its error (gray parts in Figures 2-9) seem to need explanation.

Response: "Inter-hemispheric gradients (IHG) are estimated from the annual mean atmospheric mole fractions of a gas in the NH minus the annual mean in the SH in the same year (Fig. 2a-h). Errors of the IHG were estimated based on error propagation of the annual means in the NH and the SH in the same year (Fig. 2a-h)." has been added at page 14, line 30-32.

In order to describe it more detailed, we also added the estimated method of uncertainties of hemispheric annual means in Sect. 2.6.

Comment: Page 14, line 11: What does it mean by "30% (median) larger"? It seems to need more explanation. Response: The atmospheric mole fractions for HCFC-142b in the NH divided into the ones in the SH for each year. A series of multiples are obtained. The median of the series of multiples minus one would be the percentage. It is 30 % for HCFC-142b based on the old method, but it is 6 % based on the new method. Following are discussed based on the old results. The whole sentence means that the annual mean atmospheric mole fractions for HCFC-142b in the NH are 30 % larger than those in the SH in the same year, and the 30 % is calculated based on the median value. The median may be thought of as the "middle" value for a data set. It can be used to express the distribution based on the positions of values rather than values themselves. For average, it is easy to be influenced by an extraordinarily high or low value.

For example, the annual mean atmospheric mole fractions for HCFC-142b in the NH are 30 % larger than those in the SH in the same year based on the median. When it is the average, the percentage goes up to 75 %. That is, the annual mean atmospheric mole fractions for HCFC-142b in the NH are 75 % larger than those in the SH in the same year. This result seems to be unreasonable.

In order not to be misunderstood, similar sentences were deleted in the revised manuscript.

Comment: Page 16, lines 27-29: Atmospheric lifetime of HFC-23 is much longer than that of HFC-134a. Hence, time-profiles are expected to be different between HFC-134a and HFC-23 after the consumption restrictions imposed by the 2016 Kigali Amendment to the Montreal Protocol. This point should be discussed.

Response: "Since the atmospheric lifetime of HFC-23 is much longer than that of HFC-134a, time-profiles and IHG change are expected to be a little different between HFC-134a and HFC-23." has been added at page 17, line 26-27.

Comment: Page 16, lines 34-35, "the growth rates in both hemispheres are really similar": Do the growth rates mean those of HFC-125 or some target compounds? What does it mean by "really similar"?

Response: The growth rates mean those of all target compounds. This sentence was deleted. Following sentences has been added at page 17, line 8-13.

"From Fig. 2a-h, it is clear that the mole fractions for target compounds in the NH are always larger than those in the SH but follow similar trends; the growth rates in both hemispheres are also similar, lagged in the SH, and the trends in IHG and in emissions/growth rates are very similar. This behavior is because the majority of the emissions (typically > 95%) occur in the NH extra-tropics (O'Doherty et al., 2009; Saikawa et al., 2012; Carpenter et al., 2014; UNEP, 2018) and the interhemispheric mixing time is around one or two years. Thus the larger (the increase) in emissions in the NH, the higher the resultant IHG."

Comment: Page 17, lines 28-29, "For compounds with shorter lifetimes. . .": This sentence is difficult to understand. It should be revised.

Response: In order not to be misunderstood, the sentence was deleted in the revised manuscript.

Comment: Page 18, lines 7-9: The experimental data for HFC-125 are scattered as seen in Fig. S5. This point had better be described more clearly.

**Response:** "For HFC-125, three fitted curves are shown in Fig. S2e, reflecting that data obtained by different methods do not agree with each other. Curve 1 includes data from Miguel et al. (2000), where the  $\phi$ - $\phi$  approach (the fugacity coefficient - fugacity coefficient method) has been used to predict the experimental results and the fugacity coefficients were calculated using a modified version of the Peng-Robinson equation of state, and Battino et al. (2011) where the data were collected from the International Union of Pure and Applied Chemistry (IUPAC) Solubility Data Series and, in some cases, as averages or estimates. Curve 2 includes data from Mclinden (1990) obtained from the vapour pressure of the pure substance divided by aqueous solubility (sometimes called VP/AS), and HSDB (2015) where the data were calculated with the quantitative structure–property relationship (QSPR) or a similar theoretical method. Curve 3 includes data from Reichl (1995) and Abraham et al. (2001)-observed, which are both measured values from original publications. Considering that the data which were based on measurements match with our results (Fig. S2e) calculated by Method II (only based on the physical properties of compounds), Curve 3 (the curve in the bottom) is chosen as the water solubility fit." has been added at page 18, line 24-34.

Comment: Page 18, lines 37-38: The dipolarity/polarizability (S) for C2F6 was estimated to be equal to the average of the S of CF4 and C3F8. This estimate might have substantial errors. Influence of errors of S on the salting-out coefficients of C2F6 should be evaluated.

Response: The general errors of *S*, *A*, *B* are thought to be 0.03 (Abraham et al., 1998; 2001). We assume that the error for each is 0.01 when *S*, *A* and *B* are all not zero and we assume that the error is 0.03 for *S* and 0 for *A* and *B*

when *S* is not zero but *A* and *B* are both zero. So the error of *S* would be both 0.03 for  $CF_4$  and  $C_3F_8$ . The error of *S* for  $C_2F_6$  is calculated to be 0.02 by the following equation:

$$\sigma_{C_2F_6} = \frac{1}{2} \cdot \sqrt{\sigma_{CF_4}^2 + \sigma_{C_3F_8}^2}$$

*"and the error for the estimate of S is estimated to be 0.02 based on the error propagation"* has been added at page 19, line 23-24.

Comment: Page 19, line 32: How is uncertainty from salting-out coefficients estimated? It seems to need more explanation.

Response: In the original manuscript, the descriptors E, S, A, B and V are thought to be constants shown in Table 4 without errors because they are always the same values in different temperatures. The uncertainties of salting-out coefficients were estimated based on the errors of coefficients c, e, s, a, b and v (Table 3 in the revised manuscript) when the temperature pluses or minuses 2 K. It was calculated by

$$\sigma_{K_s} = \sigma_c + E\sigma_e + S\sigma_s + A\sigma_a + B\sigma_b + V\sigma_v$$

Based on the propagation of uncertainty, the method above was wrong. The uncertainties of salting-out coefficients should be calculated by

$$\sigma_{K_s} = \sqrt{\sigma_c^2 + E^2 \sigma_e^2 + S^2 \sigma_s^2 + A^2 \sigma_a^2 + B^2 \sigma_b^2 + V^2 \sigma_v^2}$$

But in the revised manuscript, we would like to change the estimation method of the uncertainties of salting-out coefficients considering the errors of descriptors. Abraham et al. (2001) though that it is not easy to calculate the error in the descriptors as all the descriptors are calculated simultaneously. *E* is calculated without error. *V* is the McGowan's characteristic molar volume in cm3 mol-1/100 without error. The general errors of *S*, *A*, *B* are thought to be 0.03 (Abraham et al., 1998; Abraham et al., 2001). We assume that the error for each is 0.01 when *S*, *A* and *B* are all not zero. The uncertainty of salting-out coefficients could be calculated by

$$\sigma_{K_s} = \sqrt{\sigma_c^2 + E^2 \sigma_e^2 + s^2 S^2 [\left(\frac{\sigma_s}{s}\right)^2 + \left(\frac{\sigma_s}{s}\right)^2] + a^2 A^2 [\left(\frac{\sigma_a}{a}\right)^2 + \left(\frac{\sigma_A}{A}\right)^2] + b^2 B^2 [\left(\frac{\sigma_b}{b}\right)^2 + \left(\frac{\sigma_B}{B}\right)^2] + V^2 \sigma_v^2}$$

We assume that the error is 0.03 for S and 0 for A and B when S is not zero but A and B are both zero. The uncertainty of salting-out coefficients could be calculated by

$$\sigma_{K_s} = \sqrt{\sigma_c^2 + E^2 \sigma_e^2 + s^2 S^2 \left[\left(\frac{\sigma_s}{s}\right)^2 + \left(\frac{\sigma_s}{s}\right)^2\right] + A^2 \sigma_a^2 + B^2 \sigma_b^2 + V^2 \sigma_v^2}$$

"It is not easy to calculate the error in the descriptors as all the descriptors are calculated simultaneously (Abraham et al., 2001). E is calculated without error.  $V_c$  is the McGowan's characteristic molar volume without error. The general errors of S, A, B are thought to be 0.03 (Abraham et al., 1998; 2001). We assume that the error for each is 0.01 when S, A and B are all not zero and we assume that the error is 0.03 for S and 0 for A and B when S is not zero but A and B are both zero. So the uncertainties of salting-out coefficients could be calculated by error propagation based on different functions." has been added at page 13, line 13-18.

Comment: Supplement, Eq. (2): What does it mean by i = 0 in summation? Response: No meaning. We changed it to start with i=1.

"TECHNICAL CORRECTIONS" Comment: Page 5, line 21: "can used" is "can be used". Page 6, line 32: "fin air" is "firn air". Page 12, line 11: "By combining Eq. (4), (12) and (14)" may be "By combining Eqs. (4), (14) and (16)". Page 12, line 39 – page 13, line 1: "hemispheric" is "hemisphere". Page 13, line 21; page 13, line 39: "This consistent" is "This is consistent" Page 15, line 13: "These consistent" is "These are consistent" Page 16, line 17: " $0.038 \pm 0.007$ " is " $0.138 \pm 0.007$ ". Page 16, line 17: " $0.033 \pm 0.008$ " is " $0.133 \pm 0.008$ ". Page 16, line 34: "This could attributed" is "This could be attributed". Page 17, line 9: "This consistent" is "Chese are consistent". Page 17, line 9: "This consistent" is "Chese consistent". Page 18, line 9: "is chose" is "is chosen". Page 18, line 32: "298.15-338.15 K" is "273.15-313.15 K". Page 19, line 31: "9.0695e-05" is " $9.0695 \times 10-5$ ". Page 19, line 31: "9.1858e-05" is " $9.1858 \times 10-5$ ". Page 21, line 38: "can be also be" is "can also be". Response: All technical notes and suggestions have been fully implemented.

**Anonymous Referee #2**

Comment: This work the authors compile a large set of data to investigate the atmospheric history, growth rates, seawater solubility of a series of halocarbons. These are useful information to start understanding the suitability for these gases to be used as tracers. The paper is well-written, although I found that since a major motivation for the authors to investigate all these parameters was to assess the possibility to use these halocarbons as tracers, the title should be modified to reflect this. This paper presented an important initial step to assess the new generation of ocean tracers and should be published after minor revisions.

Response: We thank the Reviewer for the positive comments. The title is changed to be as "Atmospheric Histories, Growth Rates and Solubilities in Seawater and other Natural Waters of the Potential Transient Tracers HCFC-22, HCFC-141b, HCFC-142b, HFC-134a, HFC-125, HFC-23, PFC-14 and PFC-116".

Comment: Page 5 lines 3 to 5: the terminology of "mid-February" and "mid-August" used to define means from January to March and July to September is focusing. I suggest the authors use a different terminology for these.

Page 12 lines 30 to 33: Similar to Page 5, why use such terminology? The "mid-year" mean definition for average of the monthly means also sound confusing.

Response: The terminology of "mid-February", "mid-August" and "mid-year" changed to "JFM means", "JAS means" and "annual means".

**"TECHNICAL CORRECTIONS"**

**Comment:**

Page 6 line 33: should read "The firn air data. . ."

Page 12 line 23: should read "Based on this, we reconstructed. . ."

Figures 2 through 9 and 12, a and b should be labeled.

Page 13 line 30: Define "S-Shape" here, as later in the paper, the authors mentioned it means Sigmoidal, and this should be defined before the "S-shape" terminology is used.

Page 21 line38: suggest revise to "because it is extremely volatile, therefore, it is difficult to trap and separate chromatographically"

Page 21 line 41: suggest revise to "... are only two of the many requirements..."

Response: All technical notes and suggestions have been fully implemented.

**Anonymous Referee #3**

Comment: While the overall goal of this paper is very worthwhile, there are many issues listed below by page (P), line (L) and Table numbers (especially the major points indicated) that need to be addressed before this paper is acceptable for OS.

Response: We thank the Reviewer for the comments. Following are the responding responses to the comments.

Comment: P2 L20: MAJOR POINT-1. This is reasonable for all species for oceanic production, and for PFCs for oceanic destruction. But you provide no evidence that non-negligible destruction is ruled out for the HCFCs and HFCs that could be e.g. prone to hydrolysis. Provide this evidence or state up front that the use of these species as ocean tracers depends on their verification as stable in ocean water.

P5 L21-25: Again, MAJOR POINT-1, what about the potential in situ destructions of these H-containing species? Response: Thank you very much for your question. We also aware of this issue.

As far as we know from previous studies, HCFCs, HFCs and PFCs are relatively stable in seawater as the ocean partial lifetimes (partial atmospheric lifetimes with respect to oceanic uptake) of HCFCs and HFCs ranged from thousands to millions of years (Table 1) and PFCs have atmospheric lifetimes on the order of thousands of years and very low solubility in seawater. However, the ocean partial lifetimes were estimated without considering the biological degradation of most of these compounds in the ocean as they have not been investigated. We know from the experience with CFC-113 and CCl4, for instance, that the half time of tracers in seawater can be lower than initially predicted. This manuscript is already very long, and we focus on the source of the tracers, i.e. the atmospheric history and the solubility, in this paper. We will discuss measurements (techniques, constraints, possibilities etc.) and sinks in a follow up paper that we are preparing right now. For instance, we will discuss the stability of these compounds by comparing to measurements of SF6 and CFC-12 from ocean samples collected during a few years from 3 different oceans.

Based on above discussion, "As shown in Table 1, the atmospheric lifetimes of HCFCs and HFCs with respect to hydrolysis in seawater are very long (Yvon-Lewis and Butler, 2002; Carpenter et al., 2014), ranging from thousands to millions of years, indicating that HCFCs and HFCs are relatively stable in the seawater. PFCs have atmospheric lifetimes on the order of thousands of years and very low solubilities in seawater." have been added at page 4, line 19-22.

Comment: P5 L3-6: MAJOR POINT-2. But you need to calculate polar air concentrations (where oceanic down-welling maximizes) and not entire extratropical averages. This would best be done by assimilation of all station data into a 2D or 3D model, or at least by using the high latitude AGAGE and NOAA station data only (see also the later P6 L38 comment).

P5 L14-20: The case for differences from the prior Meinshausen et al, 2017 study would be strengthened by addressing MAJOR POINT-2.

P6 L38 to P7 L2: MAJOR POINT-2 again. I do not understand why you are using only these AGAGE stations and NOAA sampling sites and neglecting AGAGE (e.g. Ny Alesund) and NOAA sites much closer to the polar downwelling regions?

Response: The main purpose of the paper is to reconstruct the atmospheric histories of target compounds for their potential to be transient tracers. As Walker et al. (2000) and Bullister (2015) did for CFCs and  $CCl_4$ , we also used the data from MHD, THD and CGO to reconstruct the annual means atmospheric mole fractions in NH and SH for HCFCs, HFCs and PFCs.

Although it is very cold in polar areas, there is very little water formation in the polar regions; in the south there is a continent, and in the north there is strong stratification preventing deep water formation. In fact, most deep and

intermediate water formation takes place in the Labrador Sea, the Greenland Sea and in the Southern Ocean; at high latitudes, but not polar. Actually, the latitude of Mace Head is similar to the deep water formation region of the Labrador Sea.

From the selected sites for *in situ* or flask measurements, we reconstructed the atmospheric histories for the midlatitude area. But actually the archived air and firn air measurements and model results are from extra-tropical regions  $(30^{\circ}-90^{\circ})$ . The compiled atmospheric mole fraction histories of compounds in this paper could represent the ones for extra-tropical regions.

But if someone thought that it can only represent the atmospheric history of compounds in the mid-latitude area and he or she wants to do more research for other areas (such as polar area), the latitude gradients could be used from Meinshausen et al. (2017). For example, if someone would like to work exactly on the Antarctic Intermediate Water (AAIW), the Antarctic Bottom Water (AABW) and the Weddell Sea Bottom Water (WSBW) at around 60-75°S in the south, or the Greenland Sea Deep Water (GSDW) and the Labrador Sea Water (LSW) at around 60-75°N in the north, he or she could plus or minus values from latitude gradients shown in the Supplement figures from Meinshausen et al. (2017).

Actually, scientists also took into account the correct input function (atmospheric history) of the source regions when they use CFC-12 and  $SF_6$  as oceanic transient tracers (Stöven and Tanhua, 2014).

Comment: Also, MAJOR POINT-3, how are you weighting the ability of the NOAA (4 samples/month?) and AGAGE (900 samples/month?) measurements for computing monthly means? Surely the AGAGE monthly means are much more precise.

Response: In the original manuscript, the monthly average of an *in situ* data series with around 900 measurement points from the AGAGE got the same weight as the monthly average from a flask measurement program with few observations from NOAA.

In the revised manuscript, we combined the datasets to create monthly averages and standard deviations at the same hemisphere using the number of measurement-weighted averages when there were measurements from multiple different networks and sites.

We also updated our method as written in Sect. 2.6.

Comment: P8-P9 Section 2.9 & Supplement S1: This old smoothing spline method is not very powerful compared to recent machine learning methods (e.g. Bodesheim et al, https://www.earth-syst-sci-data.net/10/1327/2018/).

Response: Following your advice, we read the paper very carefully. Bodesheim et al. (2018) proposed the random forest regression model to predict the diurnal cycles in high-solution based on large-scale regression models. The predicted output can be obtained from the function:

$$y^* = \frac{1}{T} \sum_{t=1}^{T} \mathbf{y}_t^*$$

Inspired by this method, we found that we only have a decision tree. In order to avoid overfitting, the decision tree can be resampled (bootstrap aggregating or bagging) to obtain more trees and then we could do the regression. This method may be called Decision Tree Regression or Regression Tree Analysis. Following are the estimated results. We compared them with the results from our old method and a new method (Take atmospheric mole fractions of HCFC-22 in NH for example).

**Fig. 1** Comparison between the smoothing spline fit and the decision trees regression on the atmospheric mole fractions of HCFC-22 in NH based on the old data and new data. The difference between old data and new data is that the new data is the number of measurements-weighted monthly averages on the AGAGE *in situ* monthly means and the NOAA flask monthly mean in recent decades, and the polluted data from the NOAA flask measurements were removed in advance.

From Fig. 1, no matter for old data or new data, the total errors of estimations from decision trees regression are higher than the ones from smoothing spline fit. Based on the errors comparison, we prefer smooth spline fit.

About the method, we tried the polynomial fit, non-linear regression and smoothing spline. Following your advice, we tried decision trees regression. As our data are unevenly distributed, that is to say, we have more data with more precise in recent decades, while prior this time the data are less with less precise. For polynomial fit and non-linear regression with empirical functions, we could obtain good fit results in recent decades, while the predictions are really bad when there are fewer data. The same situation happens to decision trees regression. We thought the estimations from the three methods will be better if the original data could be more evenly distributed. Considering the purpose of the study, the smoothing spline fit works better for our estimation.

Comment: Also, the method appears to be using only the station/sample measurement precisions in computing the errors (see MAJOR POINT-3 about these). Also, MAJOR POINT-4, there is an additional error ("representation error") that takes into account that these point station measurements are not measuring the large volume of the

surface atmosphere (extratropical, polar regions) that you are implicitly presuming that they do. In inverse and assimilation techniques these representation errors usually dominate the total measurement error except when mole fraction gradients are negligible (i.e. emissions are negligible). Thus, the errors you are reporting are lower limits to the real errors.

Response: The errors are re-calculated as written in Sect. 2.6.

The error for latitude gradients is not easy to estimate. If we think about the errors in latitude gradients based on the study (Meinshausen et al., 2017), the errors will be really significant.

Comment: P22 L4-7: This conclusion suggesting "in a way that is optimized" is presently debatable given MAJOR POINTS 2,3,4.

Response: It is not optimized. The hemispheric annual means were calculated to be the source functions of transient tracer studies rather than the real hemispheric means based on all the atmospheric observations themselves.

Comment: P8 L5-18: Some/many of these appear out of date. Check Prinn, Weiss et al, 2018b (their Section 2.6) for SIO-year calibrations, and their Table 5 for latest AGAGE/NOAA conversion factors.

Table 3: Some/many of these may be out of date. Check Prinn, Weiss et al, 2018b Section 2.6 for SIO (year) values and Table 5 for AGAGE/NOAA conversion factors.

Response: The AGAGE/NOAA conversion factors have been changed for Section 2.5 and Table 2.

**"TECHNICAL CORRECTIONS"**

Comment:

P3 L5: P4: Please also cite relevant ALE/GAGE/AGAGE papers.

P3 L13: Please also cite relevant ALE/GAGE/AGAGE papers.

P3 L21: "They" not "He"

P4 L14: Need references for this doubling statement.

P5 L35: Change to Prinn, Weiss et al 2018a (new CDIAC website) and add Prinn, Weiss et al 2018b (references are given at end of this review).

P5 L38-39: Much more relevant for making this point are the Prinn, Cunnold et al 1992 and Prinn, Weiss et al 2000 studies for the Samoa and Barbados sites showing the way ENSO and Atlantic Hurricanes enhance interhemispheric mixing and thus affect the measurements.at these stations.

P6 L9-10: Add Prinn, Weiss et al, 2018b reference for latest instrumentation.

P6 L10-11: Add Prinn, Weiss et al, 2018b (their Table 1) reference for precisions of ALL 8 of your compounds.

P8 L5: Please reference instead of the main AGAGE website (agage.mit.edu) that connects to this daughter website only after potential users have read the substantial guidelines for ethical use of AGAGE data on the main website.

Tables 2a-2h: Change Prinn et al, 2016 to Prinn, Weiss et al 2018a (new CDIAC website) and add Prinn, Weiss et al 2018b (ESSD paper).

Response: All technical notes and suggestions have been fully implemented.

**Anonymous Referee #4**

Comment: The authors have provided two important components for the use of eight halogenated compounds as tracers of ocean circulation. They have reconstructed the atmospheric source functions and estimated their solubilities in water and seawater. These two items should enable reconstructions of the surface seawater concentrations of these compounds as a function of time. However the utility of these compounds as transient tracers depends more strongly on whether they are conservative in seawater, and whether they can be precisely and

accurately measured on relatively small volumes of seawater in a timely fashion. A cautionary tale can be found in the use of CFC-113 as a tracer in the ocean. A great deal of time and effort went into developing the analytical method and determining its solubility. However CFC-113 exhibits non-conservative behavior in seawater. This paper needs to be revised before publication. There are a few scientific issues to be addressed and numerous grammatical corrections.

Response: We thank the Reviewer for the comments. As far as we know from previous studies, HCFCs, HFCs and PFCs are relatively stable in seawater as the ocean partial lifetimes (partial atmospheric lifetimes with respect to oceanic uptake) of HCFCs and HFCs ranged from thousands to millions of years (Table 1) and PFCs have atmospheric lifetimes on the order of thousands of years and very low solubility in seawater. However, the ocean partial lifetimes were estimated without considering the biological degradation of most of these compounds in the ocean as they have not been investigated. We know from the experience with CFC-113 and CCl4, for instance, that the half time of tracers in seawater can be lower than initially predicted. This manuscript is already very long, and we focus on the source of the tracers, i.e. the atmospheric history and the solubility, in this paper. We will discuss measurements (techniques, constraints, possibilities etc.) and sinks in a follow up paper that we are preparing right now. For instance, we will discuss the stability of these compounds by comparing to measurements of SF6 and CFC-12 from ocean samples collected during a few years from 3 different oceans.

Based on above discussion, "As shown in Table 1, the atmospheric lifetimes of HCFCs and HFCs with respect to hydrolysis in seawater are very long (Yvon-Lewis and Butler, 2002; Carpenter et al., 2014), ranging from thousands to millions of years, indicating that HCFCs and HFCs are relatively stable in the seawater. PFCs have atmospheric lifetimes on the order of thousands of years and very low solubilities in seawater." have been added at page 4, line 19-22.

**Scientific Issues:**

Comment: Why not direct the reader to the AGAGE website (https://agage.mit.edu/data/agage-data) for the recent atmospheric measurements?

Response: P6, L6-7, "where historic and newest atmospheric measurements are reported" has been added behind "The data are available at the AGAGE website (http://agage.eas.gatech.edu/data\_archive/)".

Comment: Why were these eight compounds chosen out of the many compounds measured by AGAGE?

Response: The atmospheric mole fractions of CFCs, HCFCs, HFCs, PFCs, Halons, very shorted-lived compounds (-Cl, -Br, -I) are reported by the AGAGE website (https://agage.mit.edu/data/agage-data). CFCs have been studied for transient tracers since the 1980s. Halons are not suitable for tracer studies because their atmospheric mole fractions are decreasing obviously. The very shorted-lived compounds (-Cl, -Br, -I) are also not proper to be alternative tracers because their lifetimes are too short and they are not stable in seawater apparently. Thus HCFCs, HFCs and PFCs are chosen as the potential transient tracers. Besides, the atmospheric mole fractions of most HCFCs, HFCs and PFCs are increasing in the atmosphere.

For HCFCs, the atmospheric mole fractions of only HCFC-22, HCFC-141b and HCFC-142b are reported. For HFCs, HFC-134a is the most abundant HFC in the atmosphere. The trends of atmospheric mole fractions of HFC-125 and HFC-32 are different from other HFCs. The lifetime of HFC-125 (31 years) is longer than HFC-32 (4.9 years) in the atmosphere. HFC-23 is chosen because it has a relatively long lifetime in the atmosphere (228 years). For PFCs, they have very long lifetimes in the atmosphere and are very stable in seawater. The concentrations of PFCs are not easy to be detected considering their low solubility in seawater. Therefore, only PFC-14 and PFC-116 with relatively high atmospheric mole fractions are chosen to be studied.

Comment: The authors provide late winter atmospheric concentrations for these compounds. Are these values much different than a linear interpolation between the mid-year values? Is there a systematic offset that makes it important to include the NH Feb and SH Aug concentrations?

Response: The annual means were calculated by averaging the monthly means of the corresponding 12 months. The JFM and JAS means are obtained by averaging the monthly means of January, February and March and the monthly means of July, August and September of the same year. The median of relative standard deviation (RSD) for JFM between calculated means and linear interpolated ones are 0.15%, 0.59%, 0.20%, 0.47%, 1.96%, 0.20%, 0.006% and 0.06% for HCFC-22, HCFC-141b, HCFC-142b, HFC-134a, HFC-125, HFC-23, PFC-14 and PFC-116. The median of RSD for JAS between calculated means and linear interpolated ones are 0.22%, 0.44%, 0.05%, 0.40%, 1.03%, 0.05%, 0.002% and 0.03% for HCFC-22, HCFC-141b, HCFC-142b, HFC-134a, HFC-134a, HFC-125, HFC-23, PFC-14 and PFC-116. These values are not much different from a linear interpolation between the annual means, but these calculated values are more precise than the interpolated ones.

Comment: Most of the use of transient tracers is to study the ventilation of waters colder than 10 C. However the measured freshwater solubility presented in this manuscript do not constrain the solubility curves at temperatures colder than 298 K where the temperature dependence of the solubility becomes significant.

Response: As shown in Table 5, the measured freshwater solubility presented in this manuscript constrain the solubility curves at temperatures 5-80 °C for HCFC-22, HCFC-141b, HCFC-142b, HFC-134a; 10-70 °C for HFC-125; 5-75 °C for HFC-23; 0-55 °C for PFC-14; 5-55 °C for PFC-116 and 0-75 °C for CFC-12. Therefore, except HFC-125 and PFC-116, the measured freshwater solubility of other compounds constrains the solubility curves at temperatures colder than 10 °C. Moreover, the measured freshwater solubility of HFC-125 and PFC-125 and PFC-116 also constrain the solubility curves at temperatures between 10 °C and 25 °C.

Editorial Comments: Comment: P2,L10 – CFC concentrations are not "variables" Response: It has been changed to "indices".

Comment: P2, L41 – Define MOZART Response: "MOZART model" has changed to "*the Model for OZone And Related chemical Tracers (MOZART)*".

Comment: P3, L29 – Is it important that HFC-125 is the 5th most abundant HFC? It makes me wonder why the 2nd through 4th most abundant HFCs are not considered by the authors.

Response: It is not important that HFC-125 is the 5th (changed to third after re-check) most abundant HFC. The reason why HFC-125 is chosen is that the trends of atmospheric mole fractions of HFC-125 and HFC-32 are different from other HFCs and the lifetime of HFC-125 (31 years) is longer than HFC-32 (4.9 years) in the atmosphere.

Comment: P4, L31 – The atmospheric growth rate can be reversed – not the concentrations Response: "reverse" has changed to "decline".

Comment: P4, L40 – intercalibrated rather than consistent? Response: It is "consistent".

Comment: P6, L10 – "see studies"? Response: It has been changed to "see Prinn et al. (2018a)". Comment: P6, L18 – be more specific than America

Response: "Northern Hemisphere (NH) samples used for this paper were filled during background conditions mostly at Trinidad Head, but also at La Jolla, California, Cape Meares, Oregon (courtesy of the Oregon Graduate Centre via CSIRO, Aspendale, and the Norwegian Institute for Air Research, Oslo, Norway), and Point Barrow, Alaska (courtesy of Robert Rhew, University of California, Berkeley)" has been added to replace "America" in P6, L31-34.

Comment: P6, L21 – all data for HFC134a are first reported for HFC134a? Response: AGAGE HFC-134a air archive data are reported here for the first time.

Comment: P7, L34 – "by in" Response: "in" has been deleted.

Comment: P9, EQN1 – csapsGCV not defined Response: "csapsGCV" has been deleted.

**Comment: P10,EQN8 - D,E,F not defined**

Response: Eq. (8) is the equation of state of seawater. We decide to delete it in the revised manuscript and only cite papers because the equation can be found in the oceanographic textbook. For more detail, see Millero and Poisson (1981).

$$\rho = \rho_0 + DS + ES^{3/2} + FS \tag{8}$$

Here  $\rho$  is the density of seawater in kg m-3, from 0 to 40 °C and from 0.5 to 43 salinity (S), have been used to determine a new 1-atm equation of state for seawater.

$$D = 8.24493 \times 10^{-1} - 4.0899 \times 10^{-3}t + 7.6438 \times 10^{-5}t^2 - 8.2467 \times 10^{-7}t^3 + 5.3875 \times 10^{-9}t^4$$
(9)

$$E = -5.72466 \times 10^{-3} + 1.0227 \times 10^{-4}t - 1.6546 \times 10^{-6}t^2$$
(10)

$$F = 4.8314 \times 10^{-4} \tag{11}$$

And  $\rho_0$  is the density of water (Bigg, 1967).

$$\rho_0 = 999.842594 + 6.793952 \times 10^{-2}t - 9.095290 \times 10^{-3}t^2 + 1.001685 \times 10^{-4}t^3$$
(12)
- 1.120083 \times 10^{-6}t^4 + 6.536336 \times 10^{-9}t^5

where t is the temperature in degree Celsius ( $^{\circ}C$ ).

**Comment: P10, L14 – "some properties"?**

Response: Abraham (1993) presented a paragraph as "The dependent variable log SP refers to some property of a series of solutes in a fixed phase (or phases) Thus SP could be L, the gas-liquid partition coefficient for a series of solutes in a given liquid or it could be P, the partition coefficient for a series of solutes between water and, say, octanol. In the case of biological properties, where SP can be some biological response as an  $LC_{50}$ , equations 13 and 14 then represent two new families of quantitative structure-activity relationships (QSARs)."

It should be "some property". The "some property" could be the gas-liquid partition coefficient, the liquid-liquid partition coefficient and  $LC_{50}$  etc. Considering the comments from Reviewer #1, we revised the text to be "In these equations, the dependent variable logSP is some property of a series of solutes in a given system. Therefore, SP could be partition coefficient, P, for a series of solutes in a given water-solvent system in Eq. (9), or L, for a series of solutes in a given gas-solvent system in Eq. (10)." at P12, L1-3.

**Comment: P10, L25 – There is no Vc in the equations**

Response: Considering the comments from Reviewer #1, we revised the text. Please see Sect. 2.7.2 in the revised manuscript.

Comment: P11, L13 - Use approximately when dropping significant figures

Response: "The salt concentration in seawater is equivalent to 0.6 M NaCl" has been changed to "The salt concentration in seawater is approximately equivalent to 0.6 M NaCl".

Comment: P11, L16 – "so as to"? Response: "so as to" has been changed to "to".

Comment: P11, L30 – Note that CGW model method does not provide a method for estimating Lo Response: When the salinity (*S*) is zero, the CGW model provides a method for estimating freshwater solubility ( $L_0$ ).

**Comment: P12, L2 – Methods do not think**

Response: "Method II thinks more about the physical properties of compounds." has been changed to "Consideration of the physical properties of compounds is more important in Method II" at P13, L34-35.

Comment: P12, L18 – A better description of the ventilation process is needed (e.g. the mixed layer deepens...) Response: "During this process, the mixed layer deepens and older water (usually with lower transient tracer mole fractions) is brought in contact with the atmosphere. The mixed layer is gaining density and tends to be transported towards the ocean interior through diffusive, advective and/or convective processes. This water then carries with it a signature of the atmospheric mole fraction, pending the saturation state of the water as it leaves the surface layer. For tracers with rapidly increasing atmospheric mole fractions and for deep mixed layers under saturation of the tracers has frequently been reported (e.g. Tanhua et al. (2008b))" is added at P14, L10-14 in Sect. 3.1.

Comment: P13, L4 – Curves and symbols in the upper panels of many figures are difficult to distinguish. Response: Fig. 2a-9a were moved to the Supplement and renamed as Fig. S1 (a-h). Fig. S1 (ai-hi) show the original data and estimated annual means. Fig. S1 (aii-hii) shows the enlarged views of Fig. S1 (ai-hi) for the recent 5-6 years.

**Comment: P13, L14 - "stable" is not the correct word**

Response: "This initial increase and then stable in the growth rate of HCFC-22 coincide with the large production and consumption reported for between the 1950s and 1990s (Fig. 1) and a freeze of production magnitudes in the developed countries in 1996." has been changed to

"Growth rates for HCFC-22 rose steadily until 1990, followed by a slight decrease, which coincides with the large production and consumption reported between the 1950s and 1990s (Fig. 1) and a freeze of production magnitudes in the developed countries in 1996." at P14, L36-38.

Comment: P13, L36 – As written "they " refers to the interhemispheric gradients Response: No. "They" is "The annual growth rates". This sentence has been deleted.

Comment: P14, L13 – Bimodal has a specific meaning, choose a more appropriate word for describing this curve. Response: This sentence has been deleted.

Comment: P14, L15 – 'other one" implies a second plateau.

Response: Yes. The growth rates show a double-peak distribution pattern. The first plateau ..., the growth rates of the other one ....

In order not to misunderstand, this sentence has been deleted.

Comment: P14, L17 – verb missing Response: "are" has been added.

Comment: P15, L8 - How do the authors distinguish "exponential" from "quadratic" from 'linear"?

Response: The three ones are distinguished by curve fit. P values were compared. The higher P value, the better fit.

Comment: P15, L13 – missing verb Response: "are" has been added.

Comment: P16, L12-13 – sentence needs rewritten Response: This sentence has been deleted.

Comment: P16, L22 – What does "annual growth rates exhibit a normal distribution" mean? Response: "annual growth rates exhibit a normal distribution" has been changed to "annual growth rates exhibit the shape of Gaussian distribution" at P17, L19.

Comment: P17, L10 – "accelerated phase out" not correct Response: "accelerated phase out" has been changed to "phased-out sooner than originally mandated" at P18, L5.

Comment: P17, L11 – "bank of HCFC-22 exists" ? Response: "bank" has been changed to "emission source" at P18, L9-10.

Comment: P17, L29 – "even out" is not a noun Response: In order not to misunderstand, this sentence has been deleted.

Comment: P18, L18 – Warner and Weiss did use a salting out coefficient as defined in Eqn 6 Response: No, they didn't. Eq. (6) is from the method of Deeds et al. (2008), not from the method of Warner and Weiss (1985). Only Eq. (3) is from the method of Warner and Weiss (1985).

Comment: P18, L24 – Fitting the solubility at temperatures between 25-65 C is not an important finding Response: Sorry to wrote it wrong. It has been changed to 0-40 °C.

Comment: P18, L29 and L31 – What happened to the factor of 10^15? Response: In the study of Warner and Weiss (1985), the average of  $K_s$  is  $0.229 \pm (1.41 \cdot 10^{-15})$  L g-1 at 298.15 K when the salinity is in the range of 0-40. It means that  $K_s$  is independent of salinity.

Comment: P21, L2 – the Cl radical is the culprit, not the prime suspect Response: "prime suspect" has been changed to "culprit". In order to shorten the paper, we delete most of sentences in this paragraph.

**"TECHNICAL CORRECTIONS"**

Comment:

P14, L21 - "rapid" instead of "raid"

P14, L31 - "are" not "were"

P15, L10 – larger instead of "large"

P15, L35 - "reduced"

P21, L10 - Use e.g. before presenting a subset of the manuscripts that use the CFCs as ocean tracers

Response: All technical notes and suggestions have been fully implemented.

**Additional Corrections**

For tables:

Original Table 2 (collected data) was moved to the Supplement (Table S1) to shorten the manuscript. Original Table 3 (primary scales) was renamed to be the new Table 2.

New Table 3 was added to sort the coefficients and descriptors of partition coefficients for Sect. 2.7.2. Position of Table 4 and Table 5 was exchanged.

Original Table S1 (atmospheric mole fractions) in the Supplement was renamed as Table S2.

Original Table S2 was renamed to be Table S4 (comparison of solubility calculated by V,  $V_c$  and  $logL^{16}$ ).

Original Table S4 was renamed to be Table S5 (Ostwald solubility functions estimated by Method II).

**For figures:**

To response one comment from Anonymous Referee #4, Fig. 2a-9a were moved to the Supplement and renamed as Fig. S1 (a-h). Fig. S1 (ai-hi) show the original data and estimated annual means. Fig. S1 (aii-hii) shows the enlarged views of Fig. S1 (ai-hi) for the recent 5-6 years.

Original Figure 2b-9b was renamed as Fig. 2 (a-h).

Original Figure 10-12 was renamed as Fig. 3-5.

Original Figure S1-S9 was renamed as Fig. S2 (a-i).

Original Figure S10 was renamed as Fig. S3.

Content of Tables and Figures for the Supplement was added.

For manuscript:

The titles of sections have been changed as follows:

|                                                                      | 4 1 Introduction                                                  |
|----------------------------------------------------------------------|-------------------------------------------------------------------|
|                                                                      | 1.1 Potential transient tracers                                   |
|                                                                      | 1.2 Production and ban histories                                  |
|                                                                      | 2 Data and Methods                                                |
|                                                                      | 4 2.1 Data from the AGAGE network                                 |
|                                                                      | 2.1.1 AGAGE in situ measurements and instrumentation              |
|                                                                      | 2.1.2 AGAGE measurements of CSIRO and SIO archived air            |
|                                                                      | 2.1.3 AGAGE measurements of CSIRO firn air                        |
|                                                                      | 4 2.2 Data from the NOAA network                                  |
|                                                                      | 2.2.1 NOAA flask measurements                                     |
| 1 Introduction                                                       | 2.2.2 NOAA measurements of archived and shipborne air samples     |
| 4 2 Data and Methods                                                 | 2.2.3 NOAA measurements of firn air                               |
| 2.1 AGAGE in situ Measurements and Instrumentation                   | 2.3 Data from the UEA network                                     |
| 2.2 AGAGE archived air                                               | 2.3.1 UEA archived air                                            |
| 2.3 AGAGE firn air                                                   | 2.4 Data from models                                              |
| 2.4 NOAA flask measurements                                          | 2.5 AGAGE, NOAA and UEA calibration scales                        |
| 2.5 NOAA archived air                                                | 2.6 Hemispheric annual means and uncertainties estimation         |
| 2.6 UEA archived air                                                 | 2.7 Seawater solubility estimation method                         |
| 2.7 Models                                                           | 2.7.1 Method I: the CGW model                                     |
| 2.8 Calibration scale                                                | 2.7.2 Method II: the pp-LFER model                                |
| 2.9 Smoothing spline fit and uncertainty                             | 2.7.3 Combined Method: combined CGW model and pp-LFER model       |
| 2.10 Seawater solubility estimation method                           | A 3 Results and Discussion                                        |
| 2.10.1 Method I: the CGW model                                       | 3.1 Atmospheric histories and growth rates                        |
| 2.10.2 Method II: the pp-LFER model                                  | 3.1.1 HCFC-22                                                     |
| 2.10.3 Combined Method: combined the CGW model and the pp-LFER model | 3.1.2 HCFC-141b                                                   |
| 3 Results and Discussion                                             | 3.1.3 HCFC-142b                                                   |
| 3.1 Atmospheric histories and growth rates                           | 3.1.4 HEC.134a                                                    |
| 3.1.1 HCFC-22                                                        | 31.5 HEC.125                                                      |
| 3.1.2 HCFC-141b                                                      | 31.6 HEC.23                                                       |
| 3.1.3 HCFC-142b                                                      | 31.7 PEC.14 (CE4)                                                 |
| 3.1.4 HFC-134a                                                       | 31.8 PEC-116                                                      |
| 3.1.5 HFC-125                                                        | 3.2 Growth Patterns                                               |
| 3.1.6 HFC-23                                                         | 4 3 3 Solubility in seawater                                      |
| 3.1.7 PFC-14                                                         | 3 3 1 Solubility in freshwater                                    |
| 3.1.8 PFC-116                                                        | 3.3.2 Salting out coefficient                                     |
| 3.2 Growth Patterns                                                  | 2.2.2 Solubility in conveter based on Combined Method             |
| 3.3 Solubility in seawater                                           | 2.2.4 Comparison of colubility in seawater based on three methods |
| 3.4 Transient Tracer potential and comparison with CFC-12            | 2.4 Transient Tracer potential and comparison with CEC 12         |
| 4 Conclusions                                                        | A Conclusions                                                     |
| Acknowledgments                                                      | 4 conclusions                                                     |
| References                                                           | Acknowledgments                                                   |
| Original sections titles                                             | Revised sections titles                                           |

This work can be applied for more waters. "Even though our intention is for application in oceanic research, the work described in this paper is potentially useful for tracer studies in a wide range of natural waters, including freshwater and saline lakes, and, for the more stable compounds, groundwaters." has been added at page 1, line 40-42.

The manuscript was shorten by deleted some background information in Introduction (Sect. 1), some very detailed description of atmospheric histories and growth rates of target compounds (Sect. 3.1), the last two paragraphs in Sect. 3.2, and most sentences in the first paragraph in Sect. 3.4 (background information of production and use history of CFC-12). The manuscript was checked and revised very carefully again. For detail, see following revised paper.

**Atmospheric Histories, Growth Rates and Solubilities in Seawater and other Natural Waters of the Potential Transient Tracers HCFC-22, HCFC-141b, HCFC-142b, HFC-134a, HFC-125, HFC-23, PFC-14 and PFC-116**

Pingyang Li1, Jens Mühle2, Stephen A. Montzka3, David E. Oram4, Benjamin R. Miller3, Ray F. Weiss2, Paul J. Fraser5 and Toste Tanhua1

1GEOMAR Helmholtz Centre for Ocean Research Kiel, Kiel, 24105, Germany 2 Scripps Institution of Oceanography, University of California, San Diego, La Jolla, California, 92093, USA 3Earth System Research Laboratory, National Oceanic and Atmospheric Administration, Boulder, Colorado, 80305, USA 4National Centre for Atmospheric Science, Centre for Ocean and Atmospheric Sciences, School of Environmental Sciences, University of East Anglia, Norwich, NR4 7TJ, UK 5Climate Science Centre, Commonwealth Scientific and Industrial Research Organization Oceans and Atmosphere, As-

5Climate Science Centre, Commonwealth Scientific and Industrial Research Organization Oceans and Atmosphere, Aspendale, Victoria, 3195, Australia

15 *Correspondence to*: Toste Tanhua (ttanhua@geomar.de)

5

10

20

Abstract. We present consistent annual mean atmospheric histories and growth rates for the mainly anthropogenic halogenated compounds HCFC-22, HCFC-141b, HCFC-142b, HFC-134a, HFC-125, HFC-23, PFC-14 and PFC-116, all potentially useful oceanic transient tracers (tracers of water transport within the ocean), for the Northern and Southern Hemispheres with the aim of providing input histories of these compounds for the equilibrium between the atmosphere and surface ocean. We use observations of these halogenated compounds made by the Advanced Global Atmospheric Gases Experiment (AGAGE), the Scripps Institution of Oceanography (SIO), the Commonwealth Scientific and Industrial Research Organization (CSIRO), the National Oceanic and Atmospheric Administration (NOAA), and the University of East Anglia (UEA). Prior to the direct observational record, we use archived air measurements, firn air measurements and published model calculations to estimate the atmospheric mole fraction histories. The results show that the atmospheric mole fractions for each

- 25 species, except HCFC-141b and HCFC-142b, have been increasing since they were initially produced. Recently, the atmospheric growth rates are decreasing for the HCFCs (HCFC-22, HCFC-141b and HCFC-142b), are increasing for the HFCs (HFC-134a, HFC-125, HFC-23), and are stable with small fluctuation for the PFCs (PFC-14 and PFC-116) investigated here. The atmospheric histories (source functions) and natural background mole fractions show that HCFC-22, s141b and s142b, and HFC-134a, s125 and s23 have the potential to be oceanic transient tracers for the next few decades only because of the
- 30 recently imposed bans on productionand consumption. When the atmospheric histories of the compounds are not monotonically changing, the equilibrium atmospheric mole fraction (and ultimately the age associated with that mole fraction) calculated from their concentration in the ocean are not unique, reducing their potential as transient tracer. Moreover, HFCs have potential to be oceanic transient tracers for a longer period in the future than HCFCs as the growth rates of HFCs are increasing and those of HCFCs are decreasing in the background atmosphere. PFC-14 and PFC-116, however, have the potential to
- 35 be the tracers for longer periods into the future due to their extremely long lifetimes, steady atmospheric growth rates and no explicit banon their emissions. In this work, we also derive solubility functions for HCFC-22, HCFC-141b, HCFC-142b, HFC-134a, HFC-125, HFC-23, PFC-14 and PFC-116 in water and seawater to facilitate their use as oceanic transient tracers. These functions are based on the Clark-Glew-Weiss (CGW) water solubility functions fit and salting-out coefficients estimated by the poly-parameter linear free energy relationships (pp-LFERs). Here we also provide three methods of seawater
- 40 solubility estimation for more compounds. Even though our intention is for application in oceanic research, the work described in this paper is potentially useful for tracer studies in a wide range of natural waters, including freshwater and saline lakes, and, for the more stable compounds, groundwaters.

| Deleted: for                                                                                |
|---------------------------------------------------------------------------------------------|
| Deleted: here available w                                                            |
| Deleted: utilise                                                                            |
| Deleted: and                                                                                |
| Deleted: estimated the atmospheric history concentrations from other sources such as |
| Deleted: ir                                                                                 |

| Deleted: values        |
|------------------------|
| Deleted: HCFCs (       |
| Deleted: HCFC          |
| Deleted: HCFC          |
| Deleted: )             |
| Deleted: HFCs (        |
| Deleted: HFC           |
| Deleted: HFC           |
| Deleted: )             |
| Deleted: concentration |
| Deleted: s             |
| Deleted: concentration |
| Deleted: thanks        |
|                        |

**1** Introduction**

Oceanic and natural waters transient tracers have time varying sources and/or sinks. Chlorofluorocarbons (CFCs) have been used traditionally as oceanographic transient tracers because of their continuously increasing atmospheric mole fractions until some years ago. They are powerful tools in oceanography where they are used to, for instance, deduce transport times,

- stimate mixing rates between water masses, study formation rates of new water masses, and determine the anthropogenic carbon (Cant) content of seawater (Weiss et al., 1985; Waugh et al., 2006; Fine, 2011; Schneider et al., 2012; Stöven et al., 2016). The production and consumption of CFCs have been phased-out as a consequence of the implementation of the Montreal Protocol (MP) on Substances that Deplete the Ozone Layer (first in developed nations by 1996, followed by developing nations by 2010) designed to halt the degradation of the Earth's protective ozone layer (Fig. 1). The atmospheric mole frac
- tions of the major CFCs have been decreasing since the mid-1990s/mid-2000s (Carpenter et al., 2014; Bullister, 2015), and although CFCs are valuable indices to quantify deep water transport, the use of CFCs as oceanographic transient tracers has become more difficult for recently ventilated water masses. During recent decades sulfur hexafluoride (SF6) has been added to the suite of transient tracers measured in the ocean (Tanhua et al., 2004; Bullister et al., 2006). Its atmospheric mole fractions are still increasing and its atmospheric distribution is measured widely. However, SF6 is also facing restrictions; for example, in Europe it has been banned for release as a tracer gas and in all applications except high-voltage switchgear since 1 January 2006 (Fig. 1). Since a combination of transient tracers is needed to constrain ventilation (Waugh et al., 2002;
  - Stöven et al., 2015), it is necessary to expl